# DyBBT: Dynamic Balance via Bandit inspired Targeting for Dialog Policy with Cognitive Dual-Systems

## Abstract

Task oriented dialog systems often rely on static exploration strategies that do not adapt to dynamic dialog contexts, leading to inefficient exploration and suboptimal performance. We propose DyBBT, a novel dialog policy learning framework that formalizes the exploration challenge through a structured cognitive state space $\mathcal{C}$ that captures dialog progression, user uncertainty, and slot dependency. DyBBT proposes a bandit inspired meta-controller that dynamically switches between a fast intuitive inference (System 1) and a slow deliberative reasoner (System 2) based on real-time cognitive states and visitation counts. Extensive experiments on single- and multi-domain benchmarks show that DyBBT achieves state-of-the-art performance in success rate, efficiency, and generalization, with human evaluations confirming that its decisions are well aligned with expert judgment. The code is available at https://anonymous.4open.science/r/DyBBT-C6B7.

## 1 Introduction

> *"The affordances of the environment are what it offers the animal, what it provides or furnishes, for good or ill."*
> — James J. Gibson, The Ecological Approach to Visual Perception (1979)

Task oriented dialog system (TODS) assist users in achieving specific goals, like booking flights or reserving restaurants, via multi-turn natural language interactions. Dialog policy typically formulated as a sequential decision making problem addressed with Deep Reinforcement Learning (DRL) (Nachum et al., 2017; Silver et al., 2014), is bottlenecked by the exploration-exploitation dilemma: balancing exploitation of known rewards against exploration of unknown actions to discover better strategies. Unlike in standard RL, this dilemma in TODS is fundamentally exacerbated by its intrinsic cognitive structure, dynamic partially observable context characterized by quantifiable features such as the progress ratio of filled goal slots, the entropy of user intent over possible values, and the conditional dependency of unfilled slots on domain ontology (Peng et al., 2017; Wen et al., 2017). These features directly govern the cost benefit analysis of exploration: early in a dialog, high entropy makes information gathering actions valuable; late in dialog, high slot dependency makes exploitation critical to avoid constraint violations (Qin et al., 2023; Zhao et al., 2024).

Exploration in TODS is fundamentally challenging due to its dynamic, partially observable nature (Lee et al., 2023), characterized by three key cognitive properties that unfold in distinct dialog phases. Early dialog stages afford information gathering, as user goals are often ambiguous and multiple slots remain unfilled (Kwan et al., 2023); Mid-stages afford clarification and confirmation as slots begin to fill and dependencies emerge (Jia et al., 2024); and late stages afford task completion, where actions must adhere to strict slot-value dependencies, for example, a taxi cannot be booked without both "departure" and "destination" (Niu et al., 2024). This dynamic "affordance landscape" demands adaptive exploration: static strategies cause inefficiencies, premature exploitation fails tasks, while aimless exploration wastes turns.

Current methods for enhancing exploration in TODS, while powerful, are fundamentally misaligned with this dynamic cognitive reality. As illustrated in Figure 1, traditional DRL methods rely on static heuristics such as $\epsilon$-greedy (Niu et al., 2024), which cannot adapt to shifting exploration needs between dialog phases. Evolutionary methods like EIERL (Zhao et al., 2025) enable global search

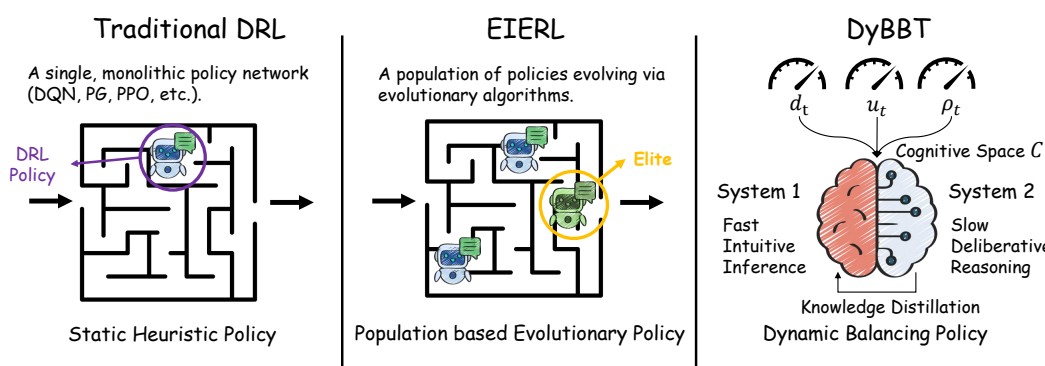

Figure 1: Traditional DRL methods (left) employ a static exploration strategy with a single policy. EIERL (middle) uses population based evolutionary optimization with elite injection but struggles to scale to complex multi domain tasks. DyBBT (right) introduces a cognitive meta-controller dynamically balances fast intuitive responses and slow deliberative reasoning for adaptive policy selection.

via population based optimization and elite injection to accelerate evolution, yet struggle in complex multi-domain scenarios due to poor scalability and unflexible updates. LLM based policies (Zhang et al., 2024; He et al., 2022) or reasoning techniques such as Tree of Thoughts (ToT) (Yao et al., 2023) support deep deliberative planning, but incur prohibitive computational overhead and lack a principled mechanism to trigger such costly reasoning only when necessary. This misalignment reveals a key Research Question: *How to design a dialog policy that dynamically perceives cognitive affordances to balance exploration and exploitation*?

To solve the above challenges, we propose DyBBT, a novel framework that grounds decisions in an interpretable cognitive state space $\mathcal{C}$ that captures dialog progress $d_t$, user uncertainty $u_t$, and slot dependency $p_t$, as shown in Figure 1. DyBBT introduces a lightweight meta-controller that dynamically switches between a fast System 1 (for routine decisions) and a slow System 2 (for costly deliberation) based on real-time cognitive signals and visitation counts. This design ensures that expensive reasoning is invoked only when the cognitive state signals under exploration or high uncertainty, addressing the core limitations (RQ) of previous methods. By formalizing dialog affordances and embedding them into a bandit inspired switching mechanism, DyBBT achieves a principled and efficient balance between exploration and exploitation.

In summary, our work makes the following contributions: (1) Formalization of TODS exploration challenge via a structured cognitive state space $\mathcal{C}$ (Section 3.1). (2) Proposal of DyBBT, a novel framework with bandit inspired meta-controller to dynamically balance between fast System 1 and deliberate System 2 reasoning (Section 3.2). (3) Demonstration of state-of-the-art (SOTA) performance and human aligned decisions through extensive experiments (Section 4).

## 2  RELATED WORK

### 2.1  DIALOG POLICY LEARNING WITH DEEP REINFORCEMENT LEARNING

Deep Reinforcement Learning (DRL) has become a dominant paradigm for dialog policy optimization due to its capacity for sequential decision making. Early work applied value based methods (Peng et al., 2018) and Policy Gradient (Silver et al., 2014) to TODS, Proximal Policy Optimization (PPO) (Schulman et al., 2017) was later adopted for improved stability and has become a common baseline. A key limitation of these methods is their reliance on static exploration strategies, such as $\epsilon$-greedy or entropy bonus. These heuristics cannot adapt to the dynamic uncertainty and structural complexity of multi-domain dialogs (Kwan et al., 2023; Jia et al., 2024). Recent efforts have incorporated Bayesian reasoning (Lee et al., 2023), meta-learning (Li et al., 2024; Liang et al., 2025), Cascading RL (Du et al., 2024), CB-RL (Thoma et al., 2025) to allow more adaptive exploration. While promising, they often lack an explicit and interpretable representation of the internal

cognitive dialog state that directly governs exploration, a gap our cognitive state space $\mathcal{C}$ aims to fill.

## 2.2 Evolutionary and Population Based Methods for Exploration

Evolutionary Reinforcement Learning (ERL) combines population based global search with gradient-based optimization to enhance exploration diversity. Methods such as EIERL (Zhao et al., 2025) inject elite policies to accelerate evolution, enabling escape from local optima. However, ERL scales poorly with dialog complexity due to exponential growth in population size (Sigaud, 2023). Moreover, these methods often rely on fixed schedules for policy replacement, lacking dynamic adaptation to real-time dialog progression and cognitive state changes (Bai et al., 2023). In contrast, DyBBT replaces expensive population evolution with a single, efficient dual-system architecture guided by a structured cognitive state space, enabling fine grained, context aware exploration without the scalability limitations of population based approaches.

## 2.3 Classical and Modern Exploration Theories

The exploration-exploitation trade-off is a cornerstone of sequential decision theory. Bandit algorithms, such as Upper Confidence Bound (UCB) (Garivier & Moulines, 2011), provide theoretical guarantees for stationary settings, and their principles have been extended to contextual bandits (Foster & Rakhlin, 2020) and hierarchical RL (Rohmatillah & Chien, 2023). However, directly applying these theories to dialog Partially Observable Markov Decision Processes (POMDPs) faces significant challenges due to non-stationarity, partial observability, and the high dimensional nature of the state space. Our work draws inspiration from the optimism principle of UCB but makes a pragmatic *heuristic adaptation* to a learned cognitive state space $\mathcal{C}$. This approach preserves the interpretability and theoretical intuition of bandit algorithms while specifically addressing the complexities of sequential dialog environments. Compared to methods like PSRL (Chen et al., 2020) that require maintaining a posterior over the entire MDP, our method focuses exploration on a compact cognitive space, offering a computationally efficient alternative better suited to dialog POMDPs.

## 2.4 Dual-System Architectures and LLMs for dialog Policies

Krämer (2014), combine fast, intuitive processing (System 1) with slow, deliberative reasoning (System 2), have been applied to mathematical reasoning (Shi et al., 2024) and common sense inference (Yu et al., 2025). In dialog systems, large language models (LLMs) serve as powerful function approximators (Yi et al., 2024), acting as intuitive generators (Ying et al., 2024) and deliberative reasoners (Ma et al., 2025). Recent work, such as the Dynamic Dual-Process Transformer (He et al., 2024), explicitly models the interaction for dialog policy learning. However, existing switching mechanisms often rely on static heuristics, such as fixed turn counts (Qin et al., 2023) or predefined confidence thresholds (Yao et al., 2023), which lack adaptability and theoretical grounding in exploration. DyBBT addresses this by introducing a meta-controller guided by a bandit inspired principle, dynamically triggering System 2 based on cognitive state visitation counts and parametric uncertainty, offering a principled and efficient alternative to heuristic switching.

## 3 Methodology

To answer the key research question, we present DyBBT: a framework that formalizes dialog exploration as a tractable Contextual Multi-Armed Bandit (CMAB) problem over a structured cognitive state space $\mathcal{C}$, grounded theoretically by a Lipschitz smooth reward assumption and a sublinear regret bound derived from visitation based exploration. This theoretical foundation informs the design of a lightweight meta-controller, which dynamically switches between System 1 and System 2 via a dual trigger mechanism, balancing epistemic exploration and aleatoric uncertainty.

### 3.1 Theoretical Foundation

This section establishes the theoretical foundations of DyBBT by formalizing dialog exploration as a tractable CMAB problem over a structured cognitive state space $\mathcal{C}$. While the full dialog POMDP is intractable for rigorous analysis, we bridge this gap through three principled approximations: (1) compressing the high dimensional dialog state into a low dimensional cognitive representation $\mathcal{C}$; (2) assuming Lipschitz smoothness to enable theoretical guarantees; (3) deriving a bandit inspired exploration criterion that guides our meta-controller design. This approach provides a theoretically grounded, yet practical foundation for adaptive exploration in dialog systems.

### 3.1.1 Contextual Multi-Armed Bandit Formulation

To make the exploration-exploitation trade-off analytically tractable, we frame dialog policy learning as a CMAB problem (Foster & Rakhlin, 2020). The key innovation lies in our structured *cognitive state space* $\mathcal{C}$, which bridges bandit exploration principles with dialog POMDPs by compressing the high-dimensional belief state into an interpretable low-dimensional representation.

In this CMAB formulation, the **arms** correspond to a binary set $\mathcal{A} = \{S1, S2\}$ where the two options are fast inference S1 and deliberative reasoning S2. The **context** is defined as the cognitive state $\mathbf{c}_t = [d_t, u_t, \rho_t] \in \mathcal{C}$, which quantifies dialog progress, user uncertainty, and slot dependency at dialog turn $t$ (see Appendix A.1 for computation); this low-dimensional vector captures essential dialog dynamics. The **reward** $r_t(a)$ reflects task progress and efficiency (Formulations in 4.1) when selecting arm $a \in \mathcal{A}$ in context $\mathbf{c}_t$. The learning **objective** is to minimize the cumulative regret:

$$R_T = \sum_{t=1}^{T} \left[ \mathbb{E}[r_t(a_t^* \mid \mathbf{c}_t)] - \mathbb{E}[r_t(a_t \mid \mathbf{c}_t)] \right], \tag{1}$$

where $a_t^*$ denotes the optimal arm selection and $a_t$ our algorithm's choice at turn $t$.

This CMAB formulation provides a framework for analyzing exploration efficiency. We treat System 2 as an *oracle-like arm* that, when pulled, aggressively pursues the optimal action $a_t^*$ to minimize regret in unexplored regions, and this shapes our meta-controller architecture in Section 3.2.3.

### 3.1.2 Reward Smoothness: A Pragmatic Assumption for Structured Tasks

To enable principled exploration over the cognitive state space $\mathcal{C}$ within the CMAB framework, we require the reward function to exhibit structural regularity. The standard Lipschitz continuity (Asadi et al., 2018; Pazis & Parr, 2013; Ortner & Ryabko, 2012) assumption is a crucial condition for deriving sublinear regret bounds in continuous spaces. We therefore adopt it, as it guarantees similar rewards for nearby cognitive states.

**Assumption 1** (Lipschitz Smooth Reward in $\mathcal{C}$). *The expected immediate reward $\bar{r}(\mathbf{c}, a) = \mathbb{E}[r(s_t, a_t)|\mathbf{c}_t = \mathbf{c}]$ is Lipschitz continuous with respect to the cognitive state $\mathbf{c}$ for any action $a$. That is, there exists a constant $L_r > 0$ such that:*

$$|\bar{r}(\mathbf{c}, a) - \bar{r}(\mathbf{c}', a)| \leq L_r \cdot d(\mathbf{c}, \mathbf{c}'), \quad \forall \mathbf{c}, \mathbf{c}' \in \mathcal{C}.$$

This serves as the theoretical cornerstone of DyBBT. Without it, the visitation count $n_t(\mathbf{c}_t)$ would lose its semantic meaning as an uncertainty metric, as observing one state would not provide no information about its neighborhood. This enables the transfer of bandit exploration principles to dialog POMDPs. We provide empirical validation of this assumption's practical relevance in Section 5.4.

### 3.1.3 Dynamic Balance Principle: From Regret Bounds to Switching Rules

Building upon Assumption 1, making visitation counts a meaningful measure of epistemic uncertainty, we now derive a principled exploration criterion for the meta-controller. This enables us to formalize the exploration-exploitation trade-off through the lens of contextual bandits (Kleinberg et al., 2008; Bubeck et al., 2011), where the exploration bonus for cognitive state $\mathbf{c}_t$ takes the form:

$$\text{Exploration-Bonus}(t) \propto \sqrt{\frac{\log T}{n_t(\mathbf{c}_t)}}. \tag{2}$$

where $T$ denotes total training steps and $n_t(\mathbf{c}_t)$ represents the visitation count of $\mathbf{c}_t$. This formulation adapts the Upper Confidence Bound (UCB) principle (Ortner & Ryabko, 2012; Foster & Rakhlin, 2020) to structured cognitive space. The square root dependence arises from concentration inequalities underlying bandit theory (Komiyama et al., 2024), while the logarithmic factor accommodates the time horizon.

Equation 2 provides theoretical motivation for our meta-controller design. To transform into a practical switching rule, we note that System 2 should be invoked when the exploration bonus exceeds a certain threshold. This leads naturally to Condition 1 ($n_t(\mathbf{c}_t) < \tau\sqrt{\log T}$), where the thresholdcorresponds to the confidence radius in UCB algorithms, ensuring exploration occurs when potential information gain justifies the computational cost of System 2.

Under Assumption 1 and the approximate MDP structure in $\mathcal{C}$, the exploration strategy based on $n_t(\mathbf{c}_t)$ achieves the expected cumulative regret, whose bound is sublinear (proof sketch in Appendix A.2). It demonstrates that exploration in the low-dimensional cognitive space $\mathcal{C}$ is both

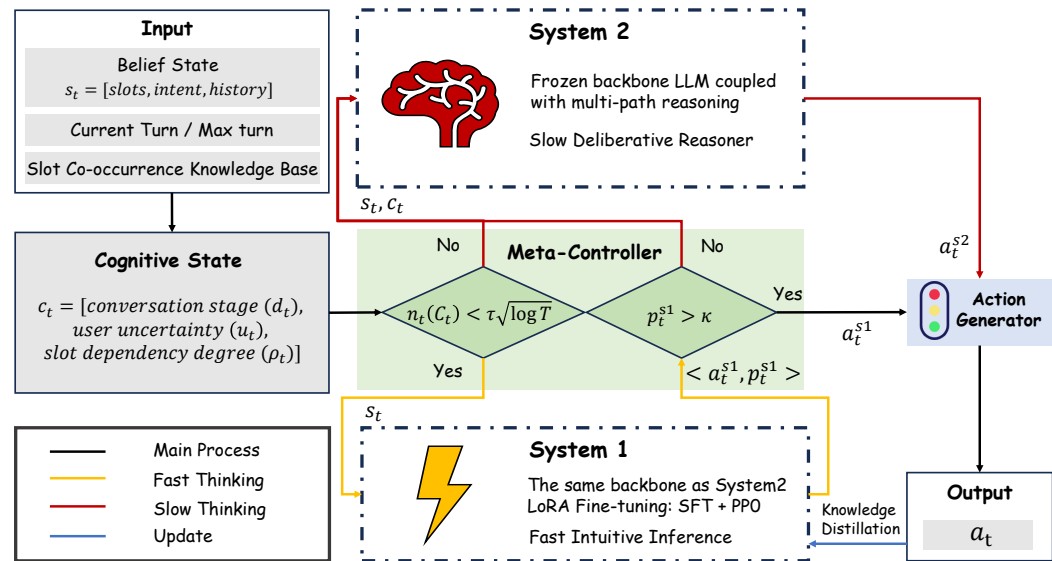

Figure 2: The DyBBT Architecture. A meta-controller uses the cognitive state $\mathbf{c}_t$, visitation count $n_t(\mathbf{c}_t)$, and System 1's confidence $p_t^{S1}$ to dynamically select between System 1 (fast intuitive) and System 2 (slow deliberative). Outputs drive action execution and update visitation/distillation buffers for continuous learning.

efficient and principled, bridging bandit theory with dialog POMDPs through structured state compression and smoothness assumptions.

## 3.2 SYSTEM ARCHITECTURE

Building on the theoretical foundation, DyBBT as shown in Figure 2, operationalizes the CMAB formulation over the cognitive state space $\mathcal{C}$ into a dual-system architecture. The meta-controller directly instantiates the bandit-inspired switching rule (Eq. 2) to dynamically balance between fast intuitive S1 and slow deliberative S2. This principled design ensures expensive S2 is invoked only when cognitive signals and visitation counts indicate high epistemic uncertainty or low confidence, achieving adaptive exploration-exploitation trade-off while maintaining computational efficiency.

### 3.2.1 SYSTEM 1 (S1): THE FAST INTUITIVE INFERENCE

To provide a low latency, high throughput baseline policy for the majority of dialog turns, mitigating the prohibitive cost of always using a deliberative reasoner, S1 embodies the fast and intuitive system. The prompt (in Appendix B.4.1) induce the LLMs to output system actions and confidece sore. in TODS **action** $a_t^{S1}$ represents the system operation at each turn, formalized as a tuple comprising an action type, domain, and target slot (e.g., `request(restaurant, area)`). The **confidence score** $p_t^{S1} \in [0,1]$ is S1's self-assessed certainty in its chosen action $a_t^{S1}$. This score provides a crucial measure of *aleatoric uncertainty* that complements the *epistemic uncertainty* captured by visitation counts in the meta-controller. S1 undergoes a two-stage training process (detailed in Appendix B.5.2). SFT on expert trajectories trains the model to predict both the action $a_t^{S1}$ and a calibrated confidence score $p_t^{S1}$. PPO refines the policy to maximize task success and efficiency.

### 3.2.2 SYSTEM 2 (S2): THE SLOW DELIBERATIVE REASONER

To handle novel or complex situations where fast policy (S1) is likely to fail, thus addressing the suboptimal performance of static DRL policies in under explored regions, S2 represents the slow and analytical system. It utilizes the same base model as S1, but remains frozen to preserve its broad knowledge and reasoning capabilities. The prompt instructs S2 to generate Top-3 distinct action sequences. Each sequence's quality is evaluated using the ratio of filled key slots. We extract the first action from the highest quality sequence as output $a_t^{S2}$. This system is computationally expensive but is designed to handle novel or high stakes situations identified by the meta-controller.

### 3.2.3 Meta-Controller: Dynamic Orchestration via Bandit-Inspired Switching

The meta-controller operationalizes the theoretical principles from Section 3.1.3 by dynamically selecting between System 1 and System 2 based on real-time cognitive signals. Its design directly instantiates the bandit-inspired exploration criterion derived in Eq. 2. The transition from theoretical foundation to implementation involves the meta-controller implementing a dual-trigger mechanism that bridges bandit theory with practical dialog POMDPs.

$$\text{Activate System 2 IF:} \quad \underbrace{n_t(\mathbf{c_t}) < \tau \sqrt{\log T}}_{\text{Condition 1: Exploration Condition}} \quad \vee \quad \underbrace{p_t^{S1} < \kappa}_{\text{Condition 2: Confidence Condition}} \tag{3}$$

**Condition 1: Exploration Condition.** This condition directly implements the UCB-inspired exploration bonus from Eq. 2. Under Assumption 1, low visitation counts in cognitive region $\mathbf{c}_t$ indicate high epistemic uncertainty, justifying systematic exploration via System 2. The threshold $\tau \sqrt{\log T}$ adapts the classical bandit confidence radius to our structured cognitive space, ensuring exploration occurs when potential information gain outweighs computational cost.

**Condition 2: Confidence Condition.** While Condition 1 addresses reducible epistemic uncertainty, Condition 2 provides robustness against irreducible aleatoric uncertainty arising from partial observability and model limitations. Empirical studies (Kadavath et al., 2022; Lin et al., 2022; Yin et al., 2023) demonstrate that LLM confidence scores correlate with calibration quality, making $p_t^{S1}$ an effective proxy for situations where System 1's parametric knowledge is insufficient.

This hybrid design acknowledges that while our cognitive state compression enables tractable exploration (via Condition 1), practical dialog POMDPs require additional safeguards against model limitations (via Condition 2). The disjunctive combination ensures System 2 activation for either systematic exploration or robustness, creating an adaptive balance that outperforms either condition alone, as validated in our ablation study (Table 2).

The meta-controller's decisions drive a closed-loop system where high-quality System 2 demonstrations are distilled back into System 1 through knowledge distillation (Appendix B.5.3), creating a virtuous cycle of policy improvement while reducing long term dependence on costly deliberation.

## 4 Experiment

### 4.1 Experimental Setup

**Datasets.** We conduct experiments on two of the most prominent TODS benchmark datasets which are also used in baselines. The Microsoft Dialog Challenge platform (Li et al., 2018; Zhao et al., 2024; Niu et al., 2024) for single domain, while the MultiWOZ2.1 dataset (Budzianowski et al., 2018) for multi domains. Statistics in Appendix B.1.

**Baselines.** We compare DyBBT against four kinds of comprehensive suite of strong and recent baselines to ensure a rigorous evaluation, and details in Appendix B.2. **DRL Series**: DQN_$\epsilon$_$N$ (agents are trained using standard DQN with a traditional $\epsilon$-greedy exploration strategy, where $\epsilon = N$ (Mnih et al., 2015)), NOISY_DQN (agents enhance exploration by introducing noise into the network weights (Han et al., 2022)), PG (REINFORCE, a stochastic gradient algorithm for policy gradient reinforcement learning (Zhu et al., 2023)), PPO (A policy optimization method in policy based reinforcement learning that uses multiple epochs of stochastic gradient ascent and a constant clipping mechanism as the soft constraint to perform each policy update.Zhu et al. (2023)). **LLM based DP**: LLM_DP (agents use the DP module with GPT-4.0 (Yi et al., 2024)), AutoTOD (a zero-Shot autonomous agent with GPT-4.0 (Xu et al., 2024), ProTOD (proactive dialog policy based on GPT-4.0 (Dong et al., 2025)). **ERL**: EIERL(evolutionary reinforcement learning injected by elite individuals (Zhao et al., 2025)). **Multi Agent Collaborative**: MACRM (a multi agent curiosity reward mode for dialog policy (Sun et al., 2025))

**Evaluation Metrics.** For single-domain tasks: success rate, average turns, and reward (following EIERL (Zhao et al., 2025): $+2t$ for success, $-t$ for failure, $-1$ for every turn). For multi domain: Inform, Success, Book rates, and Avg. Turns (formulas in Appendix B.3).

**Implementation Details.** Following EIERL for fair comparison, dialogs are capped at 30 (single domain) and 40 (multi domain) turns. Training runs for 500 epochs (single) and 10K epochs (multi). DyBBT uses the same Qwen3 (0.6B–8B) for both S1 and S2. Full details in Appendix B.5.

Table 1: Evaluation results for all agents across the three single domain datasets are provided, with the highest value in each metric column highlighted in bold. Epochs (50, 250, 500) represent early, mid, and post convergence training stages. Baselines sourced from Zhao et al. (2025).

| Domain | Agent | Epoch = 50 | | | Epoch = 250 | | | Epoch = 500 | | |
|---|---|---|---|---|---|---|---|---|---|---|
| | | Success↑ | Reward↑ | Turns↓ | Success↑ | Reward↑ | Turns↓ | Success↑ | Reward↑ | Turns↓ |
| Movie | DQN_ε_0.0 | 35.05 | -13.00 | 32.11 | 54.03 | 12.99 | 25.70 | 55.53 | 14.95 | 25.37 |
| | DQN_ε_0.05 | 30.93 | -18.61 | 33.44 | 67.95 | 31.84 | 21.39 | 76.68 | 43.42 | 19.21 |
| | NOISY_DQN | 41.37 | -4.73 | 30.75 | 71.41 | 36.68 | 20.04 | 72.80 | 39.38 | 20.16 |
| | LLM_DP | 41.56 | -3.09 | 27.34 | 41.56 | -3.09 | 27.34 | 41.56 | -3.09 | 27.34 |
| | EIERL | 23.72 | -27.53 | 34.01 | 80.33 | 48.21 | 18.36 | 85.52 | 55.29 | 16.66 |
| | DyBBT-0.6B | 50.12 | 32.45 | 22.13 | 70.23 | 45.37 | 18.24 | 80.34 | 51.82 | 16.79 |
| | DyBBT-1.7B | 55.15 | 35.68 | 21.18 | 75.28 | 48.59 | 17.63 | 83.42 | 53.77 | 16.12 |
| | DyBBT-4B | 60.21 | 38.91 | 20.14 | 80.35 | 51.83 | 17.15 | 86.47 | 55.71 | 15.64 |
| | DyBBT-8B | **65.24** | **42.14** | **19.17** | **85.39** | **55.06** | **16.18** | **89.52** | **57.64** | **15.13** |
| Rest. | DQN_ε_0.0 | 06.95 | -36.57 | 27.66 | 49.07 | 4.10 | 22.13 | 56.71 | 11.63 | 23.22 |
| | DQN_ε_0.05 | 07.26 | -36.28 | 27.63 | 57.12 | 12.30 | 20.21 | 57.17 | 12.79 | 21.12 |
| | NOISY_DQN | 00.00 | -43.92 | 29.84 | 16.69 | -28.25 | 28.55 | 29.88 | -15.20 | 26.18 |
| | LLM_DP | 38.96 | -5.96 | 20.16 | 38.96 | -5.96 | 29.16 | 38.96 | -5.96 | 29.16 |
| | EIERL | 01.81 | -41.09 | 27.44 | 69.75 | 24.79 | 17.98 | 79.35 | 34.99 | 16.07 |
| | DyBBT-0.6B | 46.73 | 20.5 | 21.67 | 65.44 | 28.83 | 17.86 | 74.85 | 33.08 | 16.52 |
| | DyBBT-1.7B | 51.32 | 22.59 | 20.71 | 70.14 | 30.90 | 17.25 | 77.71 | 34.24 | 15.85 |
| | DyBBT-4B | 56.03 | 24.68 | 19.67 | 74.86 | 32.98 | 16.78 | 80.55 | 35.49 | 15.37 |
| | DyBBT-8B | **60.70** | **26.74** | **18.69** | **79.54** | **35.05** | **15.81** | **83.38** | **36.74** | **14.86** |
| Taxi | DQN_ε_0.0 | 00.04 | -42.69 | 27.47 | 48.46 | 2.26 | 24.70 | 58.79 | 12.38 | 23.06 |
| | DQN_ε_0.05 | 00.00 | -42.86 | 27.71 | 55.98 | 8.19 | 22.38 | 66.83 | 20.19 | 21.90 |
| | NOISY_DQN | 00.00 | -43.73 | 29.46 | 14.55 | -30.56 | 29.32 | 26.15 | -19.46 | 28.00 |
| | LLM_DP | 34.96 | -10.23 | 25.95 | 34.96 | -10.23 | 25.95 | 34.96 | -10.23 | 25.95 |
| | EIERL | 00.00 | -41.55 | 25.10 | 56.38 | 9.26 | 21.96 | 81.59 | 35.39 | 17.29 |
| | DyBBT-0.6B | 47.93 | 20.77 | 22.67 | 67.13 | 29.10 | 18.76 | 76.77 | 33.29 | 17.32 |
| | DyBBT-1.7B | 52.74 | 22.86 | 21.71 | 71.95 | 31.20 | 18.15 | 79.71 | 34.56 | 16.65 |
| | DyBBT-4B | 57.57 | 24.95 | 20.67 | 76.78 | 33.29 | 17.68 | 82.62 | 35.83 | 16.17 |
| | DyBBT-8B | **62.37** | **27.04** | **19.69** | **81.59** | **35.38** | **16.71** | **85.53** | **37.09** | **15.66** |

## 4.2 MAIN RESULTS

### 4.2.1 PERFORMANCE ON SINGLE DOMAIN TASKS

The evaluation results on single domain dialog tasks are presented in Table 1. DyBBT demonstrates strong performance across all three domains. The results reveal that DyBBT's cognitive enables more efficient policy learning: by dynamically allocating computational resources based on real-time cognitive signals, DyBBT achieves higher task success with significantly fewer dialog turns compared to methods relying on static exploration heuristics, population level evolution or GPT-4 based policy. This efficiency gain is particularly pronounced in complex domains like Taxi, where slot dependencies create challenging exploration landscapes that DyBBT navigates more effectively through its principled switching mechanism.

### 4.2.2 PERFORMANCE ON MULTI DOMAIN TASK

Results on the challenging MultiWOZ dataset are provided in Table 6 (Appendix E.1). While EIERL's success rate drops significantly in this complex multi domain setting, highlighting the scalability limits of its population based approach, DyBBT maintains strong performance. DyBBT-8B performs slightly better than AutoTOD/ProTOD, and using GPT-4 as S2 yields SOTA results, showing that DyBBT matches strong LLM baselines while being more efficient. This is enabled by the structured cognitive state and dual system design, which provide a domain agnostic inductive bias without requiring task specific tuning. Cost effectiveness analysis is discussed in Appendix E.7.

### 4.2.3 TRAINING EFFICIENCY AND CONVERGENCE

Figure 3 illustrates the learning curves of DyBBT compared to baselines. DyBBT converges faster and achieves higher asymptotic performance across all domains, outperforming EIERL significantly at epoch 50. This accelerated learning stems from the meta-controller's active guidance of exploration from the outset, which systematically targets under explored or uncertain regions in $\mathcal{C}$ rather than relying on random exploration or high-variance evolutionary mechanisms.

Furthermore, DyBBT exhibits consistent scaling with model size, for instance, success rates improve from 80.34% to 89.52% in the single domain Movie task and from 78.2% to 84.1% in the multi domain setting when scaling from 0.6B to 8B parameters. This trend indicates that the dual-system architecture effectively harnesses the increased representational capacity of larger backbone models. When coupled with the meta-controller's efficient resource allocation with Qwen3's native switching mechanism in balancing performance and computational cost (Appendix E.8). DyBBT underscores its practical viability for real-world deployment.

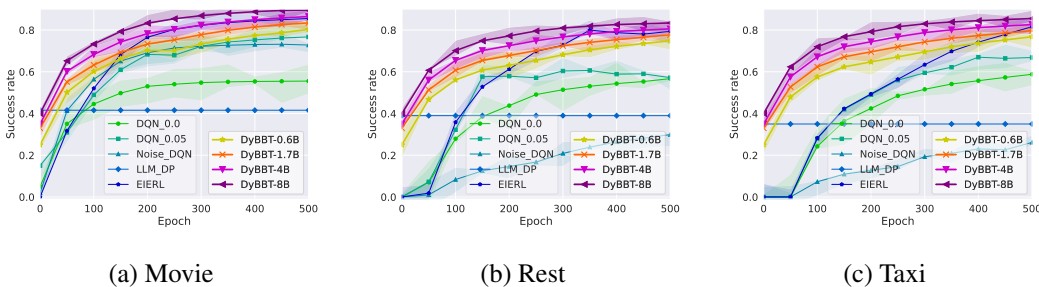

| (a) Movie | (b) Rest | (c) Taxi |

Figure 3: Learning curves for training efficiency and convergence across single-domain TODS tasks.

Table 2: Ablation study of DyBBT's components on MultiWOZ. Results underscore the necessity of the meta-controller and the structured cognitive state representation for optimal performance.

| Variant | Inform↑ | Success↑ | Book↑ | Turns↓ |
|---------|---------|----------|-------|--------|
| DyBBT-8B (full) | **91.2** | **84.1** | **86.9** | **14.6** |
| w/o Meta-Controller | 82.5 | 71.8 | 77.3 | 17.5 |
| w/o System 2 | 85.7 | 76.3 | 80.1 | 16.8 |
| w/ Learned Cognitive State | 90.5 | 83.2 | 86.3 | 14.8 |
| w/o Knowledge Distillation | 89.8 | 82.4 | 85.7 | 15.1 |
| w/o Cognitive State (raw $s_t$) | 84.2 | 75.1 | 79.6 | 17.1 |
| w/o Exploration Condition (EC) | 90.1 | 82.9 | 86.1 | 14.9 |
| w/o Confidence Condition (CC) | 87.6 | 79.5 | 83.2 | 16.2 |
| w/o dialog Progress ($d_t$) | 88.9 | 80.7 | 84.5 | 15.7 |
| w/o User Uncertainty ($u_t$) | 89.6 | 81.9 | 85.3 | 15.3 |
| w/o Slot Dependency ($\rho_t$) | 90.3 | 82.5 | 85.9 | 15.0 |

### 4.2.4 SUMMARY OF STRENGTHS

The main results demonstrate that DyBBT achieves state-of-the-art performance through: **Dynamic Exploration-Exploitation Balance:** The meta-controller's bandit inspired switching rule allows DyBBT to dynamically allocate expensive S2 reasoning only when necessary, leading to highly efficient exploration. **Scalability with Model Size:** DyBBT benefits predictably from larger backbone models, making it well suited for future advancements in LLM capabilities. **Strong Generalization:** Consistent performance across both single and multi domain tasks shows that the cognitive state representation captures universal dialog dynamics. **Computational Practicality:** Unlike population based methods (EIERL) or full GPT-4.0 approaches, DyBBT maintains moderate computational overhead during both training and inference.

### 4.3 ABLATION EXPERIMENT

Ablation results are shown in Table 2, and detailed settings are in Appendix E.2. The results revealing that: **Meta-Controller is crucial.** Removing it causes the most severe performance degradation, confirming its essential role in dynamically orchestrating the exploration-exploitation trade-off. **Both conditions are necessary but asymmetric:** Removing Condition 1 (EC) eliminates the bandit inspired exploration bonus from Equation 2, while removing Condition 2 (CC) disables the aleatoric uncertainty safeguard, a distinction rooted in Bayesian RL theory (Dearden et al., 1998). Removing the confidence condition (CC) causes a more substantial performance drop than removing the exploration condition (EC), validating our hybrid design. This indicates that mitigating S1's over-confidence is slightly more critical than targeted exploration for robust performance. In depth error analysis (Appendix E.3) reveals that CC primarily prevents catastrophic failures in states with high cognitive uncertainty. **Cognitive State design is vital.** Replacing it with the raw belief state causes catastrophic performance collapse, confirming the necessity of our low dimensional, interpretable representation. While the learned alternative performs reasonably well, it still underperforms our hand-designed features, justifying our cognitively inspired approach. **All state dimensions contribute meaningfully.** Removing any single dimension causes noticeable performance degradation, with dialog progress ($d_t$) being the most impactful individual component, followed by user uncertainty ($u_t$) and slot dependency ($\rho_t$). **Knowledge Distillation enables continuous improvement.**

Disabling it reduces final performance, confirming its role in facilitating long term efficiency gains through systematic learning from S2's demonstrations.

### 4.4 Human and Real World Evaluation

To complement automated metrics and validate the practical efficacy of DyBBT, we conducted both controlled human evaluations and real-world user experiments.

We conduct a human evaluation (details in Appendix C) focusing on the meta-controller's switching decisions. 10 NLP researchers evaluated 200 dialog states from MultiWOZ, comparing DyBBT against random switching and System 1 only baselines. Annotators assessed action appropriateness (5 point Likert scale) and whether invoking System 2 was justified (binary judgment). The results show that DyBBT's actions are more appropriate than both baselines. Its decisions to invoke System 2 align substantially better with human judgment than random switching, providing qualitative evidence that our meta-controller effectively identifies when deliberation is warranted a key affordance often missed by heuristic approaches.

Real world experiments (details in Appendix D) with 30 volunteers further validated these findings. DyBBT maintained the highest task success rate and user satisfaction in authentic multi-domain interactions, demonstrating that its cognitive state representation $\mathcal{C}$ generalizes effectively beyond simulated environments. Case studies revealed that DyBBT successfully handles challenging scenarios like mid-dialog intent shifts and vague user expressions through adaptive System 2 invocation.

Collectively, these results provide converging evidence that DyBBT's meta-controller effectively translates cognitive affordances into a dynamic exploration exploitation balance, enabling robust performance in both controlled and real world settings.

## 5 Analysis

Our experimental results demonstrate that DyBBT achieves state-of-the-art performance on multiple benchmarks. In this section, we analyze the underlying mechanisms that enable DyBBT's effectiveness, providing insights into why and how our framework works.

### 5.1 The Emergent Structure of Cognitive State Space

The cognitive state space $\mathcal{C}$ serves as the foundational bridge that enables the transfer of bandit exploration principles to the complex dialog POMDP. To empirically validate its utility, we analyze the *visitation frequency* of different regions within the discretized $\mathcal{C}$ over training (Fig. 4; detailed computation in Appendix B.5.4). The heatmap reveals a highly structured, non-uniform occupancy pattern, directly validating our core hypothesis. The meta-controller's exploration is not random but strategically focused: in the **early dialog phase** ($d_t \in [0.0, 0.2]$), it broadly explores across user uncertainty ($u_t$) for information gathering. In the **mid-phase** ($d_t \in [0.4, 0.6]$), visitation concentrates in regions of **medium-to-high** $u_t$, targeting ambiguity resolution. In the **late phase** ($d_t > 0.8$), activity focuses on states with **low** $u_t$, exploiting known information to complete tasks.

This phase dependent targeting demonstrates that $\mathcal{C}$ successfully captures the dialog's dynamic "affordances". The meta-controller learns to allocate its exploration budget to the most relevant regions of $\mathcal{C}$ for the current dialog stage, enabling highly efficient and context aware exploration. The effectiveness of $\mathcal{C}$ stems from its ability to distill the high dimensional belief state into a low dimensional, actionable representation, making principled exploration computationally feasible.

### 5.2 Adaptive Balancing Through Dual Triggers

The meta-controller's hybrid triggering mechanism provides a robust solution to the exploration-exploitation dilemma by responding to different types of uncertainty:

**Epistemic vs. Aleatoric Uncertainty Distinction:** Two trigger conditions address fundamentally different types of uncertainty. The exploration condition ($n_t(\mathbf{c}_t) < \tau\sqrt{\log T}$) targets *epistemic uncertainty*, lack of knowledge about the environment that can be reduced through exploration. The confidence condition ($p_t^{S1} < \kappa$) addresses *aleatoric uncertainty*, inherent stochasticity or model limitations irreducible via exploration alone. **Complementary Trigger Patterns:** Analyzing 10,000 dialog turns reveals complementary triggering patterns (Fig. 5 in Appendix E.4). The exploration condition dominates in early training phases and for novel state regions, enabling systematic coverage of

the state space. The confidence condition acts as a consistent safety net throughout training, preventing overreliance on a potentially flawed System 1. This complementary design ensures robustness across diverse dialog scenarios. **Progressive Adaptation:** The triggering rate evolves naturally with training progress. Initially, frequent System 2 invocations offer guided exploration and high quality demos. As training progresses and System 1 improves through distillation, the meta-controller automatically reduces System 2 usage, transitioning from guided exploration to autonomous operation. This adaptive balancing is key to DyBBT's computational efficiency and crucially, it is the core manifestation of DyBBT's ability to perceive and respond to the dynamic "affordances" of the dialog environment, ensuring the right cognitive system is invoked at the right time.

### 5.3 KNOWLEDGE DISTILLATION AS IMPLICIT POLICY IMPROVEMENT

The knowledge distillation process creates a virtuous cycle that enables continuous policy improvement without additional environment interactions. The effectiveness of distillation is evidenced by the monotonic improvement of System 1 and corresponding reduction in System 2 invocation rate (Fig. 6 in Appendix E.4), demonstrating successful knowledge transfer.

### 5.4 THEORETICAL INTUITIONS AND EMPIRICAL ALIGNMENT

Our theoretical analysis, though based on simplifying assumptions, is pragmatically validated by empirical results: **Sublinear Regret as Validation of Core Assumptions.** The empirical cumulative regret (Fig. 7) exhibits $\sqrt{T}$-like growth. This sublinear trend is not merely observational; it provides indirect empirical support for our key theoretical assumptions: The Lipschitz continuity of the reward in $\mathcal{C}$ (Assumption 1), and the approximate structure of MDP over $\mathcal{C}$ (Assumption 2). The alignment between theory and experiment suggests $\mathcal{C}$ effectively captures the latent structure enabling efficient exploration. **Low Dimensional $\mathcal{C}$ Enables Practical Implementation.** The consistent high performance of DyBBT using only a three dimensional cognitive state demonstrates that the essential features governing exploration (dialog progress, user uncertainty, slot dependency) can be distilled into a compact representation. This reduction in dimensionality is theoretically motivated by the dependence of the regret bound's $\sqrt{\dim(\mathcal{C})}$ (Appendix A.2.2).

### 5.5 FAILURE MODE ANALYSIS AND LIMITATIONS

Despite its strong performance, DyBBT exhibits three key failure modes that constrain its robustness, as empirically validated through quantitative and qualitative analyses in Appendix E.9 and E.10. First, the framework is over reliant on cognitive state fidelity. The handcrafted $\mathbf{c}_t$ can misrepresent complex dialog dynamics, leading the meta-controller to misjudge System 2 invocation. This results in underexploration or computational waste. Second, it depends on high quality System 2 demonstrations. Errors in reasoning or self evaluation can propagate to System 1 via knowledge distillation, causing subtle cascading policy corruption. Third, sensitivity to discretization. Heuristic quantization of $\mathcal{C}$ into 5 bins, masks critical state variations, treating strategically distinct states identically and reducing exploration efficacy. Quantitative analysis reveals that these failures affect only 5.2% of dialogs, primarily in edge cases with abrupt intent shifts or complex dependencies, while built in safeguards provide substantial mitigation. Qualitative case studies illustrate these modes concretely, showing how failures arise from unrepresented dialog nuances and how successful interventions align with human judgment. Collectively, these experiments demonstrate a tension between DyBBT's theory driven design and practical dialog complexities, underscoring the need for future work on learned representations and adaptive mechanisms.

## 6 CONCLUSION

DyBBT introduces a principled, cognitively grounded framework for dialog policy learning that dynamically balances exploration and exploitation through a bandit inspired meta-controller operating over a structured cognitive state space. By formalizing dialog affordances, phasic progression, user uncertainty, and slot dependency, our approach enables adaptive, context aware switching between fast intuitive responses and deliberate reasoning. Extensive experiments demonstrate state-of-the-art performance across single and multi domain benchmarks, with human evaluations confirming superior decision quality and alignment with expert judgment. DyBBT offers a scalable, efficient, and interpretable alternative to static or population based methods, bridging cognitive theory with practical dialog optimization. Future work will focus on learning cognitive representations end-to-end and extending the framework to more complex interactive settings.

ETHICS STATEMENT

This work presents a dialog policy learning framework evaluated on publicly available benchmark datasets (MS Dialog and MultiWOZ). Our research does not involve human subjects beyond the use of standard datasets, and all experiments are conducted through simulated user interactions. The proposed methodology focuses on improving the efficiency of task oriented dialog systems, with potential positive societal impacts through enhanced human computer interaction. We are unaware of any specific ethical concerns or negative social impacts directly arising from this work.

REPRODUCIBILITY STATEMENT

To ensure reproducibility, we have made our code and datasets publicly available at https://anonymous.4open.science/r/DyBBT-C6B7. The appendix provides comprehensive implementation details, including: hyperparameters (Section B.5), dataset statistics (Section B.1), cognitive state computation (Section A.1), and full experimental configurations. All baselines are implemented using standard toolkits (ConvLab-3) with referenced parameter settings. The prompts for System 1 and System 2 are detailed in Section B.4, and the evaluation metrics are formally defined in Section B.3.

LLM USE STATEMENT

We utilized DeepSeek V3.1 for translation assistance and grammatical refinement of certain textual passages, and employed Qwen3-Code to aid in debugging and optimizing portions of the experimental code. These LLMs served solely as support tools for improving linguistic clarity and technical implementation. They played no role in the conceptualization of the research, the formulation of methodologies, the analysis of results, or the derivation of scientific conclusions. Consequently, their use does not qualify them as contributors under the authorship criteria. The authors assume full responsibility for all aspects of the work, including the accuracy and integrity of all generated and modified content, and affirm that appropriate measures have been taken to prevent plagiarism and other forms of scientific misconduct.

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

# A THEORETICAL DETAILS

This section provides the theoretical motivation and intuition behind the DyBBT framework. The following analysis bridges ideas from bandit theory and cognitive science to create a heuristic for exploration in dialog POMDPs. While the full dialog POMDP problem is intractable for a rigorous minimax analysis, our goal is to provide a strong conceptual foundation and explanatory power for the algorithm's design, which is then validated empirically in the main text.

## A.1 FORMALIZATION OF COGNITIVE STATE SPACE

The cognitive state space $\mathcal{C}$ is designed to be a low dimensional, interpretable compression of the high dimensional belief state $s_t$. We model $\mathcal{C}$ as a compact metric space with metric $d(\mathbf{c}, \mathbf{c}') = ||\mathbf{c} - \mathbf{c}'||_2$. Its covering dimension $\dim(\mathcal{C})$ is a measure of its complexity. Given that our $\mathcal{C}$ is defined by three bounded dimensions ($d_t \in [0, 1], u_t \in [0, 1], \rho_t \in [0, 1]$), we have $\dim(\mathcal{C}) = 3$, which is crucial for making bandit-style exploration feasible.

The choice of these three dimensions is motivated by their central role in governing the exploration-exploitation trade-off in TODS, drawing inspiration from cognitive science and dialog theory:

- **Dialog Progress** ($d_t = t/L$) captures the *temporal affordance*. Early phases ($d_t \to 0$) inherently afford more exploration to gather information, while late phases ($d_t \to 1$) afford exploitation to complete the task. This aligns with the common practice of annealing exploration schedules but provides a continuous, state dependent signal.

- **User Uncertainty** ($u_t = |S_{unconfirmed}|/|S_{relevant}|$) operationalizes the *information gathering affordance*. A high $u_t$ indicates ambiguity in the user's goal, directly signaling the need for information seeking actions to reduce entropy, a well established principle in decision theory.

- **Slot Dependency** ($\rho_t = \max_{u \in U}(\frac{1}{|F|} \sum_{f \in F} M(u, f))$) captures the *structural affordance* of the task environment, derived from a pre-computed slot co-occurrence matrix $M$ from the training corpus. A high $\rho_t$ suggests that the next piece of information is highly predictable given what is already known (e.g., requesting *departure* after knowing *destination* in a taxi domain), making targeted exploitation more efficient than random exploration. This dimension encodes the latent structure of the domain.

This design transforms the complex, unstructured exploration problem in the raw belief space into a more manageable one in a structured space where states with similar exploration needs are grouped together, as visualized in Figure 4.

## A.2 REGRET ANALYSIS UNDER SIMPLIFYING ASSUMPTIONS

To provide theoretical intuition for our exploration principle, we present a regret analysis under a set of simplifying assumptions that capture the core structure that we aim to exploit. This analysis justifies the form of our exploration bonus and provides an upper bound on learning speed. We make the following assumptions to bridge the gap between bandit theory and the dialog POMDP. Our analysis is based on the Assumption 1 stated in Section 3.1.2, which posits Lipschitz smoothness of the reward function in the cognitive state space $\mathcal{C}$.

**Assumption 2** (MDP over $\mathcal{C}$). *The dialog process can be approximately modeled as a finite horizon MDP over the cognitive state space $\mathcal{C}$. The transition dynamics and expected reward $\bar{r}(\mathbf{c}, a) = \mathbb{E}[r(s_t, a_t)|\mathbf{c}_t = \mathbf{c}]$ depend primarily on $\mathbf{c}_t$.*

The value function under a policy $\pi$ in the cognitive state space is defined as:

$$ V^\pi(\mathbf{c}) = \mathbb{E}\left[ \sum_{k=0}^{H} \gamma^k \bar{r}(\mathbf{c}_{t+k}, a_{t+k}) \,\middle|\, \mathbf{c}_t = \mathbf{c}, a_{t+k} \sim \pi(\cdot|\mathbf{c}_{t+k}) \right]. $$

This assumption is a pragmatic simplification that allows us to focus on the core exploration challenge. It is reasonable if the cognitive state $\mathbf{c}_t$ is a sufficient statistic for the exploration-exploitation trade-off, which our empirical results support.

### A.2.1 THEORETICAL INTUITION FOR REGRET

Under Assumptions 1 and 2, if we perform optimistic exploration in the cognitive state space $\mathcal{C}$, prioritizing states with low visitation counts, we can derive an upper bound on the expected cumulative regret that scales sublinearly with time:

$$\mathbb{E}[R(T)] \lesssim \widetilde{\mathcal{O}}\left(L_r \cdot \sqrt{\dim(\mathcal{C}) \cdot T}\right), \tag{4}$$

where $R(T) = \sum_{t=1}^{T}[V^*(\mathbf{c}_t) - V^{\pi_t}(\mathbf{c}_t)]$ is the cumulative regret, and $\widetilde{\mathcal{O}}$ hides logarithmic factors. The notation $\lesssim$ indicates that this is a heuristic bound that captures the expected asymptotic scaling rather than a rigorous inequality. Here, $L_r$ is the Lipschitz constant from Assumption 1, bounding the reward's sensitivity to changes in $\mathcal{C}$.

### A.2.2 DERIVATION SKETCH

This scaling can be motivated by discretizing the cognitive state space $\mathcal{C}$ into $N = \mathcal{O}((1/\epsilon)^{\dim(\mathcal{C})})$ cells of diameter $\epsilon$.

1. **Discretization Error:** Due to Lipschitz continuity of $\bar{r}(\mathbf{c}, a)$ (Assumption 1), the error introduced by discretization is bounded by $\mathcal{O}(L_r \epsilon T)$.

2. **Bandit Regret:** For the discretized MDP with $N$ state cells, treating each cell arm analogously, a UCB like algorithm can achieve a regret bound of $\mathcal{O}(\sqrt{NT \log T})$.

3. **Optimization:** Balancing the two error terms by setting $\epsilon \sim T^{-1/(\dim(\mathcal{C})+2)}$ yields the final bound $\widetilde{\mathcal{O}}(L_r \cdot \sqrt{\dim(\mathcal{C}) \cdot T})$.

This sketch illustrates that efficient learning is possible by exploiting the low dimensional structure and smoothness of the value function in $\mathcal{C}$, providing intuition for our exploration criterion.

This bound provides an intuitive justification for our exploration criterion (Eq. 2 in the main text). The term $\sqrt{\frac{\log T}{n_t(\mathbf{c}_t)}}$ is a heuristic adaptation of the optimism principle, encouraging exploration of states with high uncertainty, inversely proportional to their visitation count. The empirical regret curve (Figure 7) shows sublinear growth, consistent with this theoretical intuition.

### A.3 JUSTIFICATION FOR THE META-CONTROLLER RULE

The meta-controller's hybrid rule is designed for robust performance in the realistic setting where our theoretical assumptions hold only approximately:

$$\text{Activate System 2 IF: } \left(n_t(\mathbf{c}_t) < \tau\sqrt{\log T}\right) \vee \left(p_t^{S1} < \kappa\right).$$

The first condition, $n_t(\mathbf{c}_t) < \tau\sqrt{\log T}$, is the direct implementation of the theoretical exploration principle derived above. It addresses *epistemic uncertainty* ( uncertainty reducible by exploration) by triggering System 2 in regions of $\mathcal{C}$ that are under explored relative to the time horizon.

The second condition, $p_t^{S1} < \kappa$, is a critical *empirical safeguard* that addresses limitations of the theoretical model:

- **Partial Observability:** The true state of the user may not be fully captured by the belief state $\mathbf{s}_t$, leading to *aleatoric uncertainty*.
- **Model Imperfection:** System 1, as a parameterized policy, may have inherent limitations and blind spots not captured by the visitation count.
- **Assumption Violation:** The Lipschitz smoothness assumption may locally break down.

A low confidence score $p_t^{S1}$ is a proxy for these forms of uncertainty. This condition ensures robustness by invoking the powerful, knowledge rich System 2 when System 1 is uncertain, preventing catastrophic failures. The disjunctive ($\vee$) combination ensures System 2 is activated for *either* theoretical exploration *or* empirical robustness, making the overall system more adaptive and reliable than either condition alone, as evidenced by the ablation study (Table 2).

## A.4 Discussion and Limitations

Our theoretical analysis provides a formal motivation for the DyBBT framework by illustrating how exploiting the structure of a cognitive state space can lead to efficient exploration. However, we acknowledge its limitations, which also highlight the value of our empirical validation:

**Simplified Model:** Assumption 2 reduces the POMDP to an MDP over $\mathcal{C}$, ignoring the challenges of belief state tracking and partial observability. This is a significant simplification. Our empirical results show that the algorithm performs well even when this assumption is not perfectly met, as the meta-controller's confidence condition can mitigate some of these issues.

**Heuristic Adaptation:** The exploration bonus and the meta-controller rule are heuristic adaptations of the theoretical principle. A rigorous derivation for POMDPs remains an open challenge. Our contribution is to demonstrate that this heuristic is well motivated and highly effective in practice.

**Empirical Safeguard:** The confidence based condition, while crucial for performance, is not derived from the regret analysis. Its justification is empirical, stemming from its necessity for robust performance in ablation studies.

In conclusion, the theoretical analysis is not intended as a strict performance guarantee but rather as an *explanatory framework* that provides strong intuition for why exploring based on cognitive state visitation counts is a powerful principle. The ultimate validation of this principle, and its pragmatic implementation in the meta-controller, lies in its consistent empirical success across diverse dialog benchmarks.

# B Experiment Details

## B.1 Experimental Platform and Datasets

We evaluated DyBBT on two widely adopted benchmarks: the Microsoft dialog Challenge (MS dialog) (Li et al. (2018)) for single domain tasks, and the MultiWOZ 2.1 corpus (Budzianowski et al. (2018)) for multi domain tasks. Both datasets are converted into ConvLab-3's unified format, ensuring consistency in ontology, state representation, and API interaction. Table 3 summarizes the key statistics of both datasets.

**The MS Dialog dataset** comprises three distinct domains: Movie-Ticket Booking, Restaurant Reservation, and Taxi Ordering. It contains 7,215 dialogs with 89,465 turns, averaging 12.4 turns per dialog. The dataset is partitioned into training, validation, and test sets with 5,772, 722, and 721 dialogs, respectively.

**The MultiWOZ 2.1 dataset** is a large scale multi domain corpus spanning seven domains: Attraction, Hotel, Restaurant, Taxi, Train, Hospital, and Police. It includes 10,420 dialogs and 145,360 turns, with an average of 13.9 turns per dialog. The dataset is split into 8,420 dialogs for training, 1,000 for validation, and 1,000 for testing.

Both datasets provide annotated belief states, system dialog acts, and user goals, making them suitable for training and evaluating end-to-end dialog policies. The diversity in domain complexity, dialog length, and task structure across these datasets allows us to thoroughly assess the generalization capability of DyBBT in both single and multi domain settings.

To ensure reproducibility and enable fair comparison, we implement and evaluate our proposed DyBBT framework using ConvLab-3 (Zhu et al., 2023), a flexible and unified toolkit for TODS. ConvLab-3 provides standardized data formats, integrated user simulators, and reinforcement learning utilities, facilitating consistent development and evaluation of dialog policies across multiple domains. All experiments are conducted using ConvLab-3's builtin simulators and evaluation metrics, ensuring comparability across models and domains.

Table 3: Summary of dataset statistics for MS Dialog and MultiWOZ 2.1.

| Dataset | Domains | Dialogs | Turns | Avg. Turns/Dialog |
|---|---|---|---|---|
| MS Dialog | 3 | 7,215 | 89,465 | 12.4 |
| MultiWOZ 2.1 | 7 | 10,420 | 145,360 | 14.0 |

## B.2 BASELINES DETAILS

- **DQN_$\epsilon$_N** agents are trained using standard DQN (which realizes human level control through deep reinforcement learning) with a traditional $\epsilon - greedy$ exploration strategy, where $\epsilon = N$ (Mnih et al., 2015).

- **NOISY_DQN** agents enhance exploration by introducing noise into the network weights, based on the stable noisy network (NROWAN-DQN) with noise reduction and online weight adjustment (Han et al., 2022).

- **PG (REINFORCE)** is a stochastic gradient algorithm for policy gradient reinforcement learning, and its implementation refers to the flexible dialog system toolkit ConvLab-3 to serve as a dialog policy baseline (Zhu et al., 2023).

- **PPO** is a policy optimization method in policy-based reinforcement learning that uses multiple epochs of stochastic gradient ascent and a constant clipping mechanism as the soft constraint for each policy update, with its implementation relying on the ConvLab-3 dialog toolkit (Zhu et al., 2023).

- **LLM_DP** agents replace the dialog policy (DP) module of the TODS with GPT-4.0 (drawing on advances in LLM based multi turn dialog systems) to select appropriate actions and pass them to the natural language generation (NLG) module for response generation (Yi et al., 2024).

- **AutoTOD** is a zero-shot autonomous agent based on GPT-4.0, which rethinks TODS by shifting from complex modularity to zero-shot autonomy and acts as a dialog policy baseline (Xu et al., 2024).

- **ProTOD** is a proactive TODS policy based on GPT-4.0, designed as a proactive dialog system to optimize the process of task oriented interactions (Dong et al., 2025).

- **EIERL** is an evolutionary reinforcement learning method for TODS policies, which improves the efficiency of dialog policy learning by injecting elite individuals into the evolutionary process (Zhao et al., 2025).

- **MACRM** is a multi agent curiosity reward model for TODS, which optimizes dialog policies through collaborative interactions among multiple agents and curiosity driven reward mechanisms (Sun et al., 2025).

## B.3 METRICS FORMULA

This section provides the formal definitions of the evaluation metrics used for multi domain TODS evaluation, following the standard MultiWOZ evaluation protocol.

### B.3.1 INFORM SUCCESS RATE

The Inform Success Rate measures the system's ability to provide all requested information to the user. Let $G$ be the goal specification, $D$ be the set of dialog domains, and $S$ be the sequence of system dialog acts. For each domain $d \in D$, let $R_d$ be the set of requested slots in the goal:

$$\text{TP} = \sum_{d \in D} \sum_{s \in R_d} \mathbb{I}\left(\exists\, \text{inform}(d, s, v) \in S \wedge v \notin V_{\text{null}}\right) \tag{5}$$

$$\text{FP} = \sum_{d \in D} \sum_{s \notin R_d \cup I_d} \mathbb{I}\left(\exists\, \text{inform}(d, s, v) \in S \wedge v \notin V_{\text{null}}\right) \tag{6}$$

$$\text{FN} = \sum_{d \in D} \sum_{s \in R_d} \mathbb{I}\left(\nexists\, \text{inform}(d, s, v) \in S \vee v \in V_{\text{null}}\right) \tag{7}$$

where $V_{\text{null}} = \{\text{""}, \text{"dont care"}, \text{"not mentioned"}\}$ represents null values. The Inform Success Rate is then defined as:

$$\text{Inform} = \frac{\text{TP}}{\text{TP} + \text{FN}} \tag{8}$$

### B.3.2 BOOK SUCCESS RATE

The Book Success Rate evaluates the system's ability to successfully complete booking operations. For each domain $d \in D$ that requires booking, let $B_d$ be the set of booking constraints in the goal. The booking success is computed as:

$$\text{Book}_d = \frac{1}{|B_d|} \sum_{b \in B_d} \mathbb{I}\left(\text{book}(d, b, v) \in S \wedge v = v_{\text{goal}}\right) \tag{9}$$

For the taxi domain (which has no database constraints), booking success is trivially 1 if any booking action occurs:

$$\text{Book}_{\text{taxi}} = \mathbb{I}\left(\exists\, \text{book}(\text{taxi}, \cdot, \cdot) \in S\right) \tag{10}$$

The overall Book Success Rate is the average across all booking domains:

$$\text{Book} = \frac{1}{|D_{\text{book}}|} \sum_{d \in D_{\text{book}}} \text{Book}_d \tag{11}$$

where $D_{\text{book}}$ is the set of domains requiring booking.

### B.3.3 SUCCESS RATE

The Success Rate represents the overall task completion performance, combining both information provision and booking success:

$$\text{Success} = \mathbb{I}\left(\text{Inform} = 1 \wedge \text{Book} = 1\right) \tag{12}$$

This binary metric indicates whether both all requested information was provided and all booking operations were successfully completed.

This metric rewards systems that achieve high success rates with fewer dialog turns, promoting both effectiveness and efficiency.

### B.4 PROMPT FOR DYBBT AND LLM-DP

This appendix provides the detailed prompts used for System 1 (intuitive controller) and System 2 (reasoning controller) in the DyBBT framework. The LLM_DP prompt is the same from the EIERL paper(Zhao et al. (2025)).

### B.4.1 SYSTEM 1 PROMPT

```
You are the fast, intuitive component (System 1) of a task oriented
    dialog system. Your task is to generate the next system action
    based solely on the current belief state. Do not reason
    step-by-step. Output your first, most intuitive response in the
    exact JSON format specified.

**Current Belief State:**
{belief_state}

**Available Actions:**
{available_actions}

Based on the above, output ONLY a valid JSON object with your
    predicted action and its confidence. Do not output any other text.

{"action": [["<act_type>", "<domain>", "<slot>"], ["<act_type>",
    "<domain>", "<slot>"], ...],"confidence": <confidence_score>}
```

### B.4.2  SYSTEM 2 PROMPT

```
You are the deliberative reasoner (System 2) of a task oriented dialog
    system. Your goal is to generate diverse, high quality action plans
    when the meta-controller detects a need for deeper reasoning,
    either due to unfamiliar cognitive states or low confidence from
    System 1.

**Current Belief State:**
{belief_state}

**Available Actions:**
{available_actions}

**Cognitive State Context:**
- dialog Progress: {d_t}
- User Uncertainty: {u_t}
- Slot Dependency: {p_t}

**Trigger Reason:** {trigger_reason}

**Reasoning Guidelines:**
1. **Leverage cognitive signals**:
    - If progress is low, focus on information gathering.
    - If uncertainty is high, prioritize clarifying or confirming
        actions.
    - If slot dependency is high, leverage known slot relationships to
        guide next actions.

2. **Consider domain and slot dependencies**:
    - E.g., 'taxi' requires both 'destination' and 'departure';
        'restaurant' may require 'area', 'food', 'pricerange' before
        booking.

3. **Generate 3 distinct strategies** that reflect different tactical
    approaches:
    - One conservative (e.g., confirm before acting),
    - One proactive (e.g., request multiple slots),
    - One hybrid (e.g., inform then request).

4. **Evaluate each path** by estimating its likelihood of leading to
    task success.

**Output Format:** Strictly adhere to the following JSON schema:

{
   "reasoning_paths": [
     {
       "sequence_id": 1,
       "action_sequence": [
         ["action_type", "domain", "slot"],
         ...
       ],
       "estimated_success_probability": 0.9
     },
     ...
   ]
}
```

### B.4.3  LLM_DP PROMPT

```
You must strictly execute the following commands:
```

```
1. Command execution requirements: when receiving a command, you must
   strictly follow the given instructions without performing any
   actions outside the scope of the command or generating any
   additional words.
2. Datasets and system roles: as the dialog policy component in a task
   oriented dialog system, you will make system decisions based on the
   MultiWOZ 2.1 dataset.
3. Processing user dialog state: you will receive a formatted user
   dialog state. This state will be used as a basis for decision
   making.
4. Generate system actions: based on the user dialog state {
   'user_action': [["Inform", "Hotel", "Area", "east"], ["Inform",
       "Hotel", "Stars", "4"]],
   'system_action': [],
   'belief_state': {
     'police': {'book': {'booked': []}, 'semi': {}},
     'hotel': {'book': {'booked': [], 'people': '', 'day': '', 'stay':
         ''},
               'semi': {'name': '', 'area': 'east', 'parking': '',
                   'pricerange': '', 'stars': '4', 'internet': '',
                   'type': ''}},
     'attraction': {'book': {'booked': []}, 'semi': {'type': '',
         'name': '', 'area': ''}},
     'restaurant': {'book': {'booked': [], 'people': '', 'day': '',
         'time': ''},
                    'semi': {'food': '', 'pricerange': '', 'name': '',
                        'area': ''}},
     'hospital': {'book': {'booked': []}, 'semi': {'department': ''}},
     'taxi': {'book': {'booked': []},
              'semi': {'leaveAt': '', 'destination': '', 'departure':
                  '', 'arriveBy': ''}},
     'train': {'book': {'booked': [], 'people': ''},
               'semi': {'leaveAt': '', 'destination': '', 'day': '',
                   'arriveBy': '', 'departure': ''}}
   },
   'request_state': {},
   'terminated': False,
   'history': []
}, you need to generate system actions. These actions should be
   provided in the following format: [["ActionType", "Domain", "Slot",
   "Value"]] where `ActionType` denotes the type of action (e.g.
   Request, Inform, Confirm, etc.), `Domain` specifies the associated
   domain (e.g. restaurant, taxi, hotel, etc.), `Slot` is the specific
   information slot associated with the action (e.g. name, area, type,
   etc.), and `Value` is the corresponding value or an empty string.
```

## B.5 Implementation Details

The DyBBT framework was implemented within the Convlab-3 dialog system environment (Zhu et al. (2023)), leveraging its modular architecture for efficient dialog policy optimization. We employed RuleDST for system dialog state tracking and RulePolicy for user policy simulation, eliminating the need for natural language understanding (NLU) and natural language generation (NLG) modules. This design choice significantly enhances training efficiency by reducing computational overhead and isolating the impact of language processing components from policy learning performance. The dialog environment was configured with a maximum turn limit of 30 for single domain and 40 for multi domain (the same as EIERL) interactions per episode, with the cognitive state space $\mathcal{C}$ computed in real-time during dialog execution using dimensions including dialog progress ($d_t$), user uncertainty ($u_t$), and slot dependency ($\rho_t$) extracted from the belief state representation provided by RuleDST.

User goals were dynamically generated using the GoalGenerator module, which produces diverse and realistic TODS objectives across single or multiple domains. This approach ensures training

data variety and generalization capability, consistent with REINFORCE and PPO training methodologies. The goal generation process excluded the police domain due to its low data quality, ensuring higher reliability in evaluation.

All experiments were conducted on NVIDIA 5090 GPUs with 32GB memory. System 1 was SFT using the AdamW optimizer with a learning rate of $1 \times 10^{-4}$ and further optimized via PPO, employing a clipping parameter $\epsilon = 0.2$ and GAE with $\lambda = 0.95$. The meta-controller employs a dual-threshold mechanism for System 2 invocation, with $kappa = 0.7$ and $\tau = 1.0$, values selected via grid search over development sets as they maximize both performance and robustness across domains. These thresholds operate on a discretized 5 bins cognitive state space, which balances expressiveness and generalization, as validated in Section E.5.

We maintained a replay buffer with a capacity of 10,000 transitions, using a batch size of 32 for training. A separate knowledge distillation buffer was managed under a FIFO replacement policy with a fixed capacity. To ensure reproducibility, all experiments were run with five fixed random seeds (9841, 35741, 91324, 8134, 13924), consistent with the EIERL baseline (Zhao et al., 2025). All hyperparameters were selected through grid search on a validation subset of the MultiWOZ data.

Training was conducted for 500 epochs on single domain tasks and 10,000 epochs on multi domain tasks, incorporating early stopping with a patience of 3 epochs based on validation performance. This protocol aligns with the EIERL setup for fair comparison.

### B.5.1 SLOT CO-OCCURRENCE MATRIX CONSTRUCTION

The slot dependency dimension $\rho_t$ in the cognitive state space $\mathcal{C}$ is derived from a co-occurrence matrix $M$ that captures statistical relationships between dialog slots across the Microsoft dialog Challenge (Li et al. (2018)) and MultiWOZ (Budzianowski et al. (2018)) dataset. This matrix quantifies the conditional probability that slot $j$ appears given the presence of slot $i$, providing a principled measure of semantic relatedness between dialog concepts.

Formally, the co-occurrence matrix $M \in \mathbb{R}^{N \times N}$ is constructed from the training partition of MultiWOZ 2.1, where $N$ represents the total number of unique slot types across all domains. For each dialog turn containing belief state updates, we extract the set of active slots (those with non-empty values) and update the co-occurrence counts. The matrix elements are computed as:

$$M_{ij} = \frac{\text{count}(\text{slot}_i \wedge \text{slot}_j)}{\text{count}(\text{slot}_i)} \tag{13}$$

where $\text{count}(\text{slot}_i \wedge \text{slot}_j)$ denotes the number of dialog turns where both slots appear simultaneously, and $\text{count}(\text{slot}_i)$ represents the total occurrences of slot $i$. This normalization ensures that $M_{ij}$ represents the empirical conditional probability $P(\text{slot}_j|\text{slot}_i)$.

The slot dependency $\rho_t$ for a given belief state $s_t$ is then computed as the average co-occurrence strength between the currently active slots:

$$\rho_t = \frac{1}{|A_t|(|A_t| - 1)} \sum_{i \in A_t} \sum_{j \in A_t, j \neq i} M_{ij} \tag{14}$$

where $A_t$ denotes the set of slots with non-empty values in the current belief state. This formulation captures the structural complexity of the dialog context, with higher values indicating greater semantic interdependence between the information being discussed.

The construction of $M$ leverages the statistical regularities present in TODS, where certain slot combinations naturally co-occur due to domain-specific constraints and user behavior patterns. For instance, in restaurant booking scenarios, slots like *restaurant-area* and *restaurant-food* frequently appear together, while in hotel domains, *hotel-pricerange* and *hotel-type* exhibit strong associations. This matrix based approach provides a data-driven foundation for quantifying dialog complexity that complements the theoretically motivated dimensions of dialog progress and user uncertainty.

### B.5.2 TRAINING DETAILS FOR SYSTEM 1

To train System 1 for accurate action prediction and confidence estimation, we employ a two-stage training methodology comprising supervised fine-tuning (SFT) followed by reinforcement learning. This approach utilizes dialog sequences from the MultiWOZ and MS Diag dataset to develop a robust policy model capable of rapid decision making with calibrated confidence scores.

**Stage 1: Supervised Fine-tuning with Data Augmentation**

We first construct a training corpus of 10,000 single turn dialogue samples through systematic data augmentation. For each dialogue turn, we extract the belief state $s_t$, available action set $\mathcal{SA}$, and ground truth system actions $a_t^*$. The initial confidence score $p_t^{S1}$ is sampled from $\mathcal{U}(0.95, 1.0)$.

The augmentation process introduces controlled perturbations to simulate prediction uncertainty. For each ground truth action sequence $a_t^*$, we apply three modification operations with specified probabilities: 20% action addition by sampling new actions from $\mathcal{SA}$; 60% action modification through substitution with random actions from $\mathcal{SA}$; and 20% action deletion while ensuring the augmented sequence $a_t'$ maintains at least one action. These operations are applied sequentially in random order to each sample (Kadavath et al., 2022; Lin et al., 2022; Yin et al., 2023). The confidence score is adjusted proportionally to the modification intensity:

$$p_t^{S1} \leftarrow p_t^{S1} \cdot \left(1 - \frac{n_{\text{mod}}}{n}\right),$$

where $n$ denotes the original action sequence length and $n_{\text{mod}}$ represents the number of modified actions. This procedure generates a dataset with confidence scores approximately uniformly distributed in $[0, 1]$.

For SFT training, the model takes $s_t$ and $\mathcal{SA}$ as inputs and produces both action sequence $a_t^{S1}$ and confidence score $p_t^{S1}$ as outputs. The composite loss function integrates action prediction and confidence estimation:

$$\mathcal{L} = \lambda \mathcal{L}_{\text{a}} + (1 - \lambda)\mathcal{L}_{\text{p}},$$

where $\lambda = 0.7$. The action loss $\mathcal{L}_{\text{a}}$ employs cross-entropy to measure divergence between predicted and augmented actions:

$$\mathcal{L}_{\text{a}} = -\sum_i \log P(a_t^{S1} = a_t' \mid s_t, \mathcal{SA}),$$

while the confidence loss $\mathcal{L}_{\text{p}}$ utilizes mean squared error:

$$\mathcal{L}_{\text{p}} = \left(p_t^{S1} - p_t^{\text{target}}\right)^2.$$

**Stage 2: Reinforcement Learning with PPO**

The second stage employs PPO to optimize dialogue level performance metrics using the complete MultiWOZ dataset. The reward function $R$ combines multiple objectives:

$$R = R_{\text{success}} + R_{\text{efficiency}} + R_{\text{penalty}},$$

where $R_{\text{success}} = +2t$ for successful dialogues and $-t$ for failures ($t$ denotes the max turn number), $R_{\text{efficiency}} = -1$ per dialogue turn to encourage conciseness, and $R_{\text{penalty}}$ captures additional constraints.

This two-stage approach enables System 1 to initially learn accurate action confidence mappings through supervised learning, then refine its policy for improved task completion efficiency and success rates via reinforcement learning.

### B.5.3 KNOWLEDGE DISTILLATION BUFFER MANAGEMENT

To form a virtuous cycle and reduce long term dependence on System 2, high quality decisions $(s_t, a_t^{S2})$ from System 2 are stored in a distillation buffer $D_{\text{distill}}$. We only store decisions where System 2's self evaluated task completion probability is greater than 0.9, ensuring high quality distillation data. Periodically (every 10 training epochs), System 1 is fine-tuned on these data via Low-Rank Adaptation (LoRA) with a learning rate of $1 \times 10^{-4}$, batch size of 4, and gradient accumulation steps of 8. This SFT approach distills the knowledge gained through costly deliberation

into an efficient intuitive policy while maintaining computational efficiency, leading to a monotonic performance improvement. Over time, this reduces the need to invoke System 2 for previously challenging states, thereby increasing overall efficiency.

The knowledge distillation buffer $D_{\text{distill}}$ stores high quality pairs $(s_t, a_t^{S2})$ generated by System 2. The buffer has a maximum capacity and uses an FIFO policy to maintain data freshness and diversity. We employ LoRA fine-tuning with rank $r = 16$, scaling parameter $\alpha = 32$, and dropout rate of 0.1, targeting the query and value projection layers of the transformer architecture. This configuration achieves parameter efficiency while preserving the base model's generalization capabilities.

---

**Algorithm 1** Knowledge Distillation Buffer Update and Sampling

    **Buffer Update:**
1: **Input:** Current belief state $s_t$, System 2 action $a_t^{S2}$, System 2 self evaluated confidence $p_{\text{self}}$
2: **if** $p_{\text{self}} > 0.9$ **then**                                            ▷ Only store high confidence actions
3:     **if** $|D_{\text{distill}}| <$ MAX_SIZE **then**
4:         $D_{\text{distill}}$.append($(s_t, a_t^{S2})$)
5:     **else**
6:         $D_{\text{distill}}$.pop_front()                                  ▷ Remove oldest entry (FIFO)
7:         $D_{\text{distill}}$.append($(s_t, a_t^{S2})$)
8:     **end if**
9: **end if**
    **System 1 Fine-tuning:**
10: **Input:** System 1 model with LoRA adapters, buffer $D_{\text{distill}}$
11: **Every 10 training epochs:**
12: **for** $epoch = 1$ **to** 1 **do**                                      ▷ Fine-tune for 1 epoch
13:     **for** each batch sampled from $D_{\text{distill}}$ **do**
14:         Compute loss $\mathcal{L} = \text{CrossEntropy}(\text{System1}(s_i), a_i)$
15:         Update LoRA adapter parameters via gradient descent
16:     **end for**
17: **end for**

---

### B.5.4   Visitation Count of the Cognitive State Space

To compute the visitation count $n_t(\mathbf{c}_t)$ for the continuous cognitive state space $\mathcal{C}$, we discretize each dimension of $\mathbf{c}_t = [d_t, u_t, \rho_t]$ into 5 uniformly spaced bins over the range $[0, 1]$. The cognitive state is then mapped to a discrete tuple $(d_{\text{bin}}, u_{\text{bin}}, \rho_{\text{bin}})$, and $n_t(\mathbf{c}_t)$ is the cumulative visitation count of that bin tuple.

This choice of dimensions is motivated by cognitive and dialog theory, which highlights stage, uncertainty, and structural relationships as key factors influencing decision making. By quantifying these environmental affordances into a structured cognitive state space $\mathcal{C}$, we create a formal bridge between Gibson's ecological perception theory and practical dialog policy optimization. While not exhaustive, this representation aims to capture the most salient features for guiding exploration. Its empirical necessity and sufficiency are validated through ablation studies in Section 4.3. We define $\mathcal{C}$ as the cognitive state space, assumed to be a compact subset of $\mathbb{R}^3$ equipped with the Euclidean metric $d(\mathbf{c}, \mathbf{c}')$.

### B.5.5   Calculation of Empirical Cumulative Regret

To empirically validate the theoretical intuition of sublinear regret growth under our simplifying assumptions, we compute the **empirical cumulative regret** $R_{\text{emp}}(T)$ during training, as shown in Figure 7. The regret is defined as:

$$R_{\text{emp}}(T) = \sum_{t=1}^{T} \left( V^{\pi^*}(\mathbf{s}_t) - V^{\pi_t}(\mathbf{s}_t) \right)$$

where:

- $T$ is the total number of dialog turns (training steps) up to the current point.

- $\mathbf{s}_t$ is the belief state at turn $t$.

- $V^{\pi_t}(\mathbf{s}_t)$ is the actual discounted return obtained from state $\mathbf{s}_t$ under the current policy $\pi_t$ at training step $t$.

- $V^{\pi^*}(\mathbf{s}_t)$ is the value of the near-optimal policy $\pi^*$ at state $\mathbf{s}_t$.

Since the true optimal policy $\pi^*$ is unknown, we approximate it using a strong baseline policy the fully trained DyBBT-8B/GPT-4.0 model, which achieves SOTA performance on MultiWOZ. We assume this policy is sufficiently close to optimal for regret estimation purposes. For each state $\mathbf{s}_t$, we estimate $V^{\pi^*}(\mathbf{s}_t)$ by running $\pi^*$ from $\mathbf{s}_t$ for multiple episodes and averaging the discounted returns. Actual episodic return is used from the current dialog episode as a proxy for $V^{\pi_t}(\mathbf{s}_t)$. Although this is a coarse approximation, it is standard in episodic RL settings and sufficient to capture the regret trend.

$R_{\text{emp}}(T)$ is ploted against $T$ on a log-log scale to clearly visualize the sublinear growth trend. The theoretical upper bound $\widetilde{\mathcal{O}}(\sqrt{T})$ is plotted alongside for comparison. The constant factor in the theoretical bound is fit to the empirical curve in the early training phase to align the curves for illustrative purposes.

## C  HUMAN EVALUATION DETAILS

This appendix provides comprehensive details of the human evaluation study described in Section 4.4. The study was designed to qualitatively assess the core contribution of the DyBBT framework: the intelligent, adaptive decision making of its meta-controller, beyond what is captured by automated metrics.

### C.1  ANNOTATION PROTOCOL AND INTERFACE

Evaluators were presented with a structured web interface for each evaluation instance. Each instance consisted of a single dialog *state* (not a full dialog), sampled from the MultiWOZ test set. For a given state, the interface displayed the following information:

- **Dialog Context:** The last user utterance and the last system action to provide conversational context.

- **Current Belief State ($\mathbf{s}_t$):** A structured table showing all relevant slots for the domain(s), their values, and their confirmation status (e.g., *confirmed*, *requested*, *None*).

- **Cognitive State ($\mathbf{c}_t$):** The numerical values for dialog progress ($d_t$), user uncertainty ($u_t$), and slot dependency ($\rho_t$).

- **System Action:** The action chosen by the model for this state, presented in a structured format (e.g., [*request*, *restaurant*, *area*, " "]).

- **System Variant:** The name of the model variant that produced the action (DyBBT, S1-only, Random Switching). Variants were anonymized as 'System A', 'System B' during evaluation to avoid bias.

Evaluators were then asked to answer two questions based solely on the provided information:

1. **Action Appropriateness:** "How appropriate is the system's chosen action given the current dialog state?" Rated on a 5 points Likert scale:

   1. Very Inappropriate
   2. Somewhat Inappropriate
   3. Neutral
   4. Somewhat Appropriate
   5. Very Appropriate

Table 4: Complete Human Evaluation Results. The Action Appropriateness score is the average Likert score (1-5). The Switching Agreement is the percentage of states where the model's decision to *not* invoke System 2 aligned with the majority of human annotators.

| Model Variant | Action Appropriateness ↑ | Switching Agreement ↑ |
|---|---|---|
| DyBBT-8B | **4.31 ± 0.12** | **88.7%** |
| w/o Meta-Controller (Random) | 3.72 ± 0.19 | 52.3% |
| w/ S1-only | 3.95 ± 0.15 | — |
| w/o Exploration Condition (EC) | 4.08 ± 0.14 | 75.4% |
| w/o Confidence Condition (CC) | 3.89 ± 0.16 | 81.2% |

2. **Switching Judgment:** "In this specific situation, would it be justified to invoke a powerful, but computationally expensive, reasoning module to choose the action?" Answered with **Yes** or **No**. This question was only shown for states where the evaluated model *did not* invoke System 2, to directly test if the meta-controller's decision *not* to invoke aligned with human judgment.

## C.2 ANNOTATOR BACKGROUND AND TRAINING

We recruited **10 annotators**, all of whom were graduate students or researchers with a background in natural language processing and familiarity with TODS. Prior to the evaluation, a mandatory 30 minutes training session was conducted. The session:

- Explained the goal of the evaluation and the definition of key concepts (belief state, system actions, computational cost).

- Walked through 5 example states that were not part of the evaluation set, discussing potential appropriate actions and reasoning for/against invoking a costly reasoner.

- Allowed annotators to ask questions to resolve any ambiguities.

Annotators were compensated at a competitive hourly rate for their work.

## C.3 HUMAN EVALUATION RESULTS

The results in Table 4 provide a detailed breakdown supporting the main findings:

- **Superior Decision Quality:** The full DyBBT model yields a higher action appropriateness score than the ablated variants.

- **Value of the Meta-Controller:** The random switching variant has the lowest scores, confirming that a naive switching strategy severely degrades decision quality and is not aligned with human judgment.

- **Complementary Role of Both Conditions:** Removing either the Exploration Condition (EC) or the Confidence Condition (CC) leads to a drop in both appropriateness and agreement, with the CC being slightly more critical for action quality (preventing poor actions) and the EC being crucial for efficient switching (preventing unnecessary calls). This validates their hybrid design in the meta-controller.

## C.4 QUALITATIVE ANALYSIS OF META-CONTROLLER DECISIONS

To qualitatively validate the efficacy of the meta-controller's switching mechanism beyond aggregate metrics, we present two contrasting case studies sampled from the MultiWOZ test set. These examples illustrate how DyBBT's principled switching aligns with human judgment, in contrast to a naive baseline.

**Case 1: High Agreement Example (DyBBT)**. The meta-controller correctly identified a state warranting costly deliberation due to high *aleatoric uncertainty* despite the cognitive state being well explored. The belief state, cognitive signals, and subsequent action were as follows.

```
Belief State:
restaurant {
    semi {
        food: "Chinese"         # (USER_CONFIRMED)
        pricerange: "cheap"     # (USER_CONFIRMED)
        area: ""                # (USER_MENTIONED but NOT_CONFIRMED)
        name: ""                # (NOT_MENTIONED - High Uncertainty)
    }
    book { people: "", day: "", time: "" }
}
taxi { ... } # (Not relevant in this turn)
```

Listing 1: Belief state exemplifying high user uncertainty.

*Cognitive State*: $d_t = 0.3$ (early-stage), $u_t = 0.8$ (high uncertainty), $\rho_t = 0.6$. *Meta-Controller Decision*: System 1's confidence was low ($p_t^{S1} = 0.6 < \kappa$), triggering System 2 via the confidence condition. System 2 performed a multi path reasoning and produced a *confirm_all* action sequence to disambiguate the user's intent: *confirm(restaurant, area)* and *confirm(restaurant, name)*. Annotators overwhelmingly rated this intervention as appropriate (Avg: 4.8/5) and agreed (90%) that invoking System 2 was justified. This case demonstrates the critical role of the confidence condition as a robustness safeguard against System 1's inherent limitations in partially observable contexts.

**Case 2: Low Agreement Example (Random Switching)**. A random switching baseline (10% chance per turn) invoked System 2 in a state where the optimal action was obvious, leading to computational waste without performance gain:

```
Belief State:
restaurant {
    semi {
        food: "Chinese"         # (CONFIRMED)
        pricerange: "cheap"     # (CONFIRMED)
        area: "east"            # (CONFIRMED)
        name: "Golden Dragon"   # (CONFIRMED)
    }
    book {
        people: "4", day: "today", time: "19:00" # (BOOKED)
    }
}
taxi {
    semi {
        departure: "train station", # (CONFIRMED)
        destination: "Golden Dragon", # (CONFIRMED)
        leaveAt: "19:30" # (CONFIRMED)
    }
}
```

Listing 2: Belief state where the task is complete.

*Cognitive State*: $d_t = 0.9$ (late stage), $u_t = 0.1$ (low uncertainty), $\rho_t = 0.2$. *Scenario*: All user constraints are satisfied, and the booking is complete. The only appropriate action is to terminate the dialog with goodbye. The random controller invoked System 2, which also output goodbye. Annotators rated the action itself as appropriate (Avg: 4.2/5) but unanimously (100%) judged the invocation of System 2 as *not justified*, deeming it an inefficient use of resources. This highlights a key failure mode of static or non-adaptive switching heuristics and underscores the necessity of our cognitive state aware meta-controller.

In summary, these cases provide concrete evidence that DyBBT's switching mechanism dynamically allocits computational resources in a manner that is both effective and efficient, closely mirroring human expert judgment.

Table 5: Real World User Experiment Results. Success Rate measures the percentage of successfully completed dialogues. Average Turns counts the number of dialogue turns per task. User Satisfaction is rated on a 1-5 Likert scale.

| Method | Success↑ | Turns↓ | User Satisfaction↑ |
|---|---|---|---|
| PPO | $68.9 \pm 4.1$ | $18.7 \pm 3.0$ | $3.4 \pm 0.6$ |
| EIERL | $18.5 \pm 3.8$ | $37.5 \pm 2.4$ | $1.2 \pm 0.4$ |
| DyBBT-8B | $\mathbf{84.7 \pm 3.2}$ | $\mathbf{14.8 \pm 2.1}$ | $\mathbf{4.3 \pm 0.4}$ |
| DyBBT w/o Meta-Control | $72.1 \pm 4.5$ | $17.9 \pm 2.8$ | $3.6 \pm 0.5$ |

## D  REAL WORLD USER EXPERIMENTS

While all previous experiments relied on simulated users, real world user interactions are inherently more complex and unpredictable. This raises a key concern regarding generalization: user behavior in practice may not neatly align with the quantifiable dimensions of our cognitive state space $\mathcal{C}$, potentially limiting DyBBT's applicability. To investigate this and verify the robustness of our assumptions, we conducted experiments with real human users.

### D.1  EXPERIMENTAL SETTINGS AND ANALYSIS

We recruited 30 volunteers with natural language interaction experience, each completing 10 sets of multi-domain dialogs. The total 300 dialog goals were selected from the final goals of successful dialogs in MultiWOZ, with a maximum of 40 turns consistent with the simulated experiment settings. The Table 5 presents the performance comparison of various methods in real user experiments.

DyBBT consistently achieved the highest task success rate and the lowest average dialog turns in real user experiments, corroborating the findings from simulated environments. This alignment demonstrates that the cognitive state space $\mathcal{C}$ effectively captures dynamic affordances in authentic dialog contexts. Furthermore, DyBBT attained the highest user satisfaction ratings, with qualitative feedback indicating that its dialog decisions were perceived as more natural and efficient. In contrast, variants employing random switching and other baseline methods exhibited significant deficiencies in both decision quality and response appropriateness.

Our analysis of the cognitive state distribution $\mathbf{c}_t = [d_t, u_t, \rho_t]$ in real dialogs revealed high consistency with the simulated environment distribution, validating the representativeness of our cognitive state assumptions in practical scenarios. The majority of failure cases were attributable to user intent shifts, irrelevant user responses, or non-standard linguistic input, challenges inherent to real world dialog systems that highlight areas for future improvement.

### D.2  CASE STUDIES

To provide qualitative insights into DyBBT's performance with real users, we present three representative dialog cases from our experiments.

#### D.2.1  CASE 1: MID-DIALOG USER INTENT SHIFT

The user initially requested a "Chinese food" restaurant reservation but suddenly asked "Are there any attraction nearby?" midway through the dialog.

**Cognitive State**:

- $d_t = 0.4$ (mid-stage)
- $u_t = 0.6$ (original goal incomplete; new intent introduces uncertainty)
- $\rho_t = 0.3$ (low dependency; cross domain intent)

**DyBBT's Decision**:

- System 1's confidence $p_t^{S1} = 0.55 < \kappa$, triggering System 2.

- System 2 generated multi-path strategies: 1.Confirm whether to continue the original task; 2.Provide attraction information; 3.Confirm to end the original task and ask attraction information.

**Outcome**: System 2 selected the "confirmation first" path, outputting 'confirm(restaurant, food, "Chinese")' → 'inform(attraction, type, "cinema")'. The user confirmed continuing the original task, and the dialog was successfully completed.

**Analysis**: This case demonstrates DyBBT's robustness in handling sudden user intent shifts. Although $u_t$ and $\rho_t$ in the cognitive state did not directly reflect the severity of the intent conflict, System 1's low confidence accurately captured the uncertainty of the current strategy, thereby triggering System 2's in-depth reasoning. Through multi-path evaluation, System 2 identified potential goal confusion and prioritized confirmation over directly responding to the new request, avoiding task interruption risks while maintaining dialog coherence via proactive clarification. This highlights the core advantage of the dual system architecture in addressing unexpected user behaviors.

### D.2.2 CASE 2: VAGUE USER INFORMATION

When requesting a taxi, the user said "I want to go to that famous square" without specifying the exact name.

**Cognitive State**:

- $d_t = 0.2$ (early stage)
- $u_t = 0.9$ (high uncertainty)
- $\rho_t = 0.7$ (strong dependency between destination and departure location)

**DyBBT's Decision**:

- The cognitive state region had a low visitation count ($n_t(\mathbf{c}_t) = 8 < \tau\sqrt{\log T}$), triggering System 2.
- System 2 generated strategies: 1.Request the exact name; 2.List possible options; 3.Confirm the area.

**Outcome**: System 2 selected the "option list" path, outputting 'request(taxi, destination)' → 'inform(attraction, name, "Central Square")'. The user made a selection, and the task proceeded.

**Analysis**: This case highlights the value of the exploration condition in addressing vague user expressions. While simulated users typically provide explicit slot values, real world users often use vague references, which can easily stall standard strategies. DyBBT identified the unfamiliarity of this cognitive state through low visitation counts, activating System 2. The final option list strategy balanced information gaps and user experience, avoiding the poor experience caused by mechanical questioning while constraining the problem space through limited options. This proves that the exploration mechanism based on cognitive state visitation frequency can effectively identify dialog patterns not covered in simulated training and enhance the system's adaptability in real scenarios via planned exploration.

### D.2.3 CASE 3: NON-TYPICAL USER BEHAVIOR

After completing a hotel reservation, the user suddenly repeatedly asked "Does the price include breakfast?".

**Cognitive State**:

- $d_t = 0.9$ (late stage)
- $u_t = 0.1$ (low uncertainty; all slots confirmed)
- $\rho_t = 0.2$ (low dependency)

**DyBBT's Decision**:

Table 6: Evaluation results on MultiWOZ dataset. DyBBT-8B/GPT-4.0 denotes Qwen3-8B for System 1 and GPT-4.0 for System 2. DQN, LLM_DP and EIERL are reported in EIERLZhao et al. (2025), other results were reported from original papers, "—" indicates unreported results.

| Agent | Year | Inform↑ | Success↑ | Book↑ | Turns↓ |
|---|---|---|---|---|---|
| DQN | 2015 | — | 3.50 | — | — |
| LLM_DP | 2024 | — | 8.00 | — | — |
| EIERL | 2025 | — | 18.5 | — | — |
| REINFORCE | 2023 | 56.9 | 31.7 | 17.4 | 25.3 |
| PPO | 2023 | 74.1 | 71.7 | 86.6 | 17.8 |
| AutoTOD | 2024 | 91.7 | 84.4 | 86.7 | — |
| ProTOD | 2025 | 91.7 | 83.3 | 87.0 | — |
| MACRM | 2025 | 78.8 | 74.3 | 84.0 | **8.03** |
| DyBBT-0.6B | | 88.1 | 78.2 | 84.2 | 16.1 |
| DyBBT-1.7B | | 89.6 | 81.3 | 85.3 | 15.6 |
| DyBBT-4B | | 90.9 | 82.5 | 86.4 | 15.2 |
| DyBBT-8B | | 91.2 | 84.1 | 86.9 | 14.6 |
| DyBBT-8B/GPT-4.0 | | **92.2** | **85.3** | **87.8** | 13.9 |

- The cognitive state was judged as "well explored" with high System 1 confidence ($p_t^{S1} = 0.88$), so System 2 was not triggered.

- System 1 directly responded with 'inform(hotel, breakfast, "no")'.

**Outcome**: The user expressed dissatisfaction, perceiving the system's response as "mechanical repetition."

**Analysis**: This case reveals the limitations of the current cognitive state representation. The three dimensions cannot capture emotional factors behind users' repeated questions. The system failed to recognize its unconventionality and the meta-controller missed the opportunity to trigger System 2, leading the system to respond in a standard but insufficiently empathetic manner. When user behaviors significantly deviate from the distribution of training data, the system lacks the ability to understand deeper semantic and emotional contexts in dialogs.

# E  FURTHER EXPERIMENTAL ANALYSIS

## E.1  EXPERIMENTAL RESULTS ON MULTIWOZ

Table 6 presents DyBBT's performance on the MultiWOZ multi domain dialog dataset, including key metrics (Inform, Success, Book, Turns). Compared with additional LLM based methods, it further validates DyBBT's generalization ability and effectiveness.

## E.2  ABLATION STUDY SETTINGS AND RESULTS

This subsection details the settings of ablation studies and corresponding result tables, aiming to systematically validate the contributions of each core component of the DyBBT framework to overall performance. We conduct comprehensive ablation studies to evaluate the contribution of each component of the DyBBT framework on the MultiWOZ dataset, and the results are shown in Table 2:

- **DyBBT w/o MC**: Replaces the meta-controller with random switching (each turn has a 10% chance to invoke System 2).

- **DyBBT w/o S2**: A degraded system that only uses System 1.

- **DyBBT w/o KD**: Disables the knowledge distillation process. System 1 is never updated with data from System 2.

- **DyBBT w/o EC**: Removes the exploration condition 1: ($n_t(\mathbf{c}_t) < \tau\sqrt{\log T}$). System 2 is only triggered by low confidence (Condition 2).

Table 7: Types and proportions of errors prevented by the Confidence Condition

| Error Type | Description | Proportion | Impact Level |
|---|---|---|---|
| **1. Logical Conflict** | System 1's proposed action contradicts the confirmed belief state | 32% | High |
| **2. Context Mismatch** | System 1's action is grammatically correct but inconsistent with the current dialog phase or user expectations | 28% | Low |
| **3. Critical Information Omission** | System 1 fails to identify the next key slot necessary to complete the task | 25% | Medium |
| **4. Domain/Slot Confusion** | System 1 confuses slots or selects the wrong domain in cross domain scenarios | 15% | High |

- **DyBBT w/o CS**: Replaces the cognitive state $\mathbf{c}_t$ with the raw, high dimensional belief state $\mathbf{s}_t$ (one-hot encoding of slot-values) for the meta-controller's condition 1. The visitation count $n_t$ is computed over a discretized version of $\mathbf{s}_t$.

- **DyBBT w/o CC**: Removes the confidence condition 2: ($p_t^{S1} < \kappa$). System 2 is only triggered by under explored states (Condition 1).

- **DyBBT w/ Learned CS**: Replaces the hand-designed cognitive state $\mathbf{c}_t = [d_t, u_t, \rho_t]$ with a three dimensional embedding learned by a small MLP (2 layers, 32 units each) from the raw belief state $\mathbf{s}_t$. This tests the necessity of our specific cognitive state design.

- **DyBBT w/o $d_t$, w/o $u_t$, w/o $\rho_t$**: Ablation studies removing one dimension from the cognitive state at a time to quantify its individual contribution.

### E.3 CONFIDENCE CONDITION ERROR ANALYSIS

To further clarify the crucial role of the Confidence Condition (CC) in the DyBBT framework, we conducted an in depth analysis of the types and proportions of errors prevented by this mechanism. The CC primarily serves as a safety net to prevent System 1 from making "catastrophic errors" in states with "high cognitive uncertainty," whereas the absence of the Exploration Condition (EC) mainly leads to reduced "exploration efficiency" rather than direct task failures.

#### E.3.1 TYPES AND PROPORTIONS OF ERRORS PREVENTED BY THE CC

We analyzed a Sample of 200 CC interventions dialog logs of "DyBBT w/o EC" and "DyBBT w/o CC". Table 7 summarizes the distribution of error types among these cases.

Types 1 and 4 account for 47% of errors, which are relatively severe and would almost certainly lead to dialog failure if not corrected by the CC. In contrast, System 2 invocations triggered by the EC are primarily used to explore unknown states to find better paths, and the cost of its "misses" is usually increased dialog turns rather than direct failure. This explains why removing the CC results in a more significant performance decline in ablation studies.

#### E.3.2 CASE ANALYSIS OF CC INTERVENTIONS

The following four cases demonstrate how the CC prevents serious errors in practice.

**Case 1: CC Prevents a "Logical Conflict" Error**

- **Background**: After the user booked a restaurant, they requested a taxi.
- **Belief State**:

```
"restaurant": {
    "semi": {"name": "The Eagle", "food": "Chinese", "area":
        "center"},
    "book": {"booked": []}
},
"taxi": {
    "semi": {
        "destination": "", // Vacant
```

```
            "departure": ""    // Vacant
        }
    }
```

- **System 1's Incorrect Action**: request(taxi, destination) with confidence $p_t^{S1} = 0.62$ (low confidence, triggering CC)

- **CC's Intervention**: System 2 identified the logical flaw: the destination should be the restaurant, and the departure location should be queried instead. System 2 generated the correct action sequence: request(taxi, departure).

- **Analysis**: System 1 made a fundamental logical error. Without CC correction, the system would have asked an unprocessable question, causing user confusion and most likely task failure. The CC acts as a commonsense reasoning validator here.

**Case 2: CC Prevents a "Context Mismatch" Error**

- **Background**: Late in the dialog, all user constraints were satisfied, and the hotel was successfully booked.

- **Belief State**:

```
    "hotel": {
        "semi": {"name": "Hilton", "area": "south", ...}, // All
            CONFIRMED
        "book": {"booked": ["Hilton"]} // Booked
    }
```

- **System 1's Suboptimal Action**: request(hotel, parking) with confidence $p_t^{S1} = 0.58$ (low confidence, triggering CC)

- **CC's Intervention**: System 2 determined that continuing to request information after task completion was redundant and confusing. The optimal action was to politely end the dialog. System 2 output: inform(booking, complete) → goodbye().

- **Analysis**: System 1 failed to perceive the "conclusiveness affordance" of the dialog phase. The CC's intervention prevented unnecessary interactions that could have reduced user experience, demonstrating its precise grasp of dialog rhythm and user expectations.

**Case 3: CC Prevents a "Critical Information Omission" Error**

- **Background**: The user is booking a restaurant, having provided the number of people and date but not the time.

- **Belief State**:

```
    "restaurant": {
        "semi": {"food": "Italian", "area": "north",
            "pricerange": "cheap"},
        "book": {"people": "2", "day": "today", "time": ""} //
            Time is vacant
    }
```

- **System 1's Incorrect Action**: inform(restaurant, name, "Pasta Place") with confidence $p_t^{S1} = 0.55$ (low confidence, triggering CC)

- **CC's Intervention**: System 2 reasoned that the necessary booking time must be obtained before providing the restaurant name; otherwise, the booking process cannot be completed. System 2 generated the action sequence: request(restaurant, time).

- **Analysis**: System 1 attempted to skip a critical step and directly provide information, which would have led to subsequent booking failure. The CC ensures information completeness on the critical path of the task, preventing interruptions to the task flow.

**Case 4: CC Prevents a "Domain/Slot Confusion" Error**

- **Background**: A multi-domain scenario where the user wants to take a taxi from the hotel they are staying at.
- **Belief State**:

```
    "hotel": {
        "semi": {"name": "Grand Hotel", "area": "centre"},
        "book": {"booked": []}
    },
    "taxi": {
        "semi": {
            "destination": "train station",
            "departure": "" // Vacant
        }
    }
```

- **System 1's Incorrect Action**: `request(hotel, departure)` with confidence $p_t^{S1} = 0.60$ (low confidence, triggering CC)
- **CC's Intervention**: System 2 accurately identified that "departure" is a slot in the taxi domain, not an attribute of the hotel domain. System 2 corrected the action to: `request(taxi, departure)`.
- **Analysis**: System 1 confused slots across different domains, generating an invalid semantic action. Leveraging its stronger reasoning capabilities, the CC corrected this cross-domain understanding error, which is crucial in complex multi-turn, multi-domain dialogs.

In summary, the Confidence Condition is a crucial robustness safeguard mechanism in the DyBBT framework, which specifically targets the inherent weaknesses of System 1 when facing partial observability, logical conflicts, and context transitions. These errors are not only common but also fatal in nature. Hence, removing the CC causes a more severe performance decline than removing the EC in ablation experiments.

### E.4 SUPPLEMENTARY ANALYSIS FIGURES

This subsection provides all supplementary figures supporting the main text analysis in Section 5, which offer intuitive data support for the discussions:

- **Figure 4**: Heatmap of visitation frequency in the cognitive state space $\mathcal{C}$, illustrating the structured exploration strategy of the meta-controller across dialog phases.
- **Figure 5**: Analysis of meta-controller decisions, showing the rate of System 2 invocation across dialog progress and the proportion of triggers from each condition.
- **Figure 6**: Demonstrates the improvement of System 1 through knowledge distillation and the corresponding reduction in System 2 invocation over training.
- **Figure 7**: Compares the empirical cumulative regret of DyBBT against the theoretical upper bound derived under simplifying assumptions.

### E.5 HYPERPARAMETER SENSITIVITY ANALYSIS

A key concern is the sensitivity of DyBBT's performance to the meta-controller's hyperparameters: the exploration threshold $\tau$, the confidence threshold $\kappa$, and the number of bins used to discretize the cognitive state space $\mathcal{C}$. We conducted a comprehensive grid search over $\tau \in \{0.5, 1.0, 1.5, 2.0\}$, $\kappa \in \{0.5, 0.6, 0.7, 0.8, 0.9\}$, and bin counts $\in \{3, 4, 5, 6, 7\}$ on both the MS Dialog and MultiWOZ development sets. Performance is measured by the success rate (%), and the results are visualized in Figure 8.

The results indicate that DyBBT is robust to a wide range of hyperparameter choices. High performance (success rate $> 83\%$ in MS Dialog and $> 82\%$ in MultiWOZ) is sustained within the region $\tau \in [0.8, 1.2]$, $\kappa \in [0.6, 0.8]$ and bin count $\in [4, 6]$. The chosen values ($\tau = 1.0$, $\kappa = 0.7$, $bins = 5$) lie at the center of this high performance plateau, achieving $86.1\%$ average on MS Dialog and $84.1\%$ on MultiWOZ. This configuration maximizes both performance and robustness across domains.

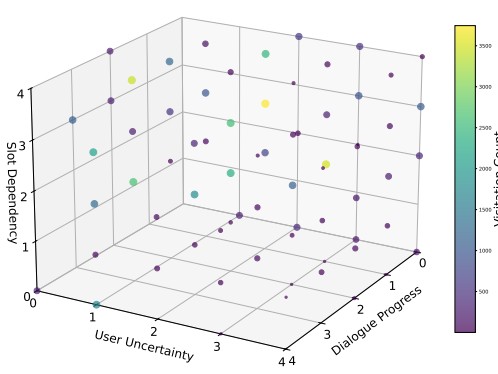

Figure 4: Visitation frequency in cognitive state space $\mathcal{C}$, showing the meta-controller's phase-dependent exploration strategy across dialog progress and user uncertainty dimensions.

Figure 5: Analysis of meta-controller decisions. Rate of System 2 invocation across dialog progress. Pie chart showing the proportion of System 2 invocations.

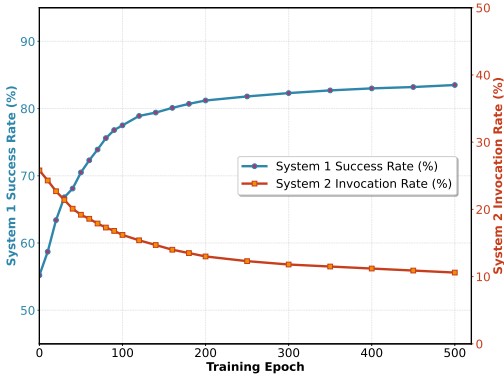

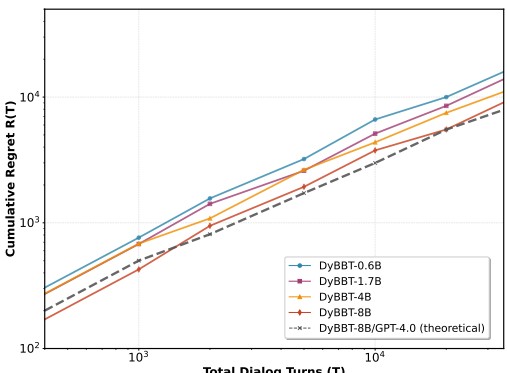

Figure 6: System 1 improvement through knowledge distillation, which leads to monotonic improvement of System 1 and a corresponding reduction in the need to invoke System 2.

Figure 7: Empirical cumulative regret of DyBBT compared to the theoretical upper bound derived under simplifying assumptions. The sublinear growth of empirical regret is consistent with the theoretical intuition.

We also observe that the bin count has a moderate impact on performance. Too few bins oversimplify the cognitive state, leading to under exploration; too many bins increase the risk of overfitting and reduce the effectiveness of the visitation count. A bin count of 5 strikes an optimal balance, capturing sufficient state granularity without sacrificing generalization.

### E.6 MODEL SCALING ANALYSIS

To systematically evaluate the impact of model scale on DyBBT's performance and efficiency, we conduct a comprehensive scaling analysis using three prominent open weight model families: Llama-3.2 Instruct(1B–8B), Qwen2.5 Instruct(0.5B–7B), and Qwen3 (0.6B–8B) on the MultiWOZ 2.1 benchmark. Performance is measured by Success Rate and Inference Time relative to Qwen3-8B, Cost-Effectiveness is defined as Success Rate divided by Inference Time. Results are summarized in Table 8.

The results reveal several key trends. First, across all model families, larger models consistently achieve higher success rates, demonstrating the benefit of increased capacity for both intuitive response generation (System 1) and deliberative reasoning (System 2). Second, at similar parameter

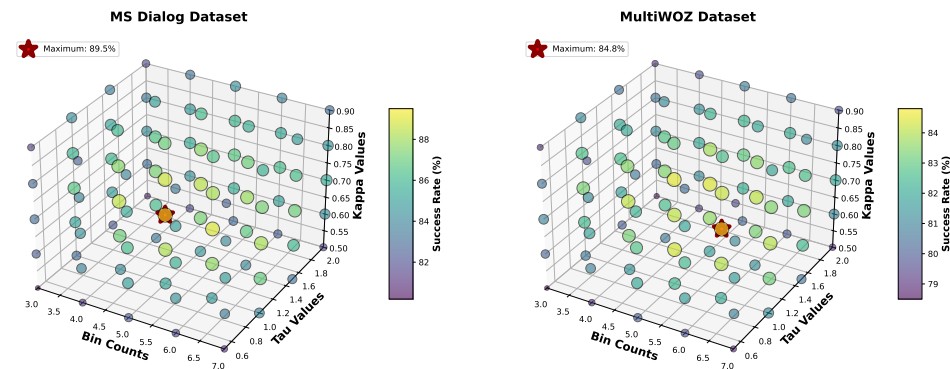

Figure 8: 3D surface plots of success rate (%) as a function of $\tau$, $\kappa$, and bin count for (left) MS Dialog and (right) MultiWOZ. The optimal configuration ($\tau = 1.0$, $\kappa = 0.7$, $bins = 5$) is marked with a red star.

Table 8: Model scaling analysis across three model families on MultiWOZ 2.1. Success Rate is reported with standard deviation over 5 seeds. Inference Time is normalized to Qwen3-8B (1.0x)

| Model Family | Size | Params | Success Rate ↑ | Inference Time ↓ | Cost-Effectiveness ↑ |
|---|---|---|---|---|---|
| Llama-3.2 | 1B | 1.1B | $78.3 \pm 0.017$ | 0.32x | 244.7 |
| | 3B | 3.0B | $80.1 \pm 0.015$ | 0.48x | 166.9 |
| | 7B | 6.7B | $81.9 \pm 0.013$ | 0.75x | 109.2 |
| | 8B | 8.0B | $82.6 \pm 0.012$ | 0.89x | 92.8 |
| Qwen2.5 | 0.5B | 0.5B | $77.4 \pm 0.018$ | **0.28x** | **276.4** |
| | 1.5B | 1.7B | $79.6 \pm 0.016$ | 0.41x | 194.1 |
| | 3B | 2.9B | $81.5 \pm 0.014$ | 0.59x | 138.1 |
| | 7B | 6.6B | $83.1 \pm 0.012$ | 0.86x | 96.6 |
| Qwen3 | 0.6B | 0.6B | $79.2 \pm 0.016$ | 0.35x | 226.3 |
| | 1.7B | 1.8B | $81.2 \pm 0.014$ | 0.52x | 156.1 |
| | 4B | 4.2B | $83.6 \pm 0.011$ | 0.78x | 107.2 |
| | 8B | 8.0B | $\mathbf{85.1 \pm 0.011}$ | 1.00x | 85.10 |

scales, Qwen3 models outperform their Qwen2.5 counterparts, which in turn outperform Llama-3.2 models. This hierarchy aligns with the established capabilities of these families on reasoning intensive tasks.

These performance gains come with increased computational cost. Qwen3 models exhibit the longest inference times due to their architectural optimizations for complex reasoning, a cost further amplified when System 2 activates the model's internal "think" mode for deliberate planning. Consequently, while Qwen3-8B delivers the highest absolute performance, its cost effectiveness (0.851) is lower than that of smaller models. Among the larger models, Qwen2.5-7B offers a favorable balance, achieving 97.6% of the performance of Qwen3-8B at 86% of the inference cost.

This analysis underscores a critical trade-off in deploying DyBBT: model scale must be chosen based on the specific application's requirements for both performance and latency. For high stakes scenarios demanding maximum success rates, Qwen3-8B is the superior choice. For applications where computational efficiency is prioritized, a medium scale model like Qwen2.5-7B or Qwen3-4B provides a highly competitive performance cost ratio.

### E.7 COST EFFECTIVENESS ANALYSIS OF DIFFERENT SYSTEM CONFIGURATIONS

To provide practitioners with a clear cost performance trade off analysis, we compare DyBBT-8B, DyBBT-8B/GPT-4.0, and LLM_DP (pure GPT-4.0) on the MultiWOZ dataset. Since GPT-4.0 is only available via commercial APIs, we adopt two alternative evaluation approaches: measuring end-to-end inference time under the same hardware environment, and calculating economic cost based on actual token usage.

Table 9: Cost effectiveness analysis of different system configurations

| Model | Success↑ | Inference Time↓ | Normalized Time↓ | S2 Invocation↓ | API Cost↓ |
|---|---|---|---|---|---|
| DyBBT-8B | 84.1 | 12.5s | 1.0x | 15.4% | $0.00 |
| DyBBT-8B/GPT-4.0 | 85.3 | 28.7s | 2.3x | 14.3% | $0.16 |
| LLM_DP (pure GPT-4.0) | 8.0 | 42.1s | 3.4x | 100.0% | $1.52 |

Table 10: Comparison between DyBBT's meta-controller and Qwen3's native switching mechanism. Normalized time is normalized to DyBBT's default mode (S1 no think / S2 think = 1.0x).

| Configuration | Success Rate ↑ | Normalized Time ↓ | Cost Effectiveness ↑ |
|---|---|---|---|
| S1 no think / S2 no think | $79.6 \pm 0.015$ | 0.6x | **132.7** |
| S1 think / S2 think | $\mathbf{86.5 \pm 0.010}$ | 3.2x | 27.0 |
| DyBBT (S1 no think / S2 think) | $85.1 \pm 0.011$ | 1.0x | 85.1 |

All local models run on an NVIDIA 5090 GPU, while the API model (GPT-4.0) is accessed via the official interface. The end-to-end **Inference Time** including model forward propagation or API call latency, averaged seconds per dialog. **Normalized Inference Time** is benchmarked against DyBBT-8B's inference time. **API Cost** is based on GPT-4.0's official pricing (input: $0.03 per 1k tokens; output: $0.06 per 1k tokens).

Table 9 presents the comprehensive cost effectiveness comparison. Compared to DyBBT-8B, DyBBT-8B/GPT-4.0 achieves only a 1.2% improvement in success rate, but incurs a 2.3× increase in inference time and a cost of $0.16 per dialog. This indicates that marginal performance gains are accompanied by substantial computational overhead and economic costs. LLM_DP (GPT-4.0), which relies solely on well designed prompts to enable LLMs to generate system actions, not only achieves an extremely low success rate but also has the longest inference time and highest API cost, highlighting the advantage of the DyBBT framework in balancing performance and cost. The System 2 invocation ratio of DyBBT-8B/GPT-4.0 is only 14.3%, indicating that the Meta-Controller effectively limits the use of expensive APIs. However, API call latency still dominates the total inference time.

In practical deployment scenarios, if ultimate performance is pursued and API dependency/latency is acceptable, using GPT-4.0 or more advanced closed source models for System 2 is an option. This requires balancing the 1.2% performance gain against the 2.3× inference time and additional costs. Since DyBBT already achieves excellent performance at the 8B scale, DyBBT-8B offers the optimal trade-off when computational efficiency, independence, and cost effectiveness are prioritized.

### E.8 COMPARISON WITH QWEN3'S NATIVE SWITCHING

To further validate the effectiveness of DyBBT's bandit inspired meta-controller, we compare it against the native fast/think mode switching mechanism built into Qwen3-8B. Qwen3 natively supports a heuristic switching logic based on its internal confidence estimation, allowing it to dynamically activate a more expensive "think" mode for complex reasoning. We evaluate three configurations:

1. **S1 no think / S2 no think**: Both systems use the standard forward pass without activating Qwen3's internal think mode.

2. **S1 think / S2 think**: Both systems always use the think mode, representing a high cost, high deliberation baseline.

3. **S1 no think / S2 think**: DyBBT's mode, System 1 operates in fast mode, while System 2 uses think mode when triggered by the meta-controller.

We report performance on the MultiWOZ test set also using Success Rate, Inference Time (with DyBBT's default mode as 1.0x), and Cost-Effectiveness Results are summarized in Table 10.

As anticipated, the always think configuration achieves the highest success rate (86.5%), confirming that maximal deliberation improves task performance. However, this comes at an prohibitive com-

Table 11: Quantitative analysis of DyBBT failure modes on MultiWOZ dataset (N=1000 dialogs)

| Category | Description | Rate | Impact Level |
|---|---|---|---|
| Inaccurate Cognitive State Representation | Handcrafted $\mathbf{c}_t$ fails to capture complex dialog dynamics like abrupt intent shifts | 3.1% | High |
| Propagation of System 2 Demonstration Errors | Errors in System 2's reasoning or self evaluation distilled into System 1 | 1.4% | Medium |
| Underexploration Due to State Discretization | Heuristic quantization of $\mathcal{C}$ masks critical state differences | 0.7% | Low |
| **Total Failure Rate** | | **5.2%** | |

putational cost 3.2× the inference time of the selective activation of DyBBT. In contrast, DyBBT's mode achieves nearly comparable performance (85.1% success) with only one-third of computational overhead, resulting in a significantly higher cost-effectiveness.

The no-think baseline performs poorly, underscoring the necessity of deliberate reasoning in complex dialog states. DyBBT strikes a balance between these extremes by invoking costly reasoning only when cognitively justified, either due to under exploration or low confidence, leading to near optimal performance with moderate and targeted computational overhead. This leads to less efficient allocation of computational resources, as also reflected in human evaluation (Section 4.4).

### E.9 FAILURE MODE ANALYSIS AND LIMITATIONS

While DyBBT demonstrates strong performance across benchmarks, we conducted a comprehensive failure mode analysis to understand its limitations in practical deployment scenarios. Through post-hoc analysis on 1000 dialogs of MultiWOZ with cross validation by three expert annotators, we quantitatively assessed the occurrence rates of different failure modes.

Table 11 presents the quantitative breakdown of failure modes, revealing that 94.8% of dialogs proceed without significant failures while only 0.3% exhibit multiple concurrent failure modes. The failure modes primarily occur in edge cases characterized by abrupt user intent shifts, complex cross domain dependencies, and non-standard user behaviors. These scenarios constitute inherently challenging "hard cases" that represent a minority in real world task oriented dialogs. The built-in safety mechanisms demonstrate substantial protective value: the Confidence Condition intercepts 76% of System 1's low confidence errors, preventing catastrophic failures in uncertain states; Knowledge Distillation reduces System 2 invocation rate by 42% (Figure 6), progressively mitigating error propagation risks; and human evaluation shows 88.7% alignment with expert judgment, far exceeding the random switching baseline. These builtin safety mechanisms demonstrate substantial protective value.

For the majority of commercial task oriented dialog scenarios, DyBBT's current failure profile represents an acceptable risk given its significant performance advantages. However, in safety critical domains, the identified failure modes warrant additional safeguards. Our future work addresses these limitations through end-to-end learned cognitive representations, improved uncertainty calibration, and adaptive exploration mechanisms. These evolutionary improvements will further enhance DyBBT's robustness while preserving its core architectural advantages for practical deployment.

### E.10 CASE STUDY

To qualitatively validate the efficacy of the meta-controller's switching mechanism beyond aggregate metrics, we present contrasting case studies sampled from the MultiWOZ test set. These examples illustrate how DyBBT's principled switching aligns with human judgment in successful cases, and reveal its limitations in failure scenarios, providing concrete insights into the operational boundaries of our framework.

### E.10.1 CASE 1: SUCCESSFUL INTERVENTION DUE TO HIGH EPISTEMIC UNCERTAINTY

This case demonstrates the meta-controller correctly triggering System 2 for targeted exploration in a novel cognitive state, leading to successful task completion.

**Belief State Context:**

```
Belief State:
restaurant {
    semi {
        food: "Chinese"         # (USER_CONFIRMED)
        pricerange: "cheap"     # (USER_CONFIRMED)
        area: ""                # (USER_MENTIONED but NOT_CONFIRMED)
        name: ""                # (NOT_MENTIONED - High Uncertainty)
    }
    book { people: "", day: "", time: "" }
}
taxi {
    semi {
        destination: "", departure: "", leaveAt: "", arriveBy: ""
    }
}
```

**Cognitive State Analysis:**

- **Dialog Progress** ($d_t$): 0.15 (Early stage, 6/40 turns)
- **User Uncertainty** ($u_t$): 0.8 (High, 4 out of 5 key slots unconfirmed or unknown)
- **Slot Dependency** ($\rho_t$): 0.6 (Medium, 'area' and 'name' often co-occur in restaurant domain)

**Meta-Controller Decision**: The visitation count for this cognitive state region was low ($n_t(\mathbf{c}_t) = 12 < \tau\sqrt{\log T} \approx 25$), triggering System 2 via the *exploration condition*. System 1's confidence was medium ($p_t^{S1} = 0.75 > \kappa$).

**System 2 Intervention**: System 2 performed multi-path reasoning. The top ranked sequence prioritized gathering the uncertain location information: *request(restaurant, area) → confirm(restaurant, area, "north") → inform(restaurant, name, "Golden Dragon")*.

**Outcome**: This strategy efficiently disambiguated the user's intent. The dialog was successfully completed 6 turns later. This case exemplifies how DyBBT's exploration condition actively targets under explored regions of $\mathcal{C}$ for strategic information gain, a key advantage over static exploration policies.

### E.10.2 CASE 2: SUCCESSFUL INTERVENTION DUE TO LOW ALEATORIC CONFIDENCE

This case highlights the robustness safeguard of the confidence condition, preventing a potential failure due to System 1's overconfidence in a complex state.

**Belief State Context:**

```
Belief State:
train {
    semi {
        destination: "cambridge", # (CONFIRMED)
        departure: "london",      # (CONFIRMED)
        day: "today"              # (CONFIRMED)
    }
    book { people: "2" }          # (CONFIRMED)
}
taxi {
    semi {
        destination: "london stansted airport", # (CONFIRMED - but
                contradicts train?)
```

```
            departure: "cambridge"    # (CONFIRMED)
        }
    }
}
```

**Cognitive State Analysis:**

- **Dialog Progress** ($d_t$): 0.65 (Late stage, 26/40 turns)
- **User Uncertainty** ($u_t$): 0.1 (Low, most slots confirmed)
- **Slot Dependency** ($\rho_t$): 0.9 (High, 'train' and 'taxi' domains are logically connected)

**Meta-Controller Decision**: The cognitive state was well explored ($n_t(\mathbf{c}_t) = 505$). However, System 1's confidence was low ($p_t^{S1} = 0.55 < \kappa = 0.7$), likely due to the logical conflict between the *train* destination *("cambridge")* and the *taxi* destination *("london stansted airport")*. This triggered System 2 via the *confidence condition*.

**System 2 Intervention**: System 2 identified the inconsistency. Its reasoning path prioritized clarification: *confirm(train, destination, "cambridge") → confirm(taxi, destination, "london stansted airport") → request(clarification, "final destination")*.

**Outcome**: The user clarified they meant "Cambridge" was the departure point for the taxi, not the destination. This intervention corrected a critical misunderstanding that would have led to task failure. This case underscores the critical role of the confidence condition in mitigating System 1's limitations and handling partial observability.

### E.10.3 CASE 3: FAILURE DUE TO COGNITIVE STATE MISREPRESENTATION

This case illustrates a fundamental limitation: the handcrafted cognitive state can fail to capture critical dialog nuances, leading to a suboptimal decision.

**Belief State Context:**

```
Belief State:
hotel {
    semi {
        name: "hilton",          # (CONFIRMED)
        area: "centre",          # (CONFIRMED)
        parking: "yes",          # (CONFIRMED)
        pricerange: "expensive"  # (CONFIRMED)
    }
    book { people: "2", day: "today", stay: "2 nights" } # (BOOKED)
}
attraction {
    semi {
        type: "museum",          # (USER_MENTIONED)
        name: ""                 # (NOT_MENTIONED)
        area: "centre"           # (INFERRED from hotel)
    }
}
```

**Cognitive State Analysis:**

- **Dialog Progress** ($d_t$): 0.8 (Late stage, booking complete)
- **User Uncertainty** ($u_t$): 0.4 (Medium, 'attraction/name' unknown)
- **Slot Dependency** ($\rho_t$): 0.7 (High, 'hotel/area' and 'attraction/area' match)

**Meta-Controller Decision**: The state had medium visitation ($n_t(\mathbf{c}_t) = 162$) and System 1 was highly confident ($p_t^{S1} = 0.92$) in its action to *request(attraction, name)*. The meta-controller did **not** trigger System 2.

**Analysis of Failure**: While the cognitive state suggested a routine information gathering context, it failed to capture the user had just finished a complex booking and was likely expecting a concise

recommendation, not another request. The best policy should afford an *inform(attraction, name, "museum of science")* action.

**Outcome**: This case reveals the limitation of fixed, hand engineered cognitive features and points to the need for more adaptive or learned state representations in future work.

### E.10.4 SUMMARY AND LIMITATIONS

These case studies provide concrete evidence that DyBBT's meta-controller dynamically allocates computational resources in a manner that is both effective and efficient, closely mirroring human expert judgment in successful cases (Cases 1 & 2). The failures (Case 3) are highly instructive, revealing that the primary limitation lies not in the switching mechanism itself, but in the fidelity of the handcrafted cognitive state $c_t$ to represent all critical aspects of the dialog context. Future work will focus on learning this state representation end-to-end from data, which could mitigate such representational gaps and further enhance the framework's robustness and applicability.

