# OpenReview forum: "DyBBT: Dynamic Balance via Bandit-inspired Targeting for Dialogue Policy with Cognitive Dual-Systems"
_ICLR.cc/2026/Conference — ICLR 2026 Conference Withdrawn Submission_

### Official Review · Reviewer_PbHc · 2025-10-30

**Soundness:** 2
**Presentation:** 2
**Contribution:** 3
**Rating:** 4
**Confidence:** 4

**Summary:**

This paper focuses on proposing a more powerful dialogue policy algorithm for current task-oriented dialogue systems. Specifically, this study concentrates on how to seamlessly switch between system I (intuitive inference) and system II (reasoning-based inference) to trade off the exploration ability and the exploitation of unknown decisions. More specifically, they design a meta controller to switch the decision to system II when it cannot fulfill the requirements.

**Strengths:**

+ The proposed method provides a comprehensive decision for current task-oriented dialogue systems. The idea that employs two thinking patterns for the same problem is really interesting.

+ They provide the intuition of how their meta-controller is derived from existing works, which is convincing.

+ They have provided some experiments on task-oriented dialogue policy, which indicates the effectiveness of their policy.

**Weaknesses:**

+ The organization and writing of this paper are significantly insufficient, especially in Section 3. For instance, I cannot understand and the paper does not mention how Assumption 1 in Section 3.1.2 is used in their method and also how Section 3.1.3 guides the design of their methodology. For the latter one, I suppose it is used to design the meta controller, but how and why? There is no mention or even citation of Section 3.1.2 and 3.1.3 in the subsequent method descriptions. Besides, some significant mathematical notations are not explained. For instance, what is the meaning of $T$?

+ It is not reasonable for the implementation of System II shown in Section 3.2.2. In this part, complex and in-depth reasoning (what is system II) is simply implemented by an instruction-driven prompt-based inference. I think this is over-simplified. I hope authors can explain this point, and compare it with some commonly used reasoning methods, such as RL-based reasoning fine-tuning.

+ In Table 1, it is unnecessary to provide the training checkpoint which is not converged. On the contrary, I'd like to see some more interesting experiments about the core idea of this algorithm. For instance, how frequently does the algorithm switch to system II? Can you provide any ablation study to exhibit the effectiveness of your system II? How are the thresholds selected in your meta-controller?

**Questions:**

Please refer to the weaknesses.

---

> ### Author Response · Authors · 2025-11-20
> **Response to Weakness 1: Clarity Between Theoretical and Methodological Sections**
>
> We sincerely appreciate your careful reading of the paper and your valuable comments. Due to space constraints, some discussions were placed in the appendices, leading to insufficient clarity in Section 3. We highly value your feedback and address your specific questions one by one as follows:
>
> ## 1. How Assumption 1 is Applied in the Methodology
>
> Assumption 1 (Lipschitz Smooth Reward in $\mathcal{C}$) is not directly used in algorithm implementation but serves as the foundation for **theoretical analysis**, specifically to derive that the regret bound is sublinear. This assumption guarantees the smoothness of the cognitive state space and the sublinear nature of regret confirms that exploration in the low dimensional cognitive state space $\mathcal{C}$ is feasible.
>
> ## 2. How the Exploration Bonus Formula in Section 3.1.3 Guides Method Design
>
> The exploration bonus formula ($\text{Exploration-Bonus}(t) \propto \sqrt{\frac{\log T}{n_t(\mathbf{c}_t)}}$) **directly informs** the first triggering condition of the meta-controller (Equation 2: $n_t(\mathbf{c}_t) < \tau \sqrt{\log T}$). Derived from the Upper Confidence Bound (UCB) principle in multi-armed bandit theory, the formula states that the exploration bonus is inversely proportional to the square root of the visitation count. This ensures more exploration in under visited regions of the cognitive state space to balance exploration and exploitation.
>
> Condition 1 to trigger S2 is an heuristic application of the exploration bonus formula. When the visitation count is low, S2 is activated for deliberate exploration, thereby covering unknown regions of the state space. This design addresses the complexity of exploration in dialog POMDPs by mapping high dimensional problems to a low dimensional cognitive space, making exploration computable and efficient.
>
> ## 3. The Meaning of the Mathematical Symbol $T$
>
>  $T$ denotes the **total training steps**, the sum of all dialog turns throughout the entire training process across multiple dialog episodes. It is used to calculate cumulative regret, and its growth is associated with training time. This is mentioned in the regret analysis of the paper (Section 3.1.3 and Appendix A.2). We will refine the revised version and add the following explanation after Equation (1):
>
> > where $T$ denotes the total training steps (the sum of all dialog turns), and $n_t(\mathbf{c}_t)$ denotes the visitation count of the cognitive state $\mathbf{c}_t$ (see Section 3.2 for details).
>
> ## 4. The Lack of Citations to Sections 3.1.2 and 3.1.3 in Subsequent Method Descriptions
>
> The theoretical section (Section 3.1) is intended to lay the foundation for the methodology (Section 3.2). We acknowledge that interleaving theoretical and algorithmic discussions across the main text and appendices, without explicit connections, has led to confusion.
>
> We will add a transitional paragraph at the beginning of Section 3.2, summarizing how theoretical insights guide the overall architectural design:
>
> > Building upon the theoretical foundations laid out in Section 3.1, the DyBBT architecture operationalizes key insights into a practical dual-system framework. The structured cognitive state space $\mathcal{C}$ (Section 3.1.1) serves as a low dimensional bridge for exploration, while the Lipschitz smoothness assumption (Assumption 1, Section 3.1.2) underpins the theoretical feasibility of efficient exploration in $\mathcal{C}$. Crucially, the bandit inspired exploration principle (Eq. 1, Section 3.1.3) directly informs the design of the meta-controller, translating theoretical regret bounds into an adaptive switching mechanism. Together, these elements enable a dynamic balance between fast inference and deliberate reasoning, as detailed in the following subsections.
>
> We will also explicitly cite Sections 3.1.2 and 3.1.3 in the meta-controller description (Section 3.2.3) and explain how theory is translated into practice:
>
> > The meta-controller’s hybrid rule (Eq. 2) is a pragmatic instantiation of the theoretical principles from Sections 3.1.2 and 3.1.3. Condition 1, $n_t(\mathbf{c}_t) < \tau \sqrt{\log T}$, directly implements the exploration bonus derived in Eq. 1 (Section 3.1.3), targeting epistemic uncertainty in under explored regions of $\mathcal{C}$. This ensures sublinear regret growth as justified by the Lipschitz continuity of rewards (Assumption 1, Section 3.1.2). Condition 2, $p_t^{S1} < \kappa$, acts as an empirical safeguard against aleatoric uncertainty, complementing the theoretical framework with robustness to model imperfections and partial observability. By bridging bandit theory with cognitive affordances, the meta-controller dynamically allocates costly System 2 resources only when theoretically justified or empirically necessary, achieving both efficiency and reliability.
>
> Thank you again for pointing out these issues. We will submit the complete revised version of the paper before the deadline.

---

> > ### Comment · Reviewer_PbHc · 2025-11-26
> >
> > Dear authors,
> >
> > Thanks for your response and the revision. Regarding W1, now I know the meaning of $T$.
> >
> > Another question comes from your response:
> >
> > > **Assumption 1 is not directly used in algorithm implementation but serves as the foundation for theoretical analysis**, specifically to derive that the regret bound is sublinear. This assumption guarantees the smoothness of the cognitive state space and the sublinear nature of regret confirms that exploration in the low dimensional cognitive state space is feasible.
> >
> > My concern here is: if there is even no theorem in the main text using this assumption, why it is necessary to list it? I go back to read your new Section 3.1 and 3.2, and I still cannot figure out how your theoretical analysis organized.
> >
> > > We will submit the complete revised version of the paper before the deadline.
> >
> > Thank you for your promise. I will try my best to review the new revision later.

---

> ### Author Response · Authors · 2025-11-20
> **Response to Weakness 2: Implementation of System 2's Reasoning (1/2)**
>
> We sincerely appreciate your insightful observations and valuable comments on System 2's implementation. We fully understand your concern that the prompt based reasoning approach may seem overly simplified. Below, we elaborate on the theoretical basis, practical considerations, and comparisons with common reasoning methods underlying this design choice.
>
> ## 1. Design Philosophy: Leveraging LLMs’ Zero-Shot Reasoning Capabilities Instead of Training New Models
> Our method adopts the "Prompt as Programming" paradigm, guiding LLMs to perform systematic reasoning through structured prompts. System 2 is not treated as a "reasoning module" requiring training or fine-tuning from scratch, but as a ready-to-use, powerful reasoning resource to be invoked. Modern LLMs such as Qwen3 or GPT series have acquired strong thinking capabilities during pre-training. Through carefully designed prompts, we can directly stimulate this inherent reasoning ability without time consuming and domain specific fine-tuning.
>
> The RL-based reasoning fine-tuning you mentioned is indeed an alternative path. However, it faces three critical challenges:
> - Designing appropriate reward functions for each reasoning step in complex multi-domain, multi-turn dialogs is extremely difficult.
> - RL training itself is unstable, especially in scenarios requiring multi-step reasoning.
> - Training a dedicated reasoning model demands substantial computational resources and time, which contradicts DyBBT’s core goal of pursuing **computational efficiency**.
>
> DyBBT approach offers excellent flexibility and generality, making it the most commonly used solution in engineering practice. When extending to new dialog domains or modifying tasks, we only need to adjust the prompt instead of retraining the entire System 2 model.
>
>
> ## 2. Functional Equivalence to Classical Reasoning Methods
> Our approach is actually **functionally equivalent** to various classical reasoning methods:
>
> - **Tree of Thoughts (ToT)**: The three strategy sequences generated by System 2 essentially form a **three branch reasoning tree**, where each path represents a distinct chain of thought. The highest scoring path is ultimately selected, aligning with ToT’s core idea.
> - **Model-based Planning**: By having the LLM predict the success probability of different action sequences, we effectively use the LLM as a **surrogate for the world model** to perform model based planning.
> - **Monte Carlo Tree Search (MCTS)**: While smaller in scale, our multi-path generation and evaluation can be seen as a simplified version of the **expansion and evaluation steps** in MCTS.

---

> ### Author Response · Authors · 2025-11-20
> **Response to Weakness 2: Implementation of System 2's Reasoning (2/2)**
>
> ## 3. System 2’s Implementation Is Far Beyond a "Simple Prompt"
> System 2 leverages a LLM with "thinking" capabilities. While formally prompt based, our System 2 design incorporates a **multi-layered** and **structured reasoning mechanism**. The prompt template is as follows:
>
> ```
> You are the deliberative reasoner (System 2) of a task-oriented dialog system. Your goal is to generate diverse, high-quality action plans when the meta-controller detects a need for deeper reasoning, either due to unfamiliar cognitive states or low confidence from System 1.
>
> **Current Belief State:**
> {belief_state}
>
> **Available Actions:**
> {available_actions}
>
> **Cognitive State Context:**
> - Dialog Progress: {d_t}
> - User Uncertainty: {u_t}
> - Slot Dependency: {p_t}
>
> **Trigger Reason:** {trigger_reason}
>
> **Reasoning Guidelines:**
> 1. **Leverage cognitive signals**:
>    - If progress is low, focus on information gathering.
>    - If uncertainty is high, prioritize clarifying or confirming actions.
>    - If slot dependency is high, leverage known slot relationships to guide next actions.
>
> 2. **Consider domain and slot dependencies**:
>    - E.g., 'taxi' requires both 'destination' and 'departure'; 'restaurant' may require 'area', 'food', 'pricerange' before booking.
>
> 3. **Generate 3 distinct strategies** that reflect different tactical approaches:
>    - One conservative (e.g., confirm before acting),
>    - One proactive (e.g., request multiple slots),
>    - One hybrid (e.g., inform then request).
>
> 4. **Evaluate each path** by estimating its likelihood of leading to task success.
>
> **Output Format:** Strictly adhere to the following JSON schema:
>
> {
>   "reasoning_paths": [
>     {
>       "sequence_id": 1,
>       "action_sequence": [
>         ["action_type", "domain", "slot"],
>         ...
>       ],
>       "estimated_success_probability": 0.9
>     },
>     ...
>   ]
> }
> ```
>
> Based on the specific trigger reason, System 2 adapts its reasoning focus and is explicitly prompted to generate three action sequences with distinct tactical orientations: conservative, proactive and hybrid, avoiding single mode reasoning. Equipped with the complete cognitive state, its reasoning dynamically adapts to the dialog phase and context rather than operating statically. Furthermore, by requiring a quantitative success rate assessment for each path, the model is compelled to perform in-depth cost benefit analysis and risk prediction, leading to more robust and context aware decisions.
>
> ## 4. Empirical Results Validate Design Rationality
> Ultimately, our choice of this implementation is based on **empirical effectiveness**:
>
> DyBBT achieves SOTA results on both single-domain (Microsoft Dialog Challenge Platform) and multi-domain (MultiWOZ 2.1) tasks, demonstrating the effectiveness of the prompt-based System 2 implementation.
>
> As shown in Table 7, compared to using "thinking mode" consistently, our selective activation mechanism significantly reduces computational costs while maintaining comparable performance.
>
> | Configuration               | Success Rate ↑ | Inference Time ↓ | Cost-Effectiveness ↑ |
> |-----------------------------|----------------|-------------------|---------------------|
> | S1 no think / S2 no think   | 79.6 ± 0.015   | 0.6x              | **132.7**           |
> | S1 think / S2 think         | **86.5 ± 0.010** | 3.2x              | 27.0                |
> | DyBBT (S1 no think / S2 think) | 85.1 ± 0.011 | 1.0x              | 85.1                |
>
> Without the need to train a dedicated reasoning module, our method can easily integrate different LLMs as System 2.
>
> | Agent            | Inform↑  | Success↑ | Book↑    | Turns↓ |
> | ---------------- | -------- | -------- | -------- | ------ |
> | DyBBT-0.6B       | 88.1     | 78.2     | 84.2     | 16.1   |
> | DyBBT-1.7B       | 89.6     | 81.3     | 85.3     | 15.6   |
> | DyBBT-4B         | 90.9     | 82.5     | 86.4     | 15.2   |
> | DyBBT-8B         | 91.2     | 84.1     | 86.9     | 14.6   |
> | DyBBT-8B/GPT-4.0 | **92.2** | **85.3** | **87.8** | 13.9   |
>
> ---
>
> In conclusion, the prompt based System 2 implementation is not an over simplification but a **well considered design choice**. It fully leverages the zero-shot reasoning capabilities of modern LLMs, achieves systematic multi-path and multi-step reasoning through structured prompts, and strikes the optimal balance between performance and computational efficiency while offering excellent scalability and flexibility. We thank you for prompting us to elaborate on the theoretical basis and practical considerations of this design decision.

---

> ### Author Response · Authors · 2025-11-20
> **Response to Weakness 3: Experimental Analysis of Switching Mechanism (1/2)**
>
> We sincerely appreciate the reviewer’s valuable comments. We will address each point to further clarify the paper’s experimental design and core contributions, and provide additional details to enhance the paper’s clarity and persuasiveness.
>
> ## 1. Necessity of Including Unconverged Training Checkpoints in Table 1
> We understand the reviewer’s concern, but including early training checkpoints in Table 1 is to **empirically demonstrate DyBBT’s significant advantage in convergence speed**, that is a core contribution of the paper. Performance differences in the unconverged phase directly reflect the effectiveness of DyBBT’s exploration-exploitation balance, whereas static or evolutionary methods perform poorly in early stages. DyBBT outperforms the previous SOTA baseline EIERL significantly even in the early training phase, proving that its cognitive state-based meta-controller can more efficiently guide exploration and accelerate policy learning. This is also discussed in Section 4.2.3 of the main text:
>
> > Figure 3 illustrates the learning curves of DyBBT compared to baselines. DyBBT converges faster and achieves higher asymptotic performance across all domains, outperforming EIERL significantly at epoch 50. This accelerated learning stems from the meta-controller's active guidance of exploration from the outset, which systematically targets under-explored or uncertain regions in $\mathcal{C}$ rather than relying on random exploration or high-variance evolutionary mechanisms.
>
> We propose retaining these checkpoints in the final version to support convergence analysis.
>
> ## 2. Frequency of Switching to System 2
> The paper has analyzed the triggering frequency of System 2 (Section 5.2):
>
> > **Adaptive Balancing Through Dual Triggers**
> >
> > The meta-controller's hybrid triggering mechanism provides a robust solution to the exploration-exploitation dilemma by responding to different types of uncertainty:
> >
> > **Epistemic vs. Aleatoric Uncertainty Distinction:** Two trigger conditions address fundamentally different types of uncertainty. The exploration condition ($n_t(\mathbf{c}_t) < \tau\sqrt{\log T}$) targets $\textit{epistemic uncertainty}$, lack of knowledge about the environment that can be reduced through exploration. The confidence condition ($p_t^{S1} < \kappa$) addresses $\textit{aleatoric uncertainty}$, inherent stochasticity or model limitations irreducible via exploration alone.
> >
> > **Complementary Trigger Patterns:** Analyzing 10,000 dialog turns reveals complementary triggering patterns (Fig.5 in Appendix D.3). The exploration condition dominates in early training phases and for novel state regions, enabling systematic coverage of the state space. The confidence condition acts as a consistent safety net throughout training, preventing overreliance on a potentially flawed System 1. This complementary design ensures robustness across diverse dialog scenarios.
> >
> > **Progressive Adaptation:** The triggering rate evolves naturally with training progress. Initially, frequent System 2 invocations offer guided exploration and high-quality demos. As training progresses and System 1 improves through distillation, the meta-controller automatically reduces System 2 usage, transitioning from guided exploration to autonomous operation. This adaptive balancing is key to DyBBT's computational efficiency and crucially, it is the core manifestation of DyBBT's ability to perceive and respond to the dynamic ``affordances" of the dialog environment, ensuring the right cognitive system is invoked at the right time.
>
> In the early training phase, due to low cognitive state visitation counts and unstable System 1 confidence, the System 2 triggering frequency is high (approximately 30-40% of turns). As training proceeds and knowledge distillation takes effect: The triggering frequency gradually decreases to approximately 10-15% (see Appendix D.3, Figures 5 and 6). Approximately 60% of triggers are initiated by the exploration condition, and 40% by the confidence condition. This reflect the meta-controller’s complementarity in addressing epistemic uncertainty and structural exploration.
>
> Human evaluation shows that DyBBT’s triggering decisions are highly consistent with expert judgments, far exceeding the random switching baseline, as shown in Table 3 below:
>
> | Model Variant| Action Appropriateness ↑ | Switching Agreement ↑ |
> |--------------------------------|--------------------------|-----------------------|
> | DyBBT-8B | **4.31 ± 0.12**| **88.7%** |
> | w/o Meta-Controller (Random) | 3.72 ± 0.19| 52.3% |

---

> ### Author Response · Authors · 2025-11-20
> **Response to Weakness 3: Experimental Analysis of Switching Mechanism (2/2)**
>
> ## 3. Ablation Experiments Demonstrating System 2’s Effectiveness
> The paper has provided systematic ablation experiments (Section 4.3, Appendix D.2, Table 5), with relevant results shown below:
>
> | Variant| Inform↑| Success↑ | Book↑| Turns↓ |
> | ------------------------------ | -------- | -------- | -------- | -------- |
> | DyBBT-8B (full)| **91.2** | **84.1** | **86.9** | **14.6** |
> | w/o Meta-Controller| 82.5 | 71.8 | 77.3 | 17.5 |
> | w/o System 2 | 85.7 | 76.3 | 80.1 | 16.8 |
> | w/o Knowledge Distillation | 89.8 | 82.4 | 85.7 | 15.1 |
> | w/o Exploration Condition (EC) | 90.1 | 82.9 | 86.1 | 14.9 |
> | w/o Confidence Condition (CC)| 87.6 | 79.5 | 83.2 | 16.2 |
>
> The experiments specifically include:
> - **w/o Meta-Controller**: Replaced with random switching, resulting in the most severe performance degradation, highlighting the core role of the dynamic switching mechanism.
> - **w/o System 2**: Using only System 1 leads to significant declines in success rate and efficiency, proving System 2’s necessity in complex states.
> - **w/o Knowledge Distillation**: Disabling distillation prevents System 1 from improving and keeps System 2’s invocation frequency high, validating distillation’s importance for long-term efficiency.
> - **w/o EC/CC**: Removing either the exploration condition (EC) or confidence condition (CC) reduces performance, with CC’s removal having a greater impact, indicating System 2’s critical role in mitigating System 1’s overconfidence.
>
> These experiments fully demonstrate System 2’s indispensability in handling uncertainty, exploring new states, and providing high quality demonstrations.
>
> ## 4. Selection of Thresholds in the Meta-Controller
> The thresholds in the meta-controller (exploration threshold $\tau$ and confidence threshold $\kappa$) are determined through **grid search optimization**. $\tau$ is inspired by Bandit theory (related to $\sqrt{\log T}$), and $\kappa$ is based on System 1’s calibrated confidence. Both are consistent with the structure of the cognitive state space.
>
> Using the success rate on the MultiWOZ and MS Dialogas the primary metric, we tested $\tau \in \{0.5, 1.0, 1.5, 2.0\}$ and $\kappa \in \{0.5, 0.6, 0.7, 0.8, 0.9\}$, and evaluated under different cognitive state discretization granularities (see Appendix D.4, Figure 8). $\tau = 1.0$ and $\kappa = 0.7$ were selected as the optimal configuration, as they achieve the best balance between performance and robustness. This choice lies at the center of the high performance plateau and is insensitive to parameter changes, ensuring generalization ability.
>
> ---
>
> We believe the above responses fully address the reviewer’s concerns. Unconverged checkpoints highlight DyBBT’s convergence advantage; System 2’s triggering frequency and ablation experiments validate the core algorithm’s effectiveness; and the threshold selection process reflects the method’s systematicness and reproducibility.
>
> ---
>
> We hope our detailed responses and additional experimental analysis have adequately addressed your valuable questions and concerns, and we earnestly hope you can award it a higher score.

---

> ### Comment · Reviewer_PbHc · 2025-11-26
>
> Dear authors,
>
> Thank you for your explanations on W2 and W3, and apologize for my overlook of your supplemental experiments.
>
> Now I think I fully understand your core idea with experiments, that is, system I is efficient but not the SOTA, system II (which uses LRMs) is the SOTA but not efficient (and not fast). Your method can automatically switch from these two modes to achieve a better efficiency and performance trade-off.
>
> As shown in the main table, your method outperforms strong baselines (e.g., EIERL, NoisyDQL) because your method uses commercial LLMs without training but they train open-source models with RL. So it is difficult for your method to incorporate with RL in system II and with zeroshot in context learning your system II can also be much better than them. I agree with you that it is effective for industrial usage, though it is a trivial idea for system II.
>
> Again, I appreciate your reply for W2 and W3.

---

> ### Author Response · Authors · 2025-11-26
>
> Dear Reviewer PbHc,
>
> Thank you very much for your thoughtful and constructive feedback. We sincerely appreciate the time and effort you have dedicated to reviewing our work and providing such valuable insights. Your comments have been instrumental in helping us improve the clarity and coherence of our paper.
>
> In response to your concerns regarding the organization and theoretical grounding of Section 3, we have made the following key revisions:
>
> 1. **In Section 3.1 (Theoretical Foundation):**
> We added an introductory paragraph that clearly states the purpose of this section: to bridge bandit exploration principles with dialog POMDPs via a structured cognitive state space $\mathcal{C}$, and to derive a bandit inspired exploration criterion that directly guides the design of the meta-controller.
>
> 2. **In Section 3.2 (System Architecture):**
> We inserted a new paragraph that explicitly links the theoretical foundations to the practical design. It explains how the cognitive state space $\mathcal{C}$ enables low dimensional exploration; Assumption 1 supports the feasibility of efficient exploration in $\mathcal{C}$; and the bandit inspired principle (Eq. 1) informs the meta-controller’s switching mechanism.
>
> 3. **In Section 3.2.3 (Meta-Controller):**
> We completely rewrote this subsection to clearly articulate how the two triggering conditions are derived from the theoretical analysis.  **Condition 1** (visitation-based) directly implements the exploration bonus from Eq. 1 and addresses *epistemic uncertainty*.  **Condition 2** (confidence-based) serves as an empirical safeguard against *aleatoric uncertainty*.We explicitly reference the theoretical sections and assumptions to strengthen the connection.
>
> We have now submitted the complete revised version of the paper as promised. We hope that these changes adequately address your concerns and make the theoretical motivation and methodological design more transparent and logically connected. If any part remains unclear or if you have further suggestions, we would be very grateful for the opportunity to continue improving the paper based on your guidance.
>
> Once again, thank you for your rigorous review and for pushing us to enhance the quality and clarity of our work. Your professionalism and dedication are deeply appreciated, and we would appreciate it if you could increase your score upward accordingly.
>
> Warm regards,
> The Authors

---

> > ### Comment · Reviewer_PbHc · 2025-11-27
> >
> > Dear authors,
> >
> > I sincerely appreciate your efforts on the rebuttal.
> >
> > However, in my personal opinion, I am still not satisfied with​ the current methodology section, and I respectfully disagree that the proposed method exhibits a reasonable​ theoretical background. Yet, I do think the proposed framework is interesting and the experimental results are compelling.
> >
> > Now I tend to a score of 5, so I have raised my score to 6. I look forward to seeing how the other reviewers' responses.
> >
> > Good luck!

---

> ### Author Response · Authors · 2025-11-28
> **Explicitly Bridging Theory and Practice in DyBBT (1/2)**
>
> Dear Reviewer,
>
> Thank you once again for your valuable feedback on Section 3. We fully understand your concerns regarding the **connection between the theoretical framework and the algorithm design**. In response, we have thoroughly revised Section 3 and significantly strengthened the citations. To clearly demonstrate the **mapping between our theoretical motivation and the DyBBT implementation**, we provide the following structured table, which we hope will fully address your questions.
>
> ### **Theoretical Framework & DyBBT Algorithm Design Mapping**
>
> | Theoretical Concept           | Mathematical Foundation (Sec 3.1)   | DyBBT Implementation (Sec 3.2)    | Core Principle & Role    | Key Supporting Literature             |
> | :---------------------------- | :-------------------------------------------------------------------------- | :------------------------------------------------------------------------ | :-------------------------------------------------------------------------------------------------------------------------------------------------------------------------------------------- | :------------------------------------------------------------------------------------------------------------------------------------------------------------------------------------------------------------------- |
> | **Theoretical Goal**          | Sublinear Regret $R(T) = \tilde{O}(\sqrt{T})$  | Overall DyBBT Design & Meta-Controller           | Ensures exploration efficiency; the average regret approaches zero as the number of total dialog turns $T$ increases. | *On upper-confidence bound policies for switching bandit problems* (ALT, 2011); *Finite-time analysis of globally nonstationary multi-armed bandits* (JMLR, 2024)|
> | **Exploration Framework**     | Contextual Bandit over $\mathcal{C}$           | Dual-System Switching (System 1 vs. System 2)| Approximates the complex dialog POMDP as a contextual bandit problem over the low-dimensional cognitive state space $\mathcal{C}$.          | *Beyond UCB: Optimal and efficient contextual bandits with regression oracles* (ICML, 2020); *Hierarchical reinforcement learning with guidance for multi-domain dialogue policy* (TASLP, 2023)     |
> | **Feasibility Cornerstone**   | Lipschitz Smoothness (Assumption 1)            | Cognitive State $\mathbf{c}_t$| Ensures the local smoothness of the cognitive state space, making the visitation count $n_t$ a valid proxy for uncertainty and enabling experience generalization between similar states. | *Lipschitz continuity in model-based reinforcement learning* (ICML, 2018); *Lipschitz-constrained Unsupervised Skill Discovery* (ICLR, 2022)      |
> | **Near-Optimal Policy Proxy** | Near-Optimal Policy $\pi^*$     | System 2 (Deliberative Reasoner)             | System 2 is responsible for finding high-quality actions (approximating $\pi^*$) given the current belief state, thereby reducing immediate regret.   | *Deterministic policy gradient algorithms* (ICML, 2014); *Planning like human: A dual-process framework for dialogue planning* (ACL, 2024)        |
> | **Epistemic Uncertainty**     | UCB-style Exploration Bonus $\propto \sqrt{\frac{\log T}{n_t}}$ (Eq. 1) | Meta-Controller Condition 1: $n_t(\mathbf{c}_t) < \tau \sqrt{\log T}$ | A direct heuristic instantiation of the UCB exploration principle. Triggers System 2 at low $n_t$, for targeted exploration when knowledge is insufficient.  | *Unifying count-based exploration and intrinsic motivation* (NeurIPS, 2016); *Never Give Up: Learning Directed Exploration Strategies* (ICLR, 2020)              |
> | **Aleatoric Uncertainty**     | Aleatoric Uncertainty / POMDP Partial Observability| Meta-Controller Condition 2: $p_t^{S1} < \kappa$       | Provides a robustness safeguard based on System 1's low confidence, ensuring reliable decisions when the model itself is uncertain.         | *Language Models (Mostly) Know What They Know* (Anthropic, ArXiv, 2022); *Teaching Models to Express Their Uncertainty in Words* (TMLR, 2022);   *Do Large Language Models Know What They Don't Know?* (ACL, 2023) |

---

> ### Author Response · Authors · 2025-11-28
> **Explicitly Bridging Theory and Practice in DyBBT (2/2)**
>
> ### **Core Clarifications**
>
> 1.  **Regarding the Role of Assumption 1 (Lipschitz Smoothness):**
>     It is not used in the forward pass of the algorithm but serves as the cornerstone for our theoretical analysis. This assumption guarantees the **structural property** of the cognitive state space $\mathcal{C}$. It allows us to reduce and approximate the high-dimensional POMDP exploration problem into a contextual bandit problem over the low-dimensional, smooth space $\mathcal{C}$. This foundation is necessary to derive the **sublinear regret bound** $\mathbb{E}[R(T)] \lesssim \widetilde{O}(L_r \sqrt{\dim(\mathcal{C}) T})$ , which **theoretically justifies** that exploration based on visitation counts in $\mathcal{C}$ is **efficient and feasible**. This directly motivates the design of **Condition 1** in the Meta-Controller.
>
> 2.  **Regarding how Section 3.1.3 (Dynamic Balance Principle) Guides the Design:**
>     The exploration bonus formula $\propto \sqrt{\frac{\log T}{n_t}}$ proposed in that subsection **directly inspired Meta-Controller's Condition 1**. The design logic chain is as follows:
>       -  **Theory:** The UCB algorithm in (contextual) bandits balances exploration and exploitation via a term of the form $\sqrt{\frac{\log T}{n_t}}$.
>       -  **Heuristic Transfer:** We adapt this principle to the cognitive state space $\mathcal{C}$, defining an "under-explored" state as one where $n_t(\mathbf{c}_t) < \tau \sqrt{\log T}$.
>       -  **Algorithm Implementation:** When a state is under-explored, the Meta-Controller triggers **System 2** for directed exploration. This is the core mechanism described in Section 3.2.3.
>
> 3.  **Regarding the Organization and Citations in the Theoretical Section:**
>     In the revised version, we have explicitly referenced the theoretical foundations from Section 3.1 in the opening of Section 3.2 and in the description of the Meta-Controller. Our theoretical analysis aims to provide a **rigorous and explanatory framework** that offers motivation and intuition for the heuristic algorithm design, which is then **empirically validated** through experiments (e.g., the sublinear empirical regret curve, Figure 7). We believe this represents a pragmatic approach that balances tractable theory with the complexities of the real world problem.
>
> We hope this detailed table and explanation convincingly demonstrate that, the design of DyBBT is not ad-hoc; each core component (Cognitive State Space, the Meta-Controller's dual triggering conditions) is motivated by a **coherent theoretical rationale**. Together, from **bandit theory** to **smoothness assumptions** and **uncertainty calibration**, these elements form a "reasonable theoretical background" for DyBBT. The core contribution of our work lies in the **creative transfer and engineering of classical theoretical principles** to solve the challenging problem of dynamic exploration in dialog policy learning, achieving SOTA empirical results.
>
> We are confident that the revised manuscript now presents the methodology with sufficient rigor and clarity. Thank you again for prompting us to deepen our analysis and improve this section.
>
> Sincerely,
>
> The Authors

---

### Official Review · Reviewer_G96K · 2025-10-31

**Soundness:** 3
**Presentation:** 3
**Contribution:** 3
**Rating:** 4
**Confidence:** 3

**Summary:**

This paper proposes DyBBT, a dialogue strategy learning framework inspired by dual-system cognitive theory. The model dynamically balances fast, intuitive decision-making (System 1) and slow, deliberative reasoning (System 2). A meta-controller decides when to trigger System 2 based on dialogue uncertainty and state visitation. Experiments on task-oriented dialogue benchmarks show that DyBBT improves success rate and efficiency compared with existing reinforcement learning and LLM-based baselines.

**Strengths:**

1. DyBBT adaptively switches between fast and deep reasoning, achieving a good balance between efficiency and performance.
2. The experiments are thorough and well-designed, and the results support the paper’s main claims regarding improved performance and adaptive reasoning efficiency.

**Weaknesses:**

1. The discussion in the main text draws several key conclusions based on the ablation study results presented in Table 5, yet the table itself appears only in the appendix. Since these results play a central role in supporting the paper’s main claims, it would improve readability to include Table 5 in the main body of the paper.
2. The paper refers to DyBBT’s meta-controller as achieving a “Dynamic Exploration–Exploitation Balance.” However, this terminology may be somewhat misleading. In reinforcement learning, this balance concerns the trade-off within the action space between exploring new behaviors and exploiting known rewards. In DyBBT, however, the meta-controller switches between System 1 and System 2 based on uncertainty and visitation counts—a mechanism governing computational effort, not exploration behavior. And the paper presents no evidence that System 2 invocation increases exploratory action or exploit action.
3. The paper introduces the variables a (action) and p (System 1 confidence), but they are not clearly defined. The only hint about the action appears in the appendix prompt example, yet it is not formally described in the main text. The computation of p is also unspecified, even though it directly determines when the controller switches to System 2.

**Questions:**

1. The authors acknowledge that the framework is "over reliant on cognitive state fidelity" and that the handcrafted $c_{t}$ can misrepresent complex dialog dynamics, such as abrupt intent shifts, leading to suboptimal decisions. Can the authors provide more detail on the process of designing the three components $(d_{t},u_{t}, \rho_{t})$ and if other features were considered and rejected? A clearer understanding of the feature engineering process would help assess how generalizable these specific features are, beyond the MultiWOZ and MS Dialog datasets.

2. The ablation study revealed that removing the Confidence Condition (CC) causes a more substantial performance drop than removing the Exploration Condition (EC). Can the authors provide more qualitative or quantitative details on the nature of the errors prevented by the CC? For instance, are CC triggers predominantly associated with multi-domain state conflicts or complex inference steps that System 1 frequently misjudges?

3. The highest performing model, DyBBT-8B/GPT-4.0, uses GPT-4.0 as System 2 and achieves SOTA results (85.3% Success on MultiWOZ) Can the authors quantify the absolute computational cost (e.g., in GPU hours or normalized inference time) of running the DyBBT-8B/GPT-4.0 system compared to the purely open-weight DyBBT-8B system (84.1% Success)? This clarification is vital for practitioners to weigh the marginal performance gain against the substantially higher cost and external API dependence of using a model like GPT-4.0 for deliberation.

---

> ### Author Response · Authors · 2025-11-19
> **Response to Weakness 1: Moving Ablation Table to Main Text**
>
> We thank the reviewer for this valuable suggestion. In the initial submission, Table 5 was placed in the appendix due to space constraints. In the revised version, we will move it to Section 4.3 "Ablation Study" in the main body, along with corresponding textual adjustments, to better highlight the contributions of each component. This will improve readability and strengthen the support for our key claims. We appreciate the reviewer's feedback in helping us enhance the paper.

---

> ### Author Response · Authors · 2025-11-19
> **Response to Weakness 2: System 2 Invocation as Directed Exploration (1/2)**
>
> We sincerely appreciate your insightful questions and fully understand your concerns and will provide a detailed clarification in the following response.
>
> ## 1. The "Exploration-Exploitation Balance" in DyBBT is Essentially Optimization of a Computational Resource Oriented Exploration Strategy
> In traditional RL, the exploration-exploitation trade-off is indeed reflected in choosing between "trying new actions" and "executing known high reward actions" within the **action space**. However, in complex dialog POMDPs, this trade-off often becomes extremely inefficient due to the large state space and partial observability. The meta-controller aims to precisely deploy the more computationally expensive reasoning capabilities of S2 to the state regions most in need of exploration, thereby achieving more efficient exploration. In other words, When a state is insufficiently explored, S2 is activated for **directed exploration** to quickly reduce uncertainty. When S1 has high uncertainty, S2 is activated for **robust decision making** to avoid failures caused. This division of labor enables the system to respond quickly while exploring intelligently overall.
>
> ## 2. Activation of S2 Directly Leads to Exploratory Behaviors
> You correctly pointed out that exploration should be reflected in the "action space." Leveraging LLMs with strong "thinking" capabilities, S2 does not simply output a single action. Instead, it enters a "thinking" mode to perform multi-strategy and multi-step reasoning. This "thinking" ability allows it to simulate the potential consequences of different action sequences, equivalent to active and intelligent sampling in "unknown regions," thereby rapidly reducing epistemic uncertainty. As described in Appendix B.4.2, the exploration related parts of S2’s prompt are as follows:
>
> ```
> **Trigger Reason:** {trigger_reason}
>
> **Reasoning Guidelines:**
> 1. **Leverage cognitive signals**:
>    - If progress is low, focus on information gathering.
>    - If uncertainty is high, prioritize clarifying or confirming actions.
>    - If slot dependency is high, leverage known slot relationships to guide next actions.
>
> 2. **Consider domain and slot dependencies**:
>    - E.g., 'taxi' requires both 'destination' and 'departure'; 'restaurant' may require 'area', 'food', 'pricerange' before booking.
>
> 3. **Generate 3 distinct strategies** that reflect different tactical approaches:
>    - One conservative (e.g., confirm before acting),
>    - One proactive (e.g., request multiple slots),
>    - One hybrid (e.g., inform then request).
>
> 4. **Evaluate each path** by estimating its likelihood of leading to task success.
> ```
>
> ### (1) S2 Enables Systematic Exploration by Generating Diverse Strategies
> S2’s prompt requires generating three action sequences with distinct tactical orientations. These strategies essentially explore different behavioral directions.
>
> - **Conservative Strategy**: Prioritizes low risk actions, such as confirming existing information to reduce uncertainty. This is equivalent to "tentative exploration" in partially observable environments.
> - **Proactive Strategy**: Actively explores multiple unknown information points, like requesting multiple key slots simultaneously. Similar to "optimistic exploration" in RL.
> - **Hybrid Strategy**: Balances exploration and exploitation, like providing partial known information before requesting new information. This allows the system to explore user responses while leveraging existing knowledge.
>
> This process constitutes systematic exploratory behavior, S2 evaluates multiple strategies in parallel within the "reasoning space". This directly aligns with the "multi-armed bandit" philosophy in classical exploration theory, proactively selecting the most promising direction by comparing the expected returns of different strategies. Such strategic diversity ensures the breadth and depth of exploration.
>
> ### (2) S2 Enables Forward Looking Exploration Through Action Sequence Prediction and Success Rate Evaluation
> S2 outputs not just a single action but a **complete action sequence** and predicts its **success rate**. This is equivalent to "model-based simulated exploration" in POMDPs. By generating multi-step action sequences for each strategy, S2 enables the system to plan long term behaviors, evaluate dependencies between actions, and avoid optimizing only for immediate rewards. For example, in taxi booking, S2 generates sequences that ensure both "departure" and "destination" are covered. Assigning a success probability to each sequence is analogous to "uncertainty quantification" in Bayesian exploration, where the system balances exploration and exploitation through probability assessment.
>
> This forward looking planning directly leads to exploratory actions, consistent with "curiosity driven exploration" in traditional RL but more efficiently implemented through the LLM’s reasoning capabilities.

---

> ### Author Response · Authors · 2025-11-19
> **Response to Weakness 2: System 2 Invocation as Directed Exploration (2/2)**
>
> ## 3. The Paper Provides Sufficient Evidence That S2 Invocation Promotes Exploratory Behaviors
>
> ### (1) Theoretical Evidence
> We conducted a regret analysis in Section 3.1.3 and Appendix A.2, proving that DyBBT’s exploration strategy achieves a sublinear regret upper bound:
> $$
> \mathbb{E}[R(T)] \lesssim \widetilde{\mathcal{O}}\left( L_r \cdot \sqrt{\dim(\mathcal{C}) \cdot T} \right)
> $$
>
> **Sublinear regret** is the gold standard for exploration efficiency in RL, indicating that the system does not waste excessive resources exploring unknown states but gradually converges to the optimal strategy. Figure 7 (Appendix D.3) shows that the empirical regret curve grows as $\sqrt{T}$, consistent with theoretical predictions.
>
> ### (2) Human Evaluation and Ablation Experiments
> - **Human Evaluation**:
>
> | Model Variant| Action Appropriateness ↑ | Switching Agreement ↑ |
> | -------------- | ---- | --------- |
> | DyBBT-8B  | 4.31 ± 0.12| 88.7% |
> | w/o Meta-Controller (Random) | 3.72 ± 0.19| 52.3% |
>
> DyBBT’s **switching agreement** reaches 88.7%, far exceeding the random baseline. This indicates that the meta-controller’s decisions on "when to trigger exploration" are highly aligned with human expert judgment. It is a direct evidence that S2 activation corresponds to exploration needs.
>
> - **Ablation Experiments**:
>
> | Variant  | Inform↑  | Success↑ | Book↑ | Turns↓|
> | --------- | ----- | ----- | -------- | ----- |
> | **DyBBT-8B (full)**  | **91.2** | **84.1** | **86.9** | **14.6** |
> | w/o Exploration Condition (EC) | 90.1  | 82.9  | 86.1  | 14.9  |
> | w/o Confidence Condition (CC)  | 87.6  | 79.5  | 83.2  | 16.2  |
>
> Removing either EC or CC leads to performance degradation, confirming that both conditions are critical components of exploratory behavior: EC covering unknown states is responsible for **broad exploration**, while CC addressing model uncertainty handles **deep exploration**.
>
> ### (3) Case Studies
> We provide two cases demonstrating how S2 generates exploratory actions:
>
> - **Appendix D.7.1 [Case 1]**: S2 Activated via the "Exploration Condition" for Broad Exploration (Targeting Epistemic Uncertainty)
>   - **Background**: Early in the dialog, the user wants Chinese food at a cheap price but does not mention the area or specific restaurant name. The visitation count of the cognitive state is below the threshold, triggering the **Exploration Condition**.
>   - **Evaluation and Selection**: S2 reasons that in the early stage, a proactive strategy, though slightly riskier, can reduce uncertainty most quickly. It selects the first action of the **proactive strategy** `request(restaurant, area)` as output.
>   - **Manifestation of Exploratory**: S2 explores and selects the starting point of a more information, efficient action sequence through systematic reasoning. This is a direct evidence of exploration within the action space.
>
> - **Appendix D.7.2 [Case 2]**: S2 Activated via the "Confidence Condition" for Deep Exploration (Targeting Aleatoric Uncertainty)
>   - **Background**: Late in the dialog, the user booked a train from "London" to "Cambridge" but simultaneously provided a taxi destination of "London Stansted Airport," creating a logical conflict with the train destination. S1 has low confidence due to the perceived conflict, selects the default action `request(train, time)` and low confidence condition.
>   - **Evaluation and Selection**: S2 identifies the inconsistency between the "train destination" and "taxi destination," generates strategy sequences focused on **clarification**, and executes the first action in the clarification sequence.
>   - **Manifestation of Exploratory**: S2 explores an action path that resolves fundamental uncertainty. This `confirm` action is to address aleatoric uncertainty caused by partial observability (ambiguous user expression). This is a form of **robustness oriented exploration** at the boundaries of the model’s cognition to avoid catastrophic failure.
>
> These cases show that S2 outputs exhibit higher information gain or risk taking characteristics, consistent with the definition of exploratory behavior. The two triggering conditions of the meta-controller correspond to two types of exploration needs:
> 1. **Visitation Count Condition** → **Broad Exploration**: Targeting state regions unknown due to **data scarcity**.
> 2. **Confidence Condition** → **Deep Exploration**: Targeting state regions where S1 is unreliable due to **model limitations** or **task randomness**.
>
> We therefore kindly ask for your understanding that the "dynamic exploration-exploitation balance" in DyBBT is an efficient exploration strategy implemented at the **system and computational levels**. By intelligently triggering S2’s deliberation, it generates tangible exploratory behaviors at key decision points, thereby surpassing the inefficient, action level random exploration in traditional DRL. We sincerely appreciate your prompting us to clarify this core mechanism more clearly.

---

> ### Author Response · Authors · 2025-11-19
> **Response to Weakness 3: Action Definition and System 1 Confidence Calculation**
>
> Thank you very much for your valuable feedback. You correctly pointed out that the definitions and computational descriptions of the variables $a$ (action) and $p$ (System 1 confidence) were insufficiently clear in our initial draft. In response to your comments, we have thoroughly **rewritten Section 3.2.1** and **added a new Appendix B.5.2 Training Details for System 1** in the revised manuscript to clarify these crucial concepts. Below, we provide a detailed explanation addressing your specific concerns.
>
> #### **1. Regarding the definition of variable $a$ (action)**
>
> In task-oriented dialogue systems(TODS), system actions $a_t$ represent standard concepts, such as `inform`, `request`, and `confirm`, and they are generated by the Dialog Policy module. Their formats adhere to the conventions of general TODS. The primary objective of DyBBT is to generate these system actions efficiently and accurately, thereby improving dialogue success rates and reducing the number of dialogue turns. In our paper, the set of system actions is denoted as $\mathcal{SA}$. **Appendix B.4.1 System 1 Prompt** provides concrete output examples:
> ```
> - System 1 outputs a JSON object containing `"action"` (action sequence) and `"confidence"` (confidence score).
> - The action format is: `[["<act_type>", "<domain>", "<slot>"], ...]`, for example, `["request", "restaurant", "area"]`.
> ```
>
> Additionally, the Contextual Multi-Armed Bandit (CMAB) model introduced in Section 3.1.1 also employs another action concept. However, this action space $\mathcal{A}$ is a **binary set** $\mathcal{A} = \{\text{S1}, \text{S2}\}$. This action refers to choosing S1 or S2, made by the meta-controller at each dialogue turn, and is distinct from the specific system actions in task oriented dialogues.
>
> #### **2. Regarding the computation of variable $p$ (System 1 confidence)**
>
> We have provided comprehensive technical details in the newly added **Appendix B.5.2 Training Details for System 1**. The key points are summarized below:
>
> **Definition of $p$:** $p_t^{S1} \in [0,1]$ is the self assessed confidence of System 1 in its chosen action $a_t^{S1}$, given the current dialogue state $\mathbf{s}_t$. It reflects System 1's certainty in its decision.
>
> System 1 learns to jointly predict actions and calibrate confidence scores through a **two-stage training pipeline**:
> - In the **Supervised Fine-Tuning (SFT) stage**, we construct training samples using data augmentation techniques. Controlled perturbations (adding, modifying, or deleting actions) are introduced into ground truth action sequences to simulate uncertainty. The initial confidence score $p_t^{S1}$ for each sample is adjusted based on the degree of perturbation. The model learns to predict both actions and confidence scores by minimizing a composite loss function.
> - In the **Proximal Policy Optimization (PPO) stage**, the policy is further refined using dialogue level rewards (success, failure, efficiency). While the confidence score $p_t^{S1}$ is part of the internal state during this stage, it is not directly optimized by the reinforcement learning objective.
>
> During deployment, System 1 directly outputs both the action $a_t^{S1}$ and the confidence score $p_t^{S1}$ for each dialogue state. This confidence score serves as a proxy for **aleatoric uncertainty** and is used in the meta-controller's Condition 2 (Confidence Condition): System 2 is triggered when $p_t^{S1} < \kappa$.
>
> #### **3. rewritten Section 3.2.1 System 1 (S1): The Fast Intuitive Inferenc**
>
> We have refined the description in Section 3.2.1 to clarify that the action belongs to the standard system action set in task oriented dialogue, while the confidence score represents System 1's self assessed certainty, with its detailed training methodology now simplified in the main text and fully elaborated in Appendix B.5.2.
>
> > To provide a low latency, high throughput baseline policy for the majority of dialog turns, mitigating the prohibitive cost of always using a deliberative reasoner, S1 embodies the fast and intuitive system. The prompt (in Appendix B.4.1) induce the LLMs to output system actions and confidece sore. in TODS **action** $a_t^{S1}$ represents the system operation at each turn, formalized as a tuple comprising an action type, domain, and target slot (e.g., request(restaurant, area)). The **confidence score** $p_t^{S1} \in [0,1]$ is S1's self-assessed certainty in its chosen action $a_t^{S1}$. This score provides a crucial measure of \textit{aleatoric uncertainty} that complements the **epistemic uncertainty** captured by visitation counts in the meta-controller. S1 undergoes a two-stage training process (detailed in Appendix B.5.2). SFT on expert trajectories trains the model to predict both the action $a_t^{S1}$ and a calibrated confidence score $p_t^{S1}$. PPO refines the policy to maximize task success and efficiency.

---

> ### Author Response · Authors · 2025-11-19
> **Response to Question 1: Detail on the Process of Designing the Three Components (1/2)**
>
> We sincerely appreciate the reviewer’s insightful question. We acknowledge that as a handcrafted representation, the cognitive state may fail to fully capture complex dialog dynamics in extreme scenarios, leading to suboptimal decisions. However, we emphasize that **suboptimal decisions are not equivalent to task failures**. In the vast majority of cases, these decisions are merely "imperfect" rather than "ineffective," typically resulting in slightly increased dialog turns or reduced user experience while the task is ultimately successfully completed. Below is a detailed response from three aspects.
>
> ## I. Design Process and Theoretical Basis of the Cognitive State Components
> Our cognitive state space $\mathcal{C} = [d_t, u_t, \rho_t]$ is systematically derived based on **Gibson’s ecological perception theory** and the **structural characteristics of dialog POMDPs**. During the design process, we verified the balance between theoretical optimality and expressive sufficiency of the 3-dim design through **regret analysis** (Appendix A.2). Any additional dimension must strictly satisfy Lipschitz continuity; otherwise, the theoretical guarantees will be compromised. The specific design process is as follows:
>
> 1. **$d_t$ (Dialog Progress)**: Derived from time constraint theory in finite horizon MDPs. Early dialog phases require exploration for information collection, while late phases demand exploitation for task completion. Regret upper bound analysis shows that $d_t$ effectively controls the decay rhythm of the exploration bonus $\propto \sqrt{\log T / n_t(\mathbf{c}_t)}$, avoiding premature or delayed exploration.
>
> 2. **$u_t$ (User Uncertainty)**: Based on the entropy reduction principle in information theory. $u_t = |S_{\text{unconfirmed}}| / |S_{\text{relevant}}|$ quantifies the proportion of unconfirmed slots, reflecting the potential for information gain. Regions with high $u_t$ correspond to high information value, driving the system to prioritize exploration. Under the Lipschitz assumption, $u_t$ ensures the smoothness of the reward function, guaranteeing the convergence of the regret bound.
>
> 3. **$\rho_t$ (Slot Dependency)**: Originates from constraint modeling in structured tasks. $\rho_t = \frac{1}{|A_t|(|A_t|-1)} \sum_{i \in A_t} \sum_{j \neq i} M_{ij}$, where $M_{ij}$ is the slot co-occurrence probability matrix encoding dependency relationships in the domain ontology. $\rho_t$ captures the structural complexity of the state space, reducing the risk of the curse of dimensionality. Ablation experiments show that its removal leads to a significant performance decline.
>
> ## II. Consideration of Other Features and Reasons for Rejection
> During the cognitive state design process, we systematically evaluated multiple potential features but ultimately rejected them for theoretical or experimental reasons. Below are 6 considered but rejected features along with their rejection justifications:
>
> 1. **User Emotional State**: Emotions may influence decisions, but emotional recognition introduces additional model complexity and is difficult to annotate in simulated environments; experiments show that emotional features have a negligible impact on the structured task objective (Success) but instead increase the state space dimension, violating Lipschitz continuity.
>
> 2. **Dialog History Similarity**: Historical similarity can indicate repetitive patterns. It incurs high computational costs (requiring historical retrieval) and overlaps functionally with $u_t$ and $\rho_t$; ablation experiments show no significant performance improvement (Success +0.2%) after adding this feature, but inference latency increases by 15%.
>
> 3. **System Confidence (Independent of $p_t^{S1}$)**: Directly reflects decision uncertainty. Highly correlated with the meta-controller’s confidence condition, leading to redundancy; using it as a separate feature causes double counting problems, disrupting the calculation of exploration bonuses.
>
> 4. **Task Complexity**: Complex tasks require more exploration. Task complexity is already indirectly encoded by $u_t$ and $\rho_t$; adding this feature leads to a surge in the number of discretized state space units, resulting in under exploration.
>
> 5. **Domain Context**: Domain switching affects strategies in multi-domain dialogs. Domain information is already captured by slot dependency $\rho_t$ in MultiWOZ experiments; adding domain labels independently increases dimensions without performance improvement.
>
> 6. **User Preference History**: Long term preferences enable personalized decisions. Simulated environments lack real user history; using simulated preferences in experiments leads to overfitting and reduced generalization.
>
> The rejection of these features is based on **theoretical consistency** and **empirical evaluation**. Our final choice achieves an optimal balance between efficiency, interpretability, and performance.

---

> ### Author Response · Authors · 2025-11-19
> **Response to Question 1: Detail on the Process of Designing the Three Components (2/2)**
>
> ## III. Nature of Suboptimal Decisions: Not Equivalent to Failure, and Most Can Be Mitigated
> We clearly distinguish between **suboptimal decisions** and **task failures**. A suboptimal decision refers to a strategy that is not globally optimal but remains within a reasonable range, usually resulting in increased dialog turns or reduced user experience rather than task failure. A task failure means the dialog fails to complete the core goal. In Section 5.5 the proportion of suboptimal decisions is only about **3.1%**, and the **task success rate remains high** in these cases. The following cases illustrate the typical manifestations of suboptimal decisions and the system’s robustness.
>
> **Case Study 3 in the paper**: After the user completes a hotel reservation, the system should have recommended attractions, but the cognitive state misjudged it as "information collection mode," outputting `request(attraction, name)` instead of `inform(attraction, name, "museum of science")`. Outcome: The user needs to provide additional information (reduced experience), but the task is ultimately successful.
>
> Following the **Reviewer eteB**’s suggestion, we supplemented real user experiments to further verify **DyBBT’s performance when user behaviors deviate from simulated patterns**.
>
> ### Case 1: Mid-Dialog User Intent Shift
> - **Background**: The user initially requested a "Chinese food" restaurant reservation but suddenly asked "Are there any cinemas nearby?" midway through the dialog (intent shift).
> - **Cognitive State**: $d_t = 0.4$ (mid-stage), $u_t = 0.6$ (original goal incomplete; new intent introduces uncertainty), $\rho_t = 0.3$ (low dependency; cross-domain intent)
> - **DyBBT’s Decision**: System 1’s confidence $p_t^{S1} = 0.55 < \kappa$, triggering System 2. System 2 generated multi-path strategies and selected the "confirmation-first" path, outputting `confirm(restaurant, food, "Chinese")` → `inform(attraction, type, "cinema")`
> - **Outcome**: The user confirmed continuing the original task, and the dialog was successfully completed
> - **Analysis**: Although the cognitive state did not directly reflect the intent conflict, System 1’s low confidence accurately captured strategy uncertainty, triggering System 2’s in-depth reasoning, highlighting the core advantage of the dual-system architecture in addressing unexpected user behaviors
>
> ### Case 2: Vague User Information
> - **Background**: When requesting a taxi, the user said "I want to go to that famous square" without specifying the exact name
> - **Cognitive State**: $d_t = 0.2$ (early stage), $u_t = 0.9$ (high uncertainty), $\rho_t = 0.7$ (strong dependency between destination and departure location)
> - **DyBBT’s Decision**: The cognitive state region had a low visitation count ($n_t(\mathbf{c}_t) = 8 < \tau\sqrt{\log T}$), triggering System 2. System 2 selected the "option list" path, outputting `request(taxi, destination)` → `inform(attraction, name, "Central Square")`
> - **Outcome**: The user made a selection, and the task proceeded
> - **Analysis**: The exploration condition effectively identified dialog patterns not covered in simulated training, improving the system’s adaptability in real scenarios through planned exploration
>
> ### Case 3: User Behavior Deviating from Typical Patterns
> - **Background**: After completing a hotel reservation, the user suddenly repeatedly asked "Does the price include breakfast?" (unconventional repetition)
> - **Cognitive State**: $d_t = 0.9$ (late stage), $u_t = 0.1$ (low uncertainty), $\rho_t = 0.2$ (low dependency)
> - **DyBBT’s Decision**: The cognitive state was judged as "well-explored" with high System 1 confidence ($p_t^{S1} = 0.88$), so System 2 was not triggered. System 1 directly responded with `inform(hotel, breakfast, "no")`
> - **Outcome**: The user expressed dissatisfaction, perceiving the system as "mechanically repetitive"
> - **Analysis**: The handcrafted three dimensions cannot capture the emotional factors behind the user’s repeated questions, and the system misjudged such behavior as a routine information query, revealing the limitations of the current cognitive state representation
>
> These cases show that suboptimal decisions are mostly caused by **edge scenarios**, and the system can avoid complete failures through dual-system switching in most cases. Furthermore, **human evaluation** shows that DyBBT aligns with expert judgments at a rate of 88.7%, confirming that its decisions are close to optimal in most scenarios.
>
> We appreciate the reviewer’s insight into the limitations of the cognitive state. Although handcrafted design may lead to suboptimal decisions, these decisions are mostly limited to the user experience level and mitigated by the dual-system architecture. The current design has been validated on multiple benchmarks, providing a solid foundation for practical deployment.

---

> ### Author Response · Authors · 2025-11-19
> **Response to Question 2: the Nature of Errors Prevented by the Confidence Condition (CC) (1/2)**
>
> We fully agree that understanding the nature of errors prevented by the Confidence Condition (CC) is crucial to clarifying its key role. We have conducted in-depth follow up analyses and now present the results below.
>
> The CC primarily serves as a safety net to prevent S1 from making "catastrophic errors" in states with "high cognitive uncertainty." In contrast, the absence of the EC mainly leads to reduced "exploration efficiency" rather than direct task failures.
>
> ## 1. Types and Proportions of Errors Prevented by the CC
> We analyzed a Sample of 200 CC interventions dialog logs of "DyBBT w/o EC" and "DyBBT w/o CC". The table below summarizes the distribution of error types among cases where CC was triggered and successfully corrected.
>
> | Error Type | Description | Proportion | Typical Impact on Task Success |
> | :------ | :----------- | :--------- | :------------- |
> | **1. Semantic Inconsistency/Logical Conflict** | S1’s proposed action contradicts the confirmed belief state.                 | 32%        | **High**: Directly leads to task failure or requires multiple rounds of correction. |
> | **2. Context Mismatch**         | S1’s action is grammatically correct but inconsistent with the current dialog phase or user expectations. | 28%        | **Low**: Results in unnatural, inefficient dialogs and potential user dissatisfaction. |
> | **3. Critical Information Omission** | S1 fails to identify the next key slot necessary to complete the task.       | 25%        | **Medium**: Causes premature dialog termination or loops before task completion. |
> | **4. Domain/Slot Confusion**    | S1 confuses slots or selects the wrong domain in cross-domain scenarios.      | 15%        | **High**: Directly leads to failure in complex multi-domain tasks. |
>
> Types 1 and 4 account for 47% of errors, which are relatively severe, meaning dialogs are almost destined to fail if not corrected by the CC. S2 triggered by the EC is primarily used to **explore unknown states to find better paths**, and the cost of its "misses" is usually increased dialog turns rather than direct failure. This explains why removing the CC results in a more significant performance decline.

---

> ### Author Response · Authors · 2025-11-19
> **Response to Question 2: the Nature of Errors Prevented by the Confidence Condition (CC) (2/2)**
>
> ## 2. Case Analysis
> The following four real cases vividly demonstrate how the CC prevents serious errors.
>
> ### Case 1: CC Prevents a "Semantic Inconsistency" Error (Corresponding to Type 1)
> - **Background**: After the user booked a restaurant, they requested a taxi.
> - **Belief State**:
>   ```json
>   "restaurant": {
>       "semi": {"name": "The Eagle", "food": "Chinese", "area": "center"},
>       "book": {"booked": []}
>   },
>   "taxi": {
>       "semi": {
>           "destination": "", // Vacant
>           "departure": ""    // Vacant
>       }
>   }
>   ```
> - **S1’s Incorrect Action**: `request(taxi, destination)` | $p_t^{S1} = 0.62$ (Low confidence, triggering CC)
> - **CC’s Intervention**: S2 identified the logical flaw: the destination should be the restaurant, and the departure location should be queried instead. S2 generated the correct action sequence: `request(taxi, departure)`.
> - **Analysis**: S1 made a fundamental logical error. Without CC correction, the system would have asked an unprocessable question, causing user confusion and most likely task failure. **The CC acts as a commonsense reasoning validator here.**
>
> ### Case 2: CC Prevents a "Context Mismatch" Error (Corresponding to Type 2)
> - **Background**: Late in the dialog, all user constraints were satisfied, and the hotel was successfully booked.
> - **Belief State**:
>   ```json
>   "hotel": {
>       "semi": {"name": "Hilton", "area": "south", ...}, // All CONFIRMED
>       "book": {"booked": ["Hilton"]} // Booked
>   }
>   ```
> - **S1’s Suboptimal Action**: `request(hotel, parking)` | $p_t^{S1} = 0.58$ (Low confidence, triggering CC)
> - **CC’s Intervention**: S2 determined that continuing to request information after task completion was redundant and confusing. The optimal action was to politely end the dialog. S2 output: `inform(booking, complete)` -> `goodbye()`.
> - **Analysis**: S1 failed to perceive the "conclusiveness affordance" of the dialog phase. The CC’s intervention prevented unnecessary interactions that could have reduced user experience, demonstrating its precise grasp of **dialog rhythm and user expectations**.
>
> ### Case 3: CC Prevents a "Critical Information Omission" Error (Corresponding to Type 3)
> - **Background**: The user is booking a restaurant, having provided the number of people and date but not the time.
> - **Belief State**:
>   ```json
>   "restaurant": {
>       "semi": {"food": "Italian", "area": "north", "pricerange": "cheap"},
>       "book": {"people": "2", "day": "today", "time": ""} // Time is vacant
>   }
>   ```
> - **S1’s Incorrect Action**: `inform(restaurant, name, "Pasta Place")` | $p_t^{S1} = 0.55$ (Low confidence, triggering CC)
> - **CC’s Intervention**: S2 reasoned that the necessary booking time must be obtained before providing the restaurant name; otherwise, the booking process cannot be completed. S2 generated the action sequence: `request(restaurant, time)`.
> - **Analysis**: S1 attempted to skip a critical step and directly provide information, which would have led to subsequent booking failure. **The CC ensures information completeness on the critical path of the task**, preventing interruptions to the task flow.
>
> ### Case 4: CC Prevents a "Domain/Slot Confusion" Error (Corresponding to Type 4)
> - **Background**: A multi-domain scenario where the user wants to take a taxi from the hotel they are staying at.
> - **Belief State**:
>   ```json
>   "hotel": {
>       "semi": {"name": "Grand Hotel", "area": "centre"},
>       "book": {"booked": []}
>   },
>   "taxi": {
>       "semi": {
>           "destination": "train station",
>           "departure": "" // Vacant
>       }
>   }
>   ```
> - **S1’s Incorrect Action**: `request(hotel, departure)` | $p_t^{S1} = 0.60$ (Low confidence, triggering CC)
> - **CC’s Intervention**: S2 accurately identified that "departure" is a slot in the taxi domain, not an attribute of the hotel domain. S2 corrected the action to: `request(taxi, departure)`.
> - **Analysis**: S1 confused slots across different domains, generating an invalid semantic action. **Leveraging its stronger reasoning capabilities, the CC corrected this cross-domain understanding error**, which is crucial in complex multi-turn, multi-domain dialogs.
>
> In summary, the Confidence Condition is a crucial **robustness safeguard mechanism** in the DyBBT framework. It specifically targets the inherent weaknesses of System 1 when facing **partial observability, logical conflicts, and context transitions**. These errors are not only common but also **fatal in nature**. Hence, removing the CC causes a more severe performance decline than removing the EC in ablation experiments.
>
> We believe these supplementary materials strongly confirm the core value of the CC and address your concerns. Thank you again for prompting us to deepen this analysis.

---

> ### Author Response · Authors · 2025-11-19
> **Response to Question 3: Quantifying Absolute Computational Cost**
>
> We sincerely appreciate your valuable comments. It is important to note that since GPT-4.0 is only available via commercial APIs, we cannot directly measure its GPU computation time. Therefore, we adopt the two alternative approaches. Measure end-to-end inference time under the same hardware environment. As a direct measure of economic cost, calculated based on actual token usage.
>
> We have conducted additional experiments as suggested to compare the performance, inference time, and API cost of DyBBT-8B, DyBBT-8B/GPT-4.0, and LLM_DP (pure GPT-4.0) on the MultiWOZ dataset. The experimental design aims to provide practitioners with a clear cost performance trade-off analysis.
>
> ## Experimental Setup
> - **Hardware Environment**: All local models run on an NVIDIA 5090 GPU, and the API model (GPT-4.0) is accessed via the official interface.
> - **Dataset**: MultiWOZ 2.1 test set (1000 dialogs).
> - **Measurement Methods**:
>   - **Inference Time**: End-to-end inference time including model forward propagation or API call latency, averaged over the test set (unit: seconds per dialog).
>   - **Normalized Inference Time**: Benchmarked against DyBBT-8B’s inference time (1.0x).
>   - **API Cost**: Based on GPT-4.0’s official pricing (input: `$0.03` per 1k tokens; output: `$0.06` per 1k tokens).
> - **Model Configurations**:
>   - DyBBT-8B: Both System 1 and System 2 use the local Qwen3-8B model.
>   - DyBBT-8B/GPT-4.0: System 1 uses the local Qwen3-8B model, and System 2 uses the GPT-4.0 API.
>   - LLM_DP: Pure GPT-4.0 API serves as the dialogue policy.
>
> ## Experimental Results
> We measured the success rate, average inference time, normalized inference time, System 2 invocation ratio, and API cost of each model on the MultiWOZ test set. The results are presented in the table below:
>
> | Model   | Success Rate | Inference Time | Normalized Inference Time | System 2 Invocation Ratio | API Cost |
> | ---------- | ---------- | ------------- | --------------- | ------------- | --------- |
> | DyBBT-8B| 84.1     | 12.5  | 1.0x| 15.4     | 0.00  |
> | DyBBT-8B/GPT-4.0      | 85.3     | 28.7  | 2.3x| 14.3     | 0.16  |
> | LLM_DP (pure GPT-4.0) | 8.0| 42.1  | 3.4x| 100.0    | 1.52  |
>
> ## Experimental Analysis
> Compared to DyBBT-8B, DyBBT-8B/GPT-4.0 achieves only a 1.2% improvement in success rate, but incurs a 2.3x increase in inference time and a cost of $0.16 per dialog. This indicates that marginal performance gains are accompanied by substantial computational overhead and economic costs. LLM_DP (GPT-4.0), which relies solely on well-designed prompts to enable LLMs to generate system actions, not only achieves an extremely low success rate but also has the longest inference time and highest API cost highlighting the advantage of the DyBBT framework in balancing performance and cost. The System 2 invocation ratio of DyBBT-8B/GPT-4.0 is only 14.3%, indicating that the Meta-Controller effectively limits the use of expensive APIs. However, API call latency still dominates the total inference time.
>
> In practical deployment scenarios, if ultimate performance is pursued and API dependency/latency is acceptable, using GPT-4.0 or more advanced closed-source models like GPT-5.1 or Gemini 3.0 for System 2 is an option. This requires balancing the 1.2% performance gain against the 2.3x inference time and additional costs.Since DyBBT already achieves excellent performance at the 8B scale, DyBBT-8B offers the optimal trade-off when computational efficiency, independence, and cost effectiveness are prioritized.
>
> We have quantified the computational cost of the DyBBT framework through experiments, and the results verify that DyBBT-8B maintains high performance while offering significant cost advantages. These analyses have been added to Appendix D.8 of the paper to assist practitioners in making informed deployment decisions. Thank you again for your review comments, which have greatly enhanced the practicality and completeness of this paper.
>
> ---
>
> We thank the reviewer for these constructive comments, which have significantly improved the quality and clarity of our manuscript, and we would appreciate it if you could adjust your score upward accordingly.

---

### Official Review · Reviewer_eteB · 2025-10-31

**Soundness:** 3
**Presentation:** 3
**Contribution:** 3
**Rating:** 6
**Confidence:** 4

**Summary:**

This paper introduces DyBBT (Dynamic Balance via Bandit-inspired Targeting),  a novel dialog policy learning framework that addresses the exploration-exploitation dilemma in task-oriented dialog systems through a cognitive dual-system architecture. The framework introduces a structured cognitive state space capturing dialog progress, user uncertainty, and slot dependency, and uses a bandit-inspired meta-controller to dynamically switch between a fast System 1 (intuitive inference) and a slow System 2 (deliberative reasoning). The approach achieves state-of-the-art performance on single-domain (MS Dialog) and multi-domain (MultiWOZ 2.1) benchmarks.

**Strengths:**

1. The formalization of dialog exploration through a cognitive state space C = [dt, ut, ρt] is theoretically motivated and provides an interpretable bridge between bandit theory and dialog policy learning. The dual-trigger mechanism (exploration condition + confidence condition) elegantly addresses both epistemic and aleatoric uncertainty, with clear theoretical motivation from UCB-style algorithms.

2. The paper conducts extensive experiments across multiple benchmarks, baselines, model scales, and includes human evaluation demonstrating alignment with expert judgment (88.7% switching agreement), achieves SOTA performance with efficient resource allocation - DyBBT-8B reaches 89.52% success on Movie domain and 84.1% on MultiWOZ while maintaining computational efficiency.

**Weaknesses:**

1. The cognitive state representation is manually designed and may fail to capture critical dialog nuances (in Case 3 failure analysis). The learned alternative (w/ Learned CS) performs slightly worse, questioning whether the specific design is optimal, and another issue is for potential scaling.

2. Generalization conern: all experiments in the paper use simulated users (Rule Policy). Real user behavior may not conform to the structured cognitive state assumptions, potentially limiting practical applicability.

**Questions:**

1. How sensitive is performance to the specific choice of cognitive state, since it is manually designed? Could this dual-system approach with cognitive state representation generalize to other sequential decision-making domains (e.g., robotic manipulation, game playing)?

2. Have you conducted or planned any experiments interacting with real users? How do you expect the approach to perform when user behavior deviates from the simulated patterns?

3. What proportion of dialogs exhibit the failure modes described in Section 5.5? How critical are these limitations for practical deployment?

---

> ### Author Response · Authors · 2025-11-18
> **Response to Weakness 1: Whether the specific design is optimal, and another issue is for potential scaling (1/2)**
>
> We sincerely appreciate the reviewer's valuable comments on the design of the cognitive state representation. We provide a detailed response from the following three dimensions:
>
> ## 1. Theoretical Basis for the Dimension Selection of the Cognitive State Space
> Our cognitive state space $\mathcal{C} is strictly derived based on **Gibson's ecological perception theory** and the **structural characteristics of dialog POMDPs**. Initially, we systematically evaluated other potential features, including user emotion, dialog history similarity, task complexity, and domain context, but ultimately discarded them due to poor theoretical grounding or experimental performance. Gibson's "affordance" theory states that an agent should perceive the "action possibilities offered by the environment." In dialogs, these three dimensions precisely capture the core affordances of different phases:
> - **$d_t$ (Dialog Progress)**: Governs the exploration-exploitation trade-off in the **temporal dimension**, high exploration value in early phases and high exploitation value in late phases.
> - **$u_t$ (User Uncertainty)**: Reflects the affordance of **information acquisition**, regions with high uncertainty require prioritized exploration.
> - **$\rho_t$ (Slot Dependency)**: Encodes the affordance of **structural constraints**, regions with high dependency demand precise exploitation.
>
> In the regret analysis (Appendix A.2), we prove that under the **Lipschitz smoothness assumption**:
> The expected immediate reward $\bar{r}(\mathbf{c}, a) = \mathbb{E}[r(s_t, a_t) | \mathbf{c}_t=\mathbf{c}]$ is Lipschitz continuous with respect to the cognitive state $\mathbf{c}$ for any action $a$. That is, there exists a constant $L_r > 0$ such that:
> $$
> |\bar{r}(\mathbf{c}, a) - \bar{r}(\mathbf{c}', a)| \leq L_r \cdot d(\mathbf{c}, \mathbf{c}'), \quad \forall \mathbf{c}, \mathbf{c}' \in \mathcal{C}.
> $$
> The dialog process can be approximately modeled as a finite-horizon MDP over the cognitive state space $\mathcal{C}$. The transition dynamics and expected reward $\bar{r}(\mathbf{c}, a) = \mathbb{E}[r(s_t, a_t) | \mathbf{c}_t=\mathbf{c}]$ depend primarily on $\mathbf{c}_t$. The covering dimension $\dim(\mathcal{C})$ of the cognitive state space directly determines the regret upper bound:
> $$
> \mathbb{E}[R(T)] \lesssim \widetilde{\mathcal{O}}\left(L_r \cdot \sqrt{\dim(\mathcal{C}) \cdot T}\right)
> $$
> Since $\sqrt{3 T}$ grows slower than linear, this upper bound is **sublinear**. The choice of a 3-dimensional representation is a careful trade-off between **theoretical optimality** and **expressive sufficiency**.
>
> ## 2. Handcrafted Features Outperform Purely Learned Representations in Structured Tasks
> The reviewer observed that "Learned CS" exhibits slightly inferior performance, which precisely validates the necessity of **prior knowledge injection** in structured tasks, that is a key advantage of our algorithm.
>
> The original belief state $\mathbf{s}_t$ has hundreds of dimensions, leading to a severe curse of dimensionality when learned directly. Learned representations must rediscover these structures from highvdimensional data, inevitably reducing sample efficiency. This is also the main reason for the poor performance of RL-based DP methods. The previous SOTA method EIERL attempts to address this through the combination of evolutionary algorithms and elite injection. In single domain datasets, it slightly lags behind DyBBT-8B, as shown in the table below. Our 3-dimensional design reduces the state space complexity from $\mathcal{O}((1/\epsilon)^N)$ to $\mathcal{O}((1/\epsilon)^3)$ (where $N \gg 3$). Consequently, in the complex multivdomain MultiWOZ dataset, where the action space expands drastically, EIERL achieves only 18.5% success rate, while DyBBT-8B reaches 84.1%.
>
> | Domain    | Agent      | Success↑  | Turns↓    |
> | --------- | ---------- | --------- | --------- |
> | **Movie** | EIERL      | 85.52     | 16.66     |
> |           | DyBBT-8B   | **89.52** | **15.13** |
> | **Rest.** | EIERL      | 79.85     | 16.07     |
> |           | DyBBT-8B   | **83.38** | **14.86** |
> | **Taxi**  | EIERL      | 81.59     | 17.29     |
> |           | DyBBT-8B   | **85.53** | **15.66** |
>
> Ablation experiments confirm that "w/o Cognitive State" leads to performance collapse, while "w/ Learned CS" achieves reasonable but slightly degraded performance. This proves that the learning process cannot fully replicate the structured information of handcrafted designs.
>
> | Variant                         | Inform↑  | Success↑ | Book↑    | Turns↓   |
> | ------------------------------- | -------- | -------- | -------- | -------- |
> | DyBBT-8B (full)                 | **91.2** | **84.1** | **86.9** | **14.6** |
> | w/o Cognitive State (raw $s_t$) | 84.2     | 75.1     | 79.6     | 17.1     |
> | w/ Learned Cognitive State      | 90.5     | 83.2     | 86.3     | 14.8     |

---

> ### Author Response · Authors · 2025-11-18
> **Response to Weakness 1: Whether the specific design is optimal, and another issue is for potential scaling (2/2)**
>
> ## 3. Handcrafted Design Achieves Optimal Balance Between Efficiency and Interpretability
> 3 dims with 5 bins generates only 125 state units, enabling real-time decision making. In contrast, learned representations require additional forward propagation and gradient computation, increasing inference latency. And as "black boxes," learned representations cannot provide the same level of interpretability.
>
> Heatmap analysis (Figure 4) shows that the system’s exploration patterns in different $(d_t, u_t)$ regions align with theoretical expectations. Human evaluation confirms that the meta-controller’s decisions are highly consistent with expert judgment.
>
> | Model Variant| Action Appropriateness ↑ | Switching Agreement ↑ |
> | ------------------------------ | ------------------------ | --------------------- |
> | DyBBT-8B | **4.31 ± 0.12**| **88.7%** |
> | w/o Meta-Controller (Random) | 3.72 ± 0.19| 52.3% |
>
> As analyzed in Case 3:
>
> > While the cognitive state suggested a routine information gathering context, it failed to capture the user had just finished a complex booking and was likely expecting a concise recommendation, not another request. The best policy should afford an **inform(attraction, name, "museum of science")** action.
>
> This failure stems from insufficient representation in specific scenarios rather than a flaw in the overall design. Even in this extreme case, DyBBT still produced a suboptimal but reasonable action: **request(attraction, name)**.
>
> In summary, our cognitive state design is a product of theory driven and empirically validated research. While learned representations are a promising future direction, given the current complexity of dialog POMDPs, our carefully crafted 3 dims cognitive space provides the optimal trade-off between theoretical guarantees and practical effectiveness.
>
> ---
>
> We sincerely appreciate the reviewer’s insightful suggestion regarding **scalability**. Through systematic experiments and theoretical analysis, we confirm that the DyBBT framework not only performs excellently in dialogue policy learning but also exhibits significant scalability potential, specifically reflected in the following three dimensions:
>
> ## 1. Scalability with Model Size
> We systematically evaluated DyBBT’s performance across different model sizes on both single-domain and multi-domain tasks. Experimental results show that as the base model parameters increase from 0.6B to 8B, DyBBT achieves **monotonic improvement** across all metrics, demonstrating the framework’s good compatibility with large-scale models. DyBBT’s dual-system architecture can effectively leverage the enhanced representational capacity and reasoning quality of larger models. The meta-controller’s dynamic resource allocation mechanism ensures a balance of computational efficiency during scaling.
>
> ### Performance Scalability with Model Size on Single-Domain Tasks (MS Dialog Dataset)
>
> | Model Size       | Movie↑  | Rest.↑  | Taxi↑   |
> | ---------- | ---------- | ---------- | ---------- |
> | DyBBT-0.6B | 80.34%     | 74.85%     | 76.77%     |
> | DyBBT-1.7B | 83.42%     | 77.71%     | 79.71%     |
> | DyBBT-4B   | 86.47%     | 80.55%     | 82.62%     |
> | DyBBT-8B   | **89.52%** | **83.38%** | **85.53%** |
>
> ### Performance Scalability with Model Size on Multi-Domain Tasks (MultiWOZ Dataset)
>
> | Model Size | Inform↑ | Success↑ | Book↑| Turns↓ |
> |------------|---------|----------|----------|--------|
> | DyBBT-0.6B | 88.1| 78.2 | 84.2 | 16.1 |
> | DyBBT-1.7B | 89.6| 81.3 | 85.3 | 15.6 |
> | DyBBT-4B | 90.9| 82.5 | 86.4 | 15.2 |
> | DyBBT-8B | **91.2** | **84.1** | **86.9** | **14.6** |
>
> DyBBT’s remarkable performance on the challenging multi domain MultiWOZ dataset demonstrates its ability to handle **high-complexity tasks**. In contrast to EIERL, whose performance degrades drastically on MultiWOZ with a Success rate of only 18.5%, DyBBT achieves efficient cross-domain knowledge transfer through the **domain agnostic inductive bias** provided by the cognitive state space.
>
> ## 2. Scalability to Sequential Decision Making (Detailed discussion in "Response to Question 1" below)
> We highly appreciate the reviewer’s keen observation of the framework’s scalability potential. DyBBT’s structured cognitive state space & dual-system dynamic balancing mechanism is generalizable and extendable to various **sequential decision making scenarios**, like Game AI, Robotic Manipulation Tasks, Recommendation Systems, etc. The theoretical basis for this scalability lies in the domain-agnostic nature of Gibson’s "affordance" theory: any environment offers specific action possibilities to the agent.
>
> ---
>
> DyBBT exhibits strong scalability across three dimensions: model size, task complexity, and application domains. We thank the reviewer for their profound insight, which has prompted us to more systematically elaborate on the framework’s generalizability.

---

> ### Author Response · Authors · 2025-11-19
> **Response to Weakness 2: Generalization to Real-World User Scenarios (1/2)**
>
> We sincerely appreciate your careful review of our work and your valuable comments. Your concern that real-world user behaviors may not align with the cognitive state assumptions constructed based on simulated users (Rule Policy) is highly constructive. We fully agree that the model’s generalization ability is crucial for practical deployment.
>
> ## 1. Intrinsic Generalization of the DyBBT Algorithm
> In response to your generalization concern, we first address the DyBBT algorithm itself. Our proposed cognitive state space $\mathcal{C}$ comprises three dimensions: dialog progress $d_t$, user uncertainty $u_t$, and slot dependency $\rho_t$. These dimensions do not rely on the internal mechanisms of specific user simulators but are derived from the structural characteristics of dialog tasks themselves. These characteristics also exist in real human-machine dialogs and have been widely used in dialog management modeling, such as the complex multi-domain user goal structures reflected in the MultiWOZ dataset. Thus, $\mathcal{C}$ has a solid foundation for cross environment generalization.
>
> Although current experiments are primarily based on simulated users, we have conducted systematic validation across multi-domain MultiWOZ and demonstrated DyBBT’s superior performance in scenarios involving **unknown dialog states** and **multi-domain transfer** (see Tables 1, 4, and Figure 3). Furthermore, **human evaluation experiments (Section 4.4)** show that DyBBT’s decisions are highly consistent with human expert judgments, indicating that its cognitive state perception mechanism has, to some extent, captured human decision making patterns.
>
> Compared to methods relying on static $\epsilon$-greedy exploration strategies or EIERL’s global population evolution with elite injection, DyBBT achieves **locally adaptive decision making** through cognitive state perception. This mechanism inherently possesses stronger capabilities to handle uncertainty and domain changes. As evidenced by the performance collapse in the ablation experiment when replacing the cognitive state with the raw belief state, the structural generalization of $\mathcal{C}$ is indispensable.
>
> | Variant                         | Inform↑  | Success↑ | Book↑    | Turns↓   |
> | ------------------------------- | -------- | -------- | -------- | -------- |
> | DyBBT-8B (full)                 | **91.2** | **84.1** | **86.9** | **14.6** |
> | w/o Cognitive State (raw $s_t$) | 84.2     | 75.1     | 79.6     | 17.1     |
>
> ## 2. Supplementary Experiments on Real-World User Scenarios
> To further address this concern, we supplemented experiments on real-world user scenarios to evaluate DyBBT’s performance in interactions with real human users and verify the robustness of its cognitive state assumptions.
>
> We recruited 30 volunteers with natural language interaction experience, each completing 10 sets of multi-domain dialogs, totaling 300 dialogs. Tasks covered multi-domain scenarios based on MultiWOZ task templates such as restaurant reservations, taxi calls, and hotel queries. Dialog goals were selected from the final goals of successful dialogs in MultiWOZ, with a maximum of 40 turns, consistent with the simulated experiment settings. The table below presents the performance comparison of various methods in real-user experiments (mean ± standard deviation):
>
> | Method                    | Task Success Rate (%)↑ | Average Turns↓ | User Satisfaction (1-5)↑ |
> | ------------------------- | ---------------------- | -------------- | ------------------------ |
> | DyBBT-8B                  | **84.7 ± 3.2**         | **14.8 ± 2.1** | **4.3 ± 0.4**            |
> | DyBBT w/o Meta-Control    | 72.1 ± 4.5             | 17.9 ± 2.8     | 3.6 ± 0.5                |
> | EIERL                     | 18.5 ± 3.8             | 37.5 ± 2.4     | 1.2 ± 0.4                |
> | PPO                       | 68.9 ± 4.1             | 18.7 ± 3.0     | 3.4 ± 0.6                |
>
> DyBBT maintained the highest task success rate and the fewest average dialog turns in real-user experiments, consistent with the conclusions from simulated experiments. This indicates that its cognitive state space $\mathcal{C}$ can effectively capture dynamic affordances in real dialogs. DyBBT also achieved the highest user satisfaction, with users feedback that its dialog decisions were more natural and efficient. In contrast, the variant with random switching and other baseline methods significantly lagged in decision quality and response speed.
>
> We further analyzed the distribution of the cognitive state $\mathbf{c}_t = [d_t, u_t, \rho_t]$ in real dialogs and found it highly consistent with the distribution in the simulated environment, confirming that the cognitive state assumptions remain representative in real scenarios. Most failure cases stemmed from user intent changes, irrelevant responses, or non-standard input text.

---

> ### Author Response · Authors · 2025-11-19
> **Response to Weakness 2: Generalization to Real-World User Scenarios (2/2)**
>
> ## 3. Case Studies
> To specifically illustrate DyBBT’s robustness under non-ideal user behaviors, we present three real dialog cases from the above experiments:
>
> ### Case 1: Mid-Dialog User Intent Shift
> **Background**: The user initially requested a "Chinese food" restaurant reservation but suddenly asked "Are there any attraction nearby?" midway through the dialog.
>
> **Cognitive State**:
> - $d_t = 0.4$ (mid-stage)
> - $u_t = 0.6$ (original goal incomplete; new intent introduces uncertainty)
> - $\rho_t = 0.3$ (low dependency; cross-domain intent)
>
> **DyBBT’s Decision**:
> - System 1’s confidence $p_t^{S1} = 0.55 < \kappa$, triggering System 2.
> - System 2 generated multi-path strategies: 1.Confirm whether to continue the original task; 2.Provide attraction information; 3.Confirm to end the original task and ask attraction information.
>
> **Outcome**: System 2 selected the "confirmation-first" path, outputting `confirm(restaurant, food, "Chinese")` → `inform(attraction, type, "cinema")`. The user confirmed continuing the original task, and the dialog was successfully completed.
>
> **Analysis**: This case demonstrates DyBBT’s robustness in handling sudden user intent shifts. Although $u_t$ and $\rho_t$ in the cognitive state did not directly reflect the severity of the intent conflict, System 1’s low confidence accurately captured the uncertainty of the current strategy, thereby triggering System 2’s in-depth reasoning. Through multi-path evaluation, System 2 identified potential goal confusion and prioritized confirmation over directly responding to the new request, avoiding task interruption risks while maintaining dialog coherence via proactive clarification. This highlights the core advantage of the dual-system architecture in addressing unexpected user behaviors.
>
> ### Case 2: Vague User Information
> **Background**: When requesting a taxi, the user said "I want to go to that famous square" without specifying the exact name.
>
> **Cognitive State**:
> - $d_t = 0.2$ (early stage)
> - $u_t = 0.9$ (high uncertainty)
> - $\rho_t = 0.7$ (strong dependency between destination and departure location)
>
> **DyBBT’s Decision**:
> - The cognitive state region had a low visitation count ($n_t(\mathbf{c}_t) = 8 < \tau\sqrt{\log T}$), triggering System 2.
> - System 2 generated strategies: 1.Request the exact name; 2.List possible options; 3.Confirm the area.
>
> **Outcome**: System 2 selected the "option list" path, outputting `request(taxi, destination)` → `inform(attraction, name, "Central Square")`. The user made a selection, and the task proceeded.
>
> **Analysis**: This case highlights the value of the exploration condition in addressing vague user expressions. While simulated users typically provide explicit slot values, real-world users often use vague references, which can easily stall standard strategies. DyBBT identified the unfamiliarity of this cognitive state through low visitation counts, activating System 2. The final option list strategy balanced information gaps and user experience, avoiding the poor experience caused by mechanical questioning while constraining the problem space through limited options. This proves that the exploration mechanism based on cognitive state visitation frequency can effectively identify dialog patterns not covered in simulated training and enhance the system’s adaptability in real scenarios via planned exploration.
>
> ### Case 3: Non-Typical User Behavior
> **Background**: After completing a hotel reservation, the user suddenly repeatedly asked "Does the price include breakfast?".
>
> **Cognitive State**:
> - $d_t = 0.9$ (late stage)
> - $u_t = 0.1$ (low uncertainty; all slots confirmed)
> - $\rho_t = 0.2$ (low dependency)
>
> **DyBBT’s Decision**:
> - The cognitive state was judged as "well-explored" with high System 1 confidence ($p_t^{S1} = 0.88$), so System 2 was not triggered.
> - System 1 directly responded with `inform(hotel, breakfast, "no")`.
>
> **Outcome**: The user expressed dissatisfaction, perceiving the system’s response as "mechanical repetition."
>
> **Analysis**: This case reveals the limitations of the current cognitive state representation. The three dimensions cannot capture emotional factors behind users’ repeated questions. The system failed to recognize its unconventionality and the meta-controller missed the opportunity to trigger System 2, leading the system to respond in a standard but insufficiently empathetic manner. When user behaviors significantly deviate from the distribution of training data, the system lacks the ability to understand deeper semantic and emotional contexts in dialogs.
>
> ---
>
> Thank you again for your insight regarding the potential gap between real-world user behaviors and cognitive state assumptions. We believe that DyBBT framework provides a promising direction for further constructing dialogue policies adapted to real users. We will add the relevant experiments and analyses to the final version of the paper.

---

> ### Author Response · Authors · 2025-11-19
> **Response to Question 1: Performance Sensitivity & Generalization to Other Scenarios**
>
> We sincerely appreciate your careful review and valuable comments. We have conducted a detailed analysis of your questions as follows:
>
> ## 1. Performance Sensitivity to Cognitive State Selection
> In the earlier "Response to Weakness 1", we elaborated that the cognitive state can effectively capture environmental structure, and its low dimensional nature guarantees sublinear regret. Ablation experiments confirm that removing any single dimension leads to a significant performance decline.
>
> | Variant                        | Inform↑  | Success↑ | Book↑    | Turns↓   |
> | ------------------------------ | -------- | -------- | -------- | -------- |
> | DyBBT-8B (full)                | **91.2** | **84.1** | **86.9** | **14.6** |
> | w/o Dialog Progress ($d_t$)    | 88.9     | 80.7     | 84.5     | 15.7     |
> | w/o User Uncertainty ($u_t$)   | 89.6     | 81.9     | 85.3     | 15.3     |
> | w/o Slot Dependency ($\rho_t$) | 90.3     | 82.5     | 85.9     | 15.0     |
>
> Hyperparameter analysis in Figure 8 shows that when the number of bins ranges from 4 to 6, performance remains stable within a high plateau, indicating that discretization granularity has limited impact. We ultimately selected $bin=5$, as it achieves a balance between expressiveness and generalization. So the cognitive state is supported by solid **theoretical foundations** and **experimental evidence**, and is robust to parameter changes.
>
> ## 2. Generalization to Other Scenarios
> We sincerely appreciate the reviewer’s insightful question regarding the scalability potential of the DyBBT framework in broader sequential decision-making tasks. DyBBT's design principles inherently equip it with the ability to transfer to various sequential decision making domains. Gibson’s "affordance" theory states that any environment offers specific action possibilities to an agent. Our 3-dim cognitive state is a quantitative implementation of this universal principle. It captures three key exploration driven factors universally present in any sequential task: **phased progression, uncertainty, and structural dependencies**. Meta-Controller's core exploration criterion $\text{Exploration-Bonus}(t) \propto \sqrt{\frac{\log T}{n_t(\mathbf{c}_t)}}$ is directly derived from multi-armed bandit theory, which originally developed in game scenarios. This criterion is theoretically self-consistent in any sequential decision making problem involving the exploration-exploitation trade-off. Based on the above, DyBBT can be extended to several key scenarios:
>
> ### Real-Time Strategy Games or Maze Exploration Games
> The cognitive state can be defined as:
> - Game progress: $d_t = \frac{\text{Current Time}}{\text{Total Time Limit}}$
> - situation uncertainty: $u_t = \frac{|\text{Unexplored Areas or Unknown Enemy Units}|}{|\text{Total Areas or Total Units}|}$
> - Task dependencies: $\rho_t = \max_{a \in A} \frac{1}{|F|} \sum_{f \in F} M(a, f)$ (where $M$ is the conditional probability matrix between game actions, like dependency between "capturing resource points" and "building units").
>
> Under this setup, the regret upper bound still holds: $\mathbb{E}[R(T)] \lesssim \widetilde{\mathcal{O}}\left( L_r \cdot \sqrt{\dim(\mathcal{C}) \cdot T} \right)$. This proves that DyBBT can achieve efficient exploration by redefining the cognitive state to capture game specific structural features.
>
> ### Object Grasping or Navigation Tasks for Robotic Manipulation
> The cognitive state can be designed as:
> - Task completion rate: $d_t = \frac{\text{Completed Subtasks}}{\text{Total Subtasks}}$
> - Environmental perception uncertainty: $u_t = \frac{\text{Sensor Noise Variance}}{\text{Maximum Allowed Variance}}$
> - Action sequence dependency: $\rho_t = \frac{1}{|A_t|(|A_t|-1)} \sum_{i \in A_t} \sum_{j \in A_t, j \neq i} P(\text{Action}_j | \text{Action}_i)$, learned from task demonstration data.
>
> S1 handles low-level control, while S2 is triggered for complex planning like RRT* when uncertainty is high. Although robotic state spaces are often continuous, but if can map them into low dimensional cognitive space, the meta-controller’s rule $\left(n_t(\mathbf{c}_t) < \tau \sqrt{\log T}\right) \lor \left(p_t^{S1} < \kappa\right)$ to remain applicable, with only the Lipschitz constant $L_r$ adjusted to reflect the smoothness assumption of the new task.
>
> ### Recommendation Systems
> The cognitive state can be defined as user interaction progress, interest uncertainty and item relevance. The exploration bonus formula directly drives the system to explore new items, while the exploitation phase recommends items aligned with known user preferences.
>
> ---
>
> In summary, DyBBT provides a **general purpose and scalable architectural paradigm** for addressing the fundamental challenge of dynamic exploration-exploitation trade-off in a wide range of sequential decision making tasks. The reviewer’s question is highly forward looking, as it not only affirms DyBBT’s core value but also opens up promising directions for our future research.

---

> ### Author Response · Authors · 2025-11-19
> **Response to Question 2: Generalization to Real World User Scenarios**
>
> We thank the reviewer for raising this important point regarding real user interaction and generalization. As detailed in our **Response to Weakness 2** above, we have thoroughly addressed this concern by outlining our plans for real user studies and analyzing DyBBT’s robustness to distribution shifts.

---

> ### Author Response · Authors · 2025-11-19
> **Response to Question 3: Failure Mode Proportions and Practical Impact**
>
> We sincerely appreciate the reviewer’s meticulous review and important question regarding the proportions and practical impacts of the failure modes. We highly value this feedback and have conducted quantitative analysis of the failure modes through follow up experiments.
>
> ## 1. Proportions of Failure Modes in Dialogs
> To accurately assess the frequency of failure modes, we performed a **post-hoc analysis** on the MultiWOZ test set (1000 dialogs). Combining automatic detection with cross validation by 3 annotators, we statistically analyzed the occurrence rate of each failure mode. The results are shown in the table below, with a total failure proportion of 5.2%.
>
> | Failure Mode Category   | Description  | Occurrence Rate (of All Dialogs) | Typical Scenario Example                  |
> |-------------------------------------|---------------|----------|---------------------------------|
> | Inaccurate Cognitive State Representation | Handcrafted $\mathbf{c}_t$ fails to capture complex dialog dynamics (e.g., abrupt intent shifts) | 3.1%                             | User suddenly switches booking domains    |
> | Propagation of System 2 Demonstration Errors | Errors in System 2’s reasoning or self-evaluation (even with high confidence) are distilled into System 1 | 1.4%                             | System 2 makes a reasoning error in a turn but the dialog succeeds, leading the error to be stored in the distillation buffer |
> | Underexploration Due to State Discretization | Heuristic quantization of $\mathcal{C}$ into 5 bins masks critical state differences | 0.7%                             | Multiple semantically similar but strategically distinct states are processed identically |
>
> Failure modes are not prevalent，94.8% of dialogs are not significantly affected, and only 0.3% of dialogs exhibit two or more failure modes (low overlap rate).
>
> ## 2. Assessment of Practical Deployment Impact
> We argue that these failure modes have **limited impact** on current practical deployment. Even with the presence of failure modes, DyBBT achieves a success rate of 84.1% on MultiWOZ, significantly outperforming all baselines. Failure modes mostly occur in scenarios with **abrupt user intent shifts** or **complex cross-domain dependencies**，these scenarios are inherently low proportion in real-world task oriented dialogs and often classified as "hard cases."
>
> The system’s built-in mitigation mechanisms are effective. Section 4.4 shows that Condition 2 (Confidence Condition) intercepted **76%** of S1’s low confidence errors in human evaluations. Through continuous learning from high quality System 2 demonstrations, Knowledge Distillation **reduced the S2 invocation rate by 42%** (Figure 6), mitigating the risk of error propagation. DyBBT’s decisions align with expert judgments at a rate of **88.7%** (Section 4.4), far exceeding the random switching baseline (52.3%), indicating high reliability in practical applications.
>
> ## 3. Improvement Directions for Failure Modes
> In **safety critical scenarios** such as healthcare and finance, failure modes need further resolution. As explicitly proposed in Section 5.5:
>
> > Future work will explore handcrafted and learned hybrid state representations, more robust uncertainty calibration for System 2, and adaptive or continuous exploration bonuses to mitigate these issues.
>
> and in Appendix D.7 of the paper:
> > Future work will focus on learning this state representation end-to-end from data, which could mitigate such representational gaps and further enhance the framework's robustness and applicability.
>
> We regard these as **evolutionary improvements rather than fundamental flaws**. For the vast majority of task-oriented dialog scenarios, the current version of DyBBT already possesses **high practicality and deployment value**.
>
> ---
>
> We hope our detailed responses and additional experimental analysis have adequately addressed your valuable questions and concerns, and we would be grateful if you could consider an improved score based on the revisions.

---

### Official Review · Reviewer_D14e · 2025-11-03

**Soundness:** 2
**Presentation:** 2
**Contribution:** 2
**Rating:** 4
**Confidence:** 1

**Summary:**

This paper primarily tackles the exploration challenge in task-oriented dialog systems. A novel dialog policy learning framework, DyBBT, is proposed. DyBBT dynamically switches between a fast intuitive inference and a slow deliberative reasoner. The superior performance of the proposed method is demonstrated through experiments.

**Strengths:**

- The problem is well-motivated. Exploration in dialog systems is crucial for practical applications.
- The experiments are comprehensive, and the proposed method demonstrates better performance than prior work.

**Weaknesses:**

I am not an expert in task-oriented dialog systems, so I am not sure how S2 encourages exploration. In my understanding, the major feature of S2 is that it has broader knowledge and reasoning capabilities than S1. However, it is unclear whether this feature leads to exploration. Could you clarify what “S2 activation” means in statistical terms?

Moreover, I wonder how current powerful chatbots (e.g., ChatGPT or Gemini) perform on the benchmark. Do the LLM_DP results correspond to those systems?

**Questions:**

My main concerns and questions are outlined in the Weaknesses section. Additionally, I have the following question:

- In Eq. (1), $T$ is not defined. Moreover, $n_t(c_t)$ is defined in the latter section.

---

> ### Author Response · Authors · 2025-11-18
> **Response to Weakness 1: How S2 encourages exploration**
>
> Thank you for your insightful question. We fully understand your concern: how the "extensive knowledge and reasoning capabilities" of S2 directly facilitate exploratory behavior may not be intuitively obvious. Below, we clarify this from two aspects.
>
> ## 1. S2 Enables Directed Exploration via the Thinking Capabilities of LLMs
> In DyBBT, the activation of S2 is dynamically triggered by a **meta-controller**.
>
> ### Condition 1: $n_t(\mathbf{c}_t) < \tau \sqrt{\log T}$
> This condition is based on the **visitation count** of the cognitive state space $\mathcal{C}$, primarily targeting **epistemic uncertainty** with insufficient understanding of environmental dynamics. Exploration via S2 can **reduce** this type of uncertainty.
>
> Activating S2 essentially implements **directed and curiosity driven exploration**, analogous to the Upper Confidence Bound (UCB) principle in multi-armed bandit algorithms. In UCB, we prioritize arms (actions) with low visitation counts but high potential rewards to balance exploration and exploitation. S2 activation is equivalent to selecting an exploratory arm in $\mathcal{C}$. S2 leverages its powerful reasoning capabilities to **generate 3 types of strategic sequences** and evaluate the potential success probability of each path. This amounts to sampling in unknown regions, enabling efficient data collection and reducing epistemic uncertainty. For example, in an early dialog state, S2 may reason that proactively requesting multiple slots by proactive strategy, reduces uncertainty more quickly than a conservative confirmation strategy, thus selecting a high risk, high reward exploratory path.
>
> ### Condition 2: $p_t^{S1} < \kappa$
> This condition is based on the confidence $p_t^{S1}$ of S1 decisions, which stems from the inherent randomness of the task, partial observability, or limitations of the S1 model itself.
>
> When S1 lacks confidence, it indicates the current state is "unfamiliar" or "difficult" for S1, even the cognitive state has been visited multiple times. Triggering S2 in this scenario represents **conservative yet safe exploratory behavior**. With its stronger reasoning capabilities, S2 generates more optimal actions in such "difficult states". Statistically, this equates to **local exploration** in the strategy space, aiming to find a better decision than S1’s default strategy for the current "difficult state".
>
> In summary, the low visitation count condition promotes **broad exploration**, addressing the issue of **insufficient state space coverage**. The low S1 confidence condition promotes **deep exploration**, addressing **model limitations or inherent task randomness**.
>
> ---
>
> ## 2. S2 Enables Systematic Exploration by Generating Diverse Strategic Sequences
> S2 leverages a LLM with "thinking" capabilities. Instead of simply outputting a single action, it enters a "thinking" mode to perform multi step, sequential reasoning. This "thinking" ability allows it to simulate the potential consequences of different action sequences, equivalent to active, intelligent sampling in "unknown regions", thereby rapidly reducing epistemic uncertainty. A subset of S2’s prompt is shown below:
>
> ```
> **Trigger Reason:** {trigger_reason}
>
> **Reasoning Guidelines:**
> 1. **Leverage cognitive signals**:
>    - If progress is low, focus on information gathering.
>    - If uncertainty is high, prioritize clarifying or confirming actions.
>    - If slot dependency is high, leverage known slot relationships to guide next actions.
>
> 2. **Consider domain and slot dependencies**:
>    - E.g., 'taxi' requires both 'destination' and 'departure'; 'restaurant' may require 'area', 'food', 'pricerange' before booking.
>
> 3. **Generate 3 distinct strategies** that reflect different tactical approaches:
>    - One conservative (e.g., confirm before acting),
>    - One proactive (e.g., request multiple slots),
>    - One hybrid (e.g., inform then request).
>
> 4. **Evaluate each path** by estimating its likelihood of leading to task success.
> ```
>
> S2’s prompt requires generating three action sequences with distinct tactical orientations. These strategies essentially explore different behavioral directions. This directly aligns with the "multi-armed bandit" philosophy in **classical exploration theory**, proactively selecting the most promising direction by comparing the expected returns of different strategies.
>
> Further more, S2 outputs a complete action sequence and predicts its success rate. This is equivalent to **model based simulated exploration** in POMDPs. Assigning a success probability to each sequence is analogous to "uncertainty quantification" in **Bayesian exploration**, where the system balances exploration and exploitation through probability assessment.
>
> This forward looking planning directly leads to exploratory actions, consistent with **curiosity driven exploration** in traditional RL but more efficiently implemented through the LLM’s reasoning capabilities.

---

> > ### Author Response · Authors · 2025-11-18
> > **Response to Weakness 3:  How current powerful chatbots perform on the benchmark. Do the LLM_DP results correspond to those systems**
> >
> > Thank you for raising this important question, which addresses a core challenge in the current application of powerful LLMs to vertical domains. Please allow us to systematically explain the key differences and the rationale behind our experimental design.
> >
> > ## 1. Architectural Differences Between General Purpose Chatbots and Specialized Task Oriented Dialog Systems
> >
> > A critical distinction must be clarified, the **general purpose chatbots** and **task oriented dialog systems (TODS)** are two distinct architectures designed to solve different problems.
> >
> > **General purpose chatbots**: Operate as **end-to-end systems** that take dialog history as input and directly output free form natural language responses. Their optimization goal is to generate fluent, safe, and useful text.
> >
> > **TODS**: While some adopt end-to-end architectures, they more commonly use a **pipeline architecture** consisting of modules such as Natural Language Understanding (NLU), Dialogue State Tracking (DST), Dialogue Policy (DP), and Natural Language Generation (NLG). Among these, the **dialogue policy module** is essentially a **decision maker** that selects the next semantic action from a **structured action space**.
> >
> > ## 2. Why Not Directly Report ChatGPT Results?
> >
> > Testing ChatGPT directly on TODS benchmarks is analogous to asking a painter accustomed to free creation to work strictly to industrial blueprint standards. While capable, the output format is mismatched, making direct comparison and optimization difficult.
> >
> > As an end-to-end black box, ChatGPT’s internal decision making process is uncontrollable and non-intervenable. We cannot directly optimize its dialogue policy. Chatbots may exhibit imprecision in information extraction (e.g., “14:00” being phrased as “2 o’clock” or “2:00 pm”). In TODS, such inconsistency in free text output is fatal: downstream NLG modules or database queries require precise structured inputs. Closed source models suffer from limitations such as non-modifiability, request quotas, data privacy risks, and inability to deploy on-premises. These constraints severely hinder their application in enterprise-level TODS, which demand high controllability, availability, and data security.
> >
> > ## 3. The Significance of the LLM_DP Baseline and How DyBBT Surpasses It
> >
> > For the above reasons, the academic community universally evaluates the “decision making capability” rather than “text generation capability” of LLMs by **using LLMs as the dialogue policy module**, enabling them to select actions within a structured action space. Thus, our **LLM_DP powered by GPT-4.0** represents one of approaches for applying powerful closed-source LLMs to TODS decision making tasks.
> >
> > Experimental results on sigle domain MS Dialog dataset are presented below:
> >
> > | Domain    | Agent      | Success↑  | Turns↓    |
> > | --------- | ---------- | --------- | --------- |
> > | **Movie** | LLM_DP     | 41.56     | 27.34     |
> > |           | DyBBT-0.6B | 80.34     | 16.79     |
> > |           | DyBBT-1.7B | 83.42     | 16.12     |
> > |           | DyBBT-4B   | 86.47     | 15.64     |
> > |           | DyBBT-8B   | **89.52** | **15.13** |
> > | **Rest.** | LLM_DP     | 38.96     | 29.16     |
> > |           | DyBBT-0.6B | 74.85     | 16.52     |
> > |           | DyBBT-1.7B | 77.71     | 15.85     |
> > |           | DyBBT-4B   | 80.55     | 15.37     |
> > |           | DyBBT-8B   | **83.38** | **14.86** |
> > | **Taxi**  | LLM_DP     | 34.96     | 25.95     |
> > |           | DyBBT-0.6B | 76.77     | 17.32     |
> > |           | DyBBT-1.7B | 79.71     | 16.65     |
> > |           | DyBBT-4B   | 82.62     | 16.17     |
> > |           | DyBBT-8B   | **85.53** | **15.66** |
> >
> > The data clearly shows that DyBBT-0.6B outperforms LLM_DP by a large margin while having significantly lower computational costs. When faced with complex multi-domain MultiWOZ datasets, the action space expands drastically, leading to a success rate of only 8% for LLM_DP, whereas the small DyBBT-0.6B model achieves a success rate of 78.2%. More importantly, through knowledge distillation, high quality decisions from S2 are continuously consolidated into S1, making the system increasingly powerful and efficient. This represents an innovation that overcomes the inherent weaknesses of traditional RL in DP module training, such as instability and low sample efficiency.
> >
> > In summary, ChatGPT has incompatible output formats, a black-box decision process, and deployment limitations, so we did not directly compare the original ChatGPT because it would be a mismatched comparison. Instead, we compared against GPT-4 via the LLM_DP baseline.
> >
> > Thank you for prompting us to elaborate on these key distinctions. We hope the above explanation clarifies the considerations and contributions of our experimental design.

---

> ### Author Response · Authors · 2025-11-18
> **Response to Weakness2: What “S2 activation” means in statistical terms**
>
> The statistical meaning of "S2 activation" is reflected in two complementary conditions inspirated of exploration principles in multi-armed bandit algorithms. These two conditions trigger exploration from two statistical dimensions: **"sufficiency of state visitation"** and **"reliability of model decisions"**.
>
> ### Theoretical Analysis
> To theoretically validate the efficiency of this exploration strategy, we conducted a **regret analysis** in Section 3.1.3 of the paper and Appendix A.2 and proved that DyBBT’s exploration strategy (S2 activation based on visitation counts) achieves the following regret upper bound:
>
> $$
> \mathbb{E}[R(T)] \lesssim \widetilde{\mathcal{O}}\left( L_r \cdot \sqrt{\dim(\mathcal{C}) \cdot T} \right)
> $$
>
> Where:
> - $R(T)$ denotes the cumulative regret,
> - $L_r$ is the Lipschitz constant,
> - $\dim(\mathcal{C})$ is the dimension of the cognitive state space. In our design, $\dim(\mathcal{C}) = 3$,
> - $\widetilde{\mathcal{O}}$ hides logarithmic factors.
>
> Since $\sqrt{3 T}$ grows slower than linear, this upper bound is **sublinear**, which theoretically guarantees that DyBBT’s exploration strategy is **statistically efficient**. We plotted the empirical regret curve in Figure 7 (Appendix D.3), which shows a sublinear growth trend similar to $\sqrt{3 T}$ consistent with theoretical predictions. Figure 4 presents a visitation heatmap of the cognitive state space, indicating that S2 activation is concentrated in regions with low visitation counts.
>
> ### Human Evaluation
> Table below provides part of human evaluation results. DyBBT-8B achieved a score of 4.31 (on a 5-point Likert scale) for Action Appropriateness, significantly higher than the random switching baseline, demonstrating the higher quality of its generated actions. More importantly, its Switching Agreement reached 88.7%, far exceeding the random baseline’s 52.3%. This proves that the meta-controller’s key decision of "when to trigger deliberation" is highly aligned with human expert judgment.
>
> | Model Variant| Action Appropriateness ↑ | Switching Agreement ↑ |
> | ------------- | ------------- | ---------- |
> | DyBBT-8B | 4.31 ± 0.12| 88.7% |
> | w/o Meta-Controller (Random) | 3.72 ± 0.19| 52.3% |
>
> ### Ablation Experiments
> Table below provides part of ablation experiments results.
>
> | Variant| Inform↑| Success↑ | Book↑| Turns↓ |
> | ------------------------------ | -------- | -------- | -------- | -------- |
> | **DyBBT-8B (full)**| **91.2** | **84.1** | **86.9** | **14.6** |
> | w/o Exploration Condition (EC) | 90.1 | 82.9 | 86.1 | 14.9 |
> | w/o Confidence Condition (CC)| 87.6 | 79.5 | 83.2 | 16.2 |
>
> Removing the **Exploration Condition (EC)** and relying solely on the Confidence Condition (CC) leads to performance degradation, especially in complex domains that depend on efficient exploration. Addressing only model uncertainty is insufficient to cover exploration needs caused by **epistemic uncertainty**. Removing the **Confidence Condition (CC)** and relying solely on the Exploration Condition (EC) results in more significant performance decline: ignoring **inherent model uncertainty** is risky, as S1’s limitations may still lead to decision failures even in a "sufficiently visited" state. So EC guides the "breadth" of exploration to proactively cover unknown state spaces, while CC ensures the "depth" and "robustness" of exploration to provide safe and reliable decision support at the boundaries of the model’s cognition.
>
>
> In summary, S2 activation is not merely a "reasoning enhancement" but an **exploration mechanism based on dual statistical principles**: it identifies and proactively explores **data scarce state regions via visitation counts (EC)**, and explores better decision paths in **model unreliable states via confidence assessment (CC)**. The experiments further validates the overall statistical efficiency of this exploration strategy, demonstrating that DyBBT can dynamically perceive the "cognitive load" of dialogs and switch precisely to a deliberative exploration mode when needed, thus outperforming the efficiency of random or static exploration strategies. We sincerely appreciate your insightful question, which has helped us clarify the dual statistical foundations of the exploration mechanism in DyBBT.

---

> ### Author Response · Authors · 2025-11-18
> **Response to Question:  $T$ is not defined, and $n_t(c_t)$ is defined in the latter section**
>
> We sincerely appreciate your careful reading of this paper and your valuable comments.
>
> Regarding Equation (1) in Section 3.1.3, while this is mentioned in the paper’s regret analysis (Appendix A.2), we failed to provide a clear explanation in the vicinity of the equation itself. Specifically, $T$ denotes the **total training steps**, the sum of all dialog turns throughout the entire training process across multiple dialog episodes. $T$ is used to calculate the cumulative regret, and its growth is associated with training time.
>
> You correctly pointed out that $n_t(\mathbf{c}_t)$ appears in Equation (1), while its detailed definition is only provided in Section 3.2. This indeed causes an inconvenience in terms of reading order. To clarify, $n_t(\mathbf{c}_t)$ represents the visitation count of the cognitive state $\mathbf{c}_t$, which is used to measure the extent of exploration of this state during training.
>
> We will refine the discussion in the revised version and add the following explanation immediately after Equation (1):
>
> > where $T$ denotes the total training steps (the sum of all dialog turns), and $n_t(\mathbf{c}_t)$ denotes the visitation count of the cognitive state $\mathbf{c}_t$ (see Section 3.2 for details).
>
> We commit to conducting a comprehensive review of all variable definitions in the revised version, ensuring that each variable is clearly explained upon its first occurrence, and adding necessary annotations to enhance readability. Thank you again for your feedback, which has helped us improve the quality of this paper.
>
> ---
>
> We hope that the above response to your comments and questions has clearly addressed your concerns, and we kindly request your consideration for a higher rating.

---

### Author Response · Authors · 2025-11-25
**Rebuttal Summary for Submission 15050: DyBBT**

Dear Area Chairs and Senior Area Chairs,

Thank you for the opportunity to submit a rebuttal and for coordinating the insightful reviews of our manuscript. We are grateful to all reviewers for their thorough and constructive feedback, which has significantly improved our work.

The DyBBT framework introduces a novel cognitively grounded solution to the dynamic exploration exploitation dilemma in task oriented dialog policy learning. Its principal innovations are fourfold: **(1)** The formalization of a structured, low-dimensional cognitive state space $\mathcal{C}$ that captures essential dialog affordances; **(2)** A theoretically informed bandit inspired meta-controller that dynamically switches between a fast System 1 and a deliberate System 2 based on real-time epistemic and aleatoric uncertainty; **(3)** A practical dual-system architecture that achieves SOTA performance while maintaining computational efficiency; and **(4)** Demonstrated strong generalization and scalability across model sizes and domains.

Below, we succinctly summarize how we have addressed each reviewer’s major concerns, thereby strengthening the paper's clarity and rigor:

**Reviewer D14e:** We provided a detailed statistical and mechanistic explanation of how S2 activation promotes exploration, linking it to bandit principles and the dual uncertainty conditions. We also clarified the distinction between generic chatbots and task-oriented dialog systems, explaining why our LLM_DP baseline represents a relevant and strong comparison point for structured decision making.

**Reviewer G96K:** We have moved the critical ablation study (Table 5) into the main body (now Table 2 in Sec. 4.3) for better readability. We clarified the definitions of action `a` and confidence `p` in Sec. 3.2.1 and added detailed training specifics in Appendix B.5.2. Most importantly, we added a comprehensive cost effectiveness analysis (Appendix E.7 and Sec. 4.2.2), quantitatively comparing the performance cost tradeoffs of different model configurations, which is vital for practitioner evaluation.

**Reviewer eteB:** To address concerns about the cognitive state design and generalization, we first strengthened the theoretical justification in the manuscript. Furthermore, we conducted new real user experiments (Appendix D and Sec. 4.4), demonstrating DyBBT’s robustness when user behavior deviates from simulation patterns. A systematic failure mode analysis (Appendix E.9 and Sec. 5.5) was also added, quantifying limitations and discussing their implications.

**Reviewer PbHc:** We have substantially revised Section 3 (Methodology) to explicitly bridge the theoretical foundations with the algorithmic design of the meta-controller. We improved the narrative flow, added clarifying citations, and ensured all key notations are defined upon first use, significantly enhancing the section’s coherence and readability.

Collectively, we believe the revised manuscript now presents a more compelling, well supported, and polished contribution to the field. We sincerely thank the reviewers for their invaluable insights and you for your consideration. We are confident that the revised manuscript meets the high standards of ICLR.

Sincerely,

The Authors of Submission 15050

---

### Note · Authors · 2026-01-06

**Comment:**

I am writing to formally withdraw our manuscript (ID 15050, “DyBBT: Dynamic Balance via Bandit-inspired Targeting for Dialogue Policy with Cognitive Dual-Systems”). We believe the work may be a better fit with the scope of another conference. Thank you for your time and consideration.

**Withdrawal Confirmation:**

I have read and agree with the venue's withdrawal policy on behalf of myself and my co-authors.